# Three-dimensional genome reorganization foreshadows zygotic genome activation in *Drosophila*

Noura Maziak [1,2], Yuchen Zhang [3,4], Fabian Groll [1,2], Haley E. Brown [5], Alla Madich [6], Yadwinder Kaur [5], Melissa M. Harrison [5], Jian Zhou [3,4] & Juan M. Vaquerizas [1,2] ✉

How chromatin conformation relates to chromatin state remains a central challenge in genome regulation. Here we present Pico-C, a low-input Micro-C approach that enables high-resolution, temporally resolved three-dimensional genome mapping during early *Drosophila* embryogenesis. Contrary to a prevailing view of a disorganized genome before zygotic genome activation (ZGA), we uncover a dynamic and ordered emergence of chromatin loops during pre-ZGA nuclear cycles. Spatial autocorrelation analysis points to context-dependent regulatory influences on chromatin. Notably, inhibition of transcriptional elongation has site-specific effects, retaining some early loops while weakening insulation at active promoters, suggesting distinct regulatory dependencies. Machine learning models trained on sequence features identify orthogonal, motif-specific contributions to architecture. Co-depletion of the pioneer factors Zelda and GAF leads to factor-specific perturbations in chromatin architecture, further highlighting a modular regulatory logic in genome establishment. Together, our findings reveal that early genome organization is orchestrated by an interplay of overlapping yet separable regulatory inputs.

The early embryo exemplifies the principles of genome regulation, with zygotic genome activation (ZGA) marking a progressive shift toward genomic self-regulation in the embryo[1]. This highly dynamic period offers a unique opportunity to study the establishment of three-dimensional (3D) genome architecture[2]. *Drosophila melanogaster* provides an especially powerful model for this transition, combining a rich genetic toolkit with a deeply characterized developmental trajectory[3,4].

In early *Drosophila* development, the embryo undergoes a series of rapid nuclear divisions known as nuclear cycles (NCs) lacking gap phases, generating >6,000 nuclei at ~2.5 h post-fertilization[5]. Although initially controlled by maternal factors, the embryo begins activating its genome as early as nuclear cycle 8 (NC8), termed the minor wave of zygotic genome activation (minZGA), which progressively increases until the major wave (majZGA) at NC14, where mitosis slows and cellularization begins[5–8]. This transition is mediated by the early action of Zelda (Zld), followed by the GAGA factor (GAF) and chromatin-linked adapter for MSL proteins (Clamp)[9–13]. Concomitantly, the chromatin landscape is shaped through a marked increase in histone modifications and the establishment of heterochromatin via the recruitment of heterochromatin protein 1 (HP1)[14–17].

It is interesting that, despite the crucial roles of early pioneer factors in orchestrating proper development, their removal does not

[1]MRC Laboratory of Medical Sciences, London, UK. [2]Institute of Clinical Sciences, Faculty of Medicine, Imperial College London, London, UK. [3]Lyda Hill Department of Bioinformatics, University of Texas Southwestern, Dallas, TX, USA. [4]Section of Genetic Medicine, Department of Medicine, University of Chicago, Chicago, IL, USA. [5]Department of Biomolecular Chemistry, University of Wisconsin-Madison, Madison, WI, USA. [6]Department of Genetics, University of Cambridge, Cambridge, UK. ✉e-mail: j.vaquerizas@lms.mrc.ac.uk

lead to genome-wide conformational changes. Depletion of Zld or GAF results in only locus-specific changes without global effects on boundaries or loops[18-20]. Abrogation of transcriptional elongation or initiation through treatment with α-amanitin or triptolide causes a drop in insulation but does not restructure the genome[18]. Removal of HP1, while affecting B compartmentalization globally, does not impact A compartments[16]. Similarly, the removal of insulator proteins shows only local, boundary-specific changes[21,22]. These findings point to a 3D genome shaped by a convergence of multiple regulatory inputs.

To disentangle how this setup occurs in the early embryo, we developed Pico-C, a low-input Micro-C method, allowing us to generate high-resolution maps of interphase-staged embryos from NC9 to NC14. These maps capture dynamic chromatin interactions as early as NC9, well before majZGA. Leveraging the granularity of our data, we extracted architectural features with high fidelity and uncovered a diverse chromatin landscape underpinning genome architecture. Spatial autocorrelation analysis further shows that chromatin structure and state are coupled in a context-dependent manner, suggesting that chromatin conformation is shaped by layered regulatory influences.

To probe the basis of this architecture, we inhibited transcriptional elongation with α-amanitin in early embryos, which preserved some early loops but selectively disrupted active promoter-associated features. Next, machine learning models predicted that regulatory partitioning of architecture is underpinned by rich sequence diversity and shaped by orthogonal motif contributions. To evaluate whether such motif-level independence translates experimentally, we co-depleted Zld and GAF in embryos. These embryos showed loss of structure at their respective target loci, supporting our α-amanitin experiments and model predictions and reinforcing separable modes of genome establishment.

Collectively, our results highlight early genome establishment as a modular and dynamic process, sculpted by multiple, converging regulatory cues.

## Results

### Pico-C unveils fine-scale architecture at genome activation

To gain insights into chromatin conformation in NCs preceding ZGA, we developed Pico-C—an improved Micro-C method that requires substantially less input material—allowing precise embryo staging without compromising resolution (Fig. 1a,b). Our method utilizes binding nuclei to concanavalin A beads to circumvent the need for centrifugation and includes a prolonged biotinylation step to maximize biotinylated fragment pulldown (Fig. 1a). Using this approach, we can generate high-quality libraries from as few as ~60,000 nuclei—equivalent to 10 embryos at NC14. We then carefully hand staged embryos from NC9 to NC14, sorting exclusively interphase embryos using a PCNA–GFP marker[18,23], and produced 45 high-quality maps (Fig. 1b). Principal component analysis shows that these maps cluster in a clear developmental trajectory, despite the stages occurring within very short temporal periods (Extended Data Fig. 1a). These maps capture chromatin conformation at high resolution, with NC9 and NC10 at 900-bp and 400-bp resolution, respectively, and NC11–NC14 achieving >200-bp resolution (Extended Data Fig. 1b). In addition, the temporal resolution allows us to detect genome-wide trends in contact probability, revealing shorter-range enrichment at NC9–NC10 and increased interaction frequencies at longer distances (~1 Mb) at stages NC12–NC14 (Extended Data Fig. 1c). Overall, our Pico-C maps achieve the same high-quality metrics as other publicly available Micro-C datasets[24], showing a high fraction of *cis* interactions and uniform coverage across the genome (Supplementary Table 1 and Extended Data Fig. 1d–f).

Unexpectedly, our Pico-C maps reveal a highly dynamic genome before the major wave of ZGA (Fig. 1c). They capture early looping events, such as at the *zen* and *zen2* genes visible as early as NC9, and the formation of complex conformations like the bowtie structure around *ftz* and tethering elements within the Antennapedia gene

complex (ANT-C) (Fig. 1c(i)–(iii), respectively). Consistent with previous findings[25], our data also recapitulate long-range, pre-majZGA loops previously validated by imaging[26] (Extended Data Fig. 1e). In parallel, these maps allow us to capture domain boundaries from NC9 to NC14 (Fig. 1d,e), showing that boundaries are generally gained throughout early development with few stage-specific dynamics, in agreement with previous findings[18]. Aggregate analysis further confirms a progressive strengthening of insulation at these boundaries, as reported previously[18] (Fig. 1d). Despite the general gain of insulation, the resolution of our maps enables the capture of fine-detail chromatin dynamics, including early specific boundaries. For instance, we observe insulation at the promoters of *Elba3* and *Bsg25A*, forming a 3-kb miniature compartment at a *CG11929* exon at NC13, which is reduced by NC14 (Extended Data Fig. 2a). Loop calling using Mustache[27] identified interactions at different resolutions, capturing looping dynamics across multiple distances in early development (Fig. 1f,g). Aggregate analysis shows focal dots at distances ranging from 5 kb to >2 Mb (Fig. 1f). It is interesting that loops display more stage specificity than boundaries despite being predominantly gained at NC14 (Fig. 1g and Extended Data Fig. 2b). Aggregate analysis of NC12-specific loops, for example, reveals a pronounced central enrichment at NC12 compared to NC10 and NC14 (Extended Data Fig. 2b). Our maps also capture a subset of the previously characterized metaloops[28] as well as additional metaloop-like features, with some detectable as early as NC13 (Extended Data Fig. 2c).

Overall, our Pico-C maps show an extensive rewiring of the genome through coordinated conformational changes preceding ZGA at unprecedented temporal and structural resolution.

### Multifaceted chromatin landscape underpins 3D genome

To better understand the regulatory basis of the conformational dynamics that we observed, we re-analyzed hundreds of publicly available datasets including assay for transposase-accessible chromatin using sequencing (ATAC–seq), chromatin immunoprecipitation sequencing (ChIP–seq) and RNA sequencing (RNA-seq) and conducted cluster and ChromHMM analysis[29] (Supplementary Data 3 and 4). Cluster analysis of characterized architectural features at NC14 revealed a heterogeneous set of chromatin-associated profiles (Fig. 2a–d and Extended Data Fig. 3a–f). For boundaries, nine distinct clusters were enriched for different regulatory features, including active promoter marks such as H3K4me3, pioneer factors like Zld and GAF and classic insulator proteins such as CTCF, CP190 and BEAF-32 (Fig. 2a,b and Extended Data Fig. 3a,b). Among these, polymerase II (Pol II) and TATA-binding protein were more evenly distributed across clusters, suggesting a broader association with architecture (Extended Data Fig. 3b). Moreover, clusters enriched for insulators (clusters 5 and 7) displayed the highest boundary strength, highlighting distinct contributions of individual insulators to chromatin architecture (Extended Data Fig. 3c). Loop anchors similarly separated into seven distinct clusters, each marked by unique regulatory profiles and positional heterogeneity (Fig. 2c,d and Extended Data Fig. 3d–f). These clusters varied in loop span, with certain regulatory contexts—such as Su(Hw)-enriched anchors (cluster 5)—more frequently associated with long-range interactions compared to clusters characterized by pioneer factors or BEAF-32 (Extended Data Fig. 3e,f). Together, these analyses reveal that both boundaries and loops are defined by diverse regulatory environments, with distinct factor enrichments shaping the architecture of the early genome.

To contextualize the broader chromatin state across early development, we applied ChromHMM to integrate diverse epigenomic data into 20 chromatin states, aiming to map broad chromatin features across early development despite some dataset gaps. Each state represents a combinatorial likelihood of specific chromatin marks co-occurring, providing a comprehensive view of the chromatin's functional landscape across different stages and conditions. These

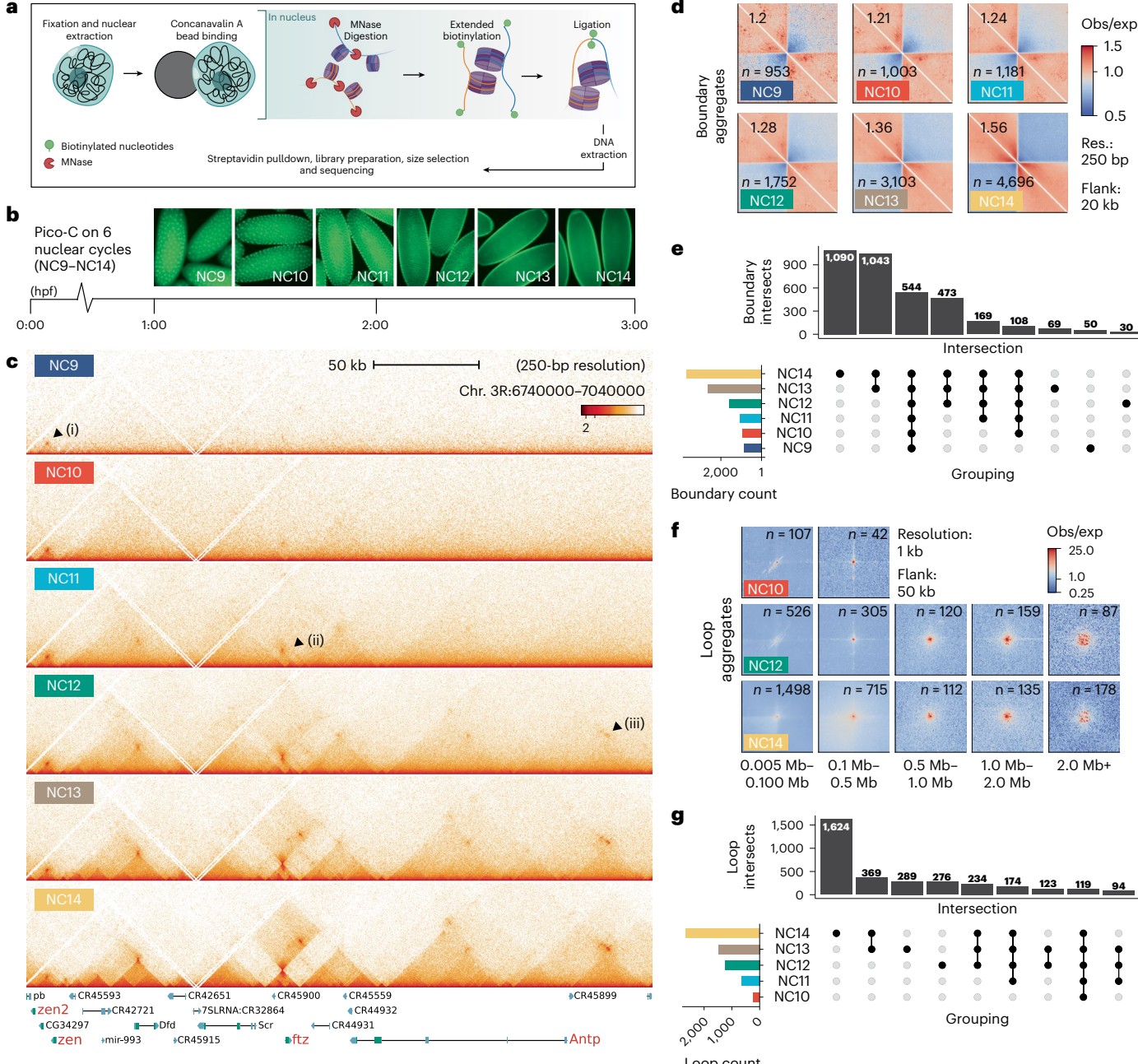

**Fig. 1 | Pico-C enables sub-kilobase chromatin maps across six developmental stages during the establishment of genome architecture in *Drosophila*.**
**a**, Schematic of the low-input Micro-C method, Pico-C, developed to capture interphase-staged dynamics in early NCs. Extracted formaldehyde- and EGS (ethylene glycol bis(succinimidyl succinate))-fixed nuclei were immobilized on concanavalin A beads, eliminating most spin-down steps and considerably reducing sample loss. Moreover, the biotinylation step was extended from 45 min to 4 h to improve pulldown efficiency. All subsequent steps, including library preparation and size selection, were done as previously described[57].
**b**, Microscopy images of *Drosophila* embryos at NC9–NC14 used for Micro-C. Embryos were hand sorted by utilizing nuclear density for staging. Nuclear PCNA–GFP was used to delineate interphase embryos as done previously[18,23,58]. **c**, Stepwise establishment of 3D genome organization of the ANT-C. Our matrices capture early looping between the *zen* and *zen2* genes (i) taking place as early

as NC9, the formation of the bowtie structure around the *ftz* gene (ii) and the establishment of tether elements at the *Antp* gene (iii). **d**, Aggregate analysis of boundaries identified using low-input Micro-C data from embryos at NC9–NC14. Boundaries were called at 500-bp resolution (res.) and aggregate plots of observed/expected contacts (Obs/exp scale) were plotted at 250-bp resolution with 20-kb flanking regions. The scores in the upper left corners indicate average insulation score at each developmental stage. **e**, UpSet plot of boundaries gained through development across NCs. Most boundaries are gained progressively through development. **f**, Aggregate analysis of chromatin loops called at 250-bp, 1-kb, 2-kb, 4-kb and 16-kb resolution, plotted across different anchor-to-anchor distances at 1-kb resolution with 50-kb flanking regions. **g**, UpSet plot of loops called in NC9–NC14 (grouping >80). NC9 is omitted due to a low loop count (*n* = 11). Most loops are gained over time, but many are stage specific. Images in **a** and **b** created with BioRender.com.

states were defined across four broader developmental timepoints: pre-minZGA (NC1–NC8), pre-majZGA(a) (NC9–NC11), pre-majZGA(b) (NC12–NC13) and ZGA (NC14) and Zelda⁻ encompassing Zld knockdown datasets from NC12–NC14 (Fig. 2e). States were ordered by temporal

enrichment and annotated based on their emission probabilities, genome-wide distributions and gene–body positioning, resulting in distinct categories including pioneer-associated states, open or nucleosomal states, H4K16ac-enriched genic states, transcriptional

states biased toward transcription start site (TSS) or transcription termination site (TTS), HP1-associated states and promoter-like states (Fig. 2e and Extended Data Fig. 4a,b).

De novo motif enrichment using HOMER[30] supported these classifications, identifying enrichment of Zld and GAF motifs in their respective pioneer states and motifs for M1BP and Dref—transcription factors known to bind promoters—in promoter-like states, along with core *Drosophila* promoter elements (Fig. 2f and Extended Data Fig. 4c). We also observed increased enrichment of two HP1-associated states (states 12 and 13) at ZGA, found predominantly at centromeric and telomeric regions, consistent with previous studies[14,16] (Fig. 2e and Extended Data Fig. 4a). State 13 (HP1[2]) showed elevated GAF emission, supporting emerging evidence that GAF contributes to heterochromatin regulation[19]. Notably, HP1[1] was also enriched for E2F1 motifs—a pro-mitotic factor that regulates position effect variegation in flies and is modulated by HP1α in mammals[31–33]—suggesting that a similar mechanism may operate as the embryo transitions out of rapid mitotic divisions (Fig. 2f and Extended Data Fig. 4a). Gene ontology analysis revealed that Genic[2] (state 5), enriched for H4K16ac, was associated with morphogenesis, consistent with previous findings[34], whereas Promoter-like[1] (state 7) was linked to embryonic processes including regionalization, cell-fate specification and messenger RNA metabolism (Extended Data Fig. 4d). Importantly, these states do not appear to be confounded by coverage bias (Extended Data Fig. 4e). Together, ChromHMM serves as a useful guide for interpreting the chromatin landscape underlying our Pico-C maps.

To investigate how the chromatin state correlates with genome organization, we applied CALDER2[35] to the NC14 Pico-C matrix, which delineated subcompartments A.1, A.2, B.1 and B.2 at 3-kb resolution. We then tracked how these subcompartments interacted across earlier stages and found that A.1–A.1 and B.2–B.2 compartments formed the most prominent interactions overall (Extended Data Fig. 5a). Notably, A.1–A.1 contacts already displayed interactions by NC12 (Fig. 2g and Extended Data Fig. 5a). ChromHMM analysis showed that A.1 subcompartments were enriched for H4K16ac-marked genic states during pre-majZGA(a), a trend also supported by H4K16ac ChIP–seq signal (Extended Data Fig. 5b). This observation is consistent with previous work showing that A compartments are preferentially associated with intergenerationally maintained H4K16ac[34]. These findings suggest that A.1 and B.2 compartments follow distinct temporal and regulatory paths during genome establishment. This interpretation is consistent with findings that HP1α depletion primarily affects B compartments while leaving A compartments largely intact[16]. Consistent with earlier observations, HP1α did not show clear compartment-specific enrichment, in agreement with prior observations that HP1α is found in both A and B compartments[16] (Extended Data Fig. 5c).

Integrating our clustering analysis, chromatin state annotations and compartment calls with our Pico-C data, we can capture a multilayered view of the genome. For example, the *even-skipped* (*eve*) locus, flanked by the well-characterized insulator sequences Nhomie (upstream) and Homie (downstream)—both bound by Su(Hw)[36–39]— are consistently classified as cluster 8 boundaries in our analysis (Fig. 2i). This region also displays active transcriptional ChromHMM states and falls within an A.1 subcompartment, highlighting convergence across orthogonal datasets (Fig. 2i). This layered perspective demonstrates a coherent regulatory landscape underpinning chromatin architecture.

### Context-dependent coupling of chromatin structure and state

Having created precise Pico-C maps and defined the underlying chromatin states, we wanted to investigate the relationship between them. To do this, we first utilized Metaloci[40] which uses spatial autocorrelation (Moran's *I*) to examine how strongly a genomic signal correlates with itself across neighboring regions in nuclear space. Higher global Moran's *I* values indicate that spatially proximal regions exhibit similar signal intensities, whereas values near zero reflect spatial randomness. Negative values suggest anti-correlation, where nearby regions display opposing signal levels. This metric enables us to assess how specific chromatin features are spatially organized and how these relationships evolve across developmental timepoints. Through a sliding-window approach we found that, genome wide, there is a general trend of increased Pol II, RNA-seq, H3K36me3, H4K16ac, H3K27me3, GAF and Clamp spatial clustering, whereas early acting Zld showed no significant change (Extended Data Fig. 6a).

Despite the genome-wide increase in Pol II clustering during ZGA, we observed local variability in spatial correlations. Most notably, at the *HDAC1* locus, the global Moran's *I* for Pol II decreased from 0.68 at NC12 to 0.38 at NC14, coinciding with loop loss (Fig. 3a,b and Extended Data Fig. 6b). Overlaying our ChromHMM states showed that these loop anchors were initially enriched for promoter-like and pioneer-associated chromatin states, but, by NC14, these regions transitioned to transcriptional states as loops resolved (Fig. 3c,d and Extended Data Fig. 6c). Rather than reflecting inactivity, this shift suggests that changes in transcriptional activity may disrupt early loops. This pattern contrasts with boundaries gained at NC14, where transcriptional states also increase but emerge from promoter-like state peaks (Extended Data Fig. 6d).

To investigate the causal role of transcription in genome architecture, we inhibited transcriptional elongation by injecting α-amanitin before NC8 and minZGA, as previously described[18], and performed Pico-C profiling to capture architectural changes at higher granularity (Fig. 3e). This treatment resulted in a global loss of zygotic transcripts accompanied by a weaker upregulation of maternal RNAs, likely reflecting impaired maternal RNA clearance (Fig. 3f and Extended Data Fig. 7a,b). Distinctly, we observed substantial downregulation of zygotically transcribed lncRNA:CR44504, a predicted precursor of the miR-309 cluster required for the degradation of many transcripts undergoing decay at ZGA[41] (Extended Data Fig. 7c).

**Fig. 2 | Diverse chromatin signatures anchor architectural features in the early embryo. a**, NC14 boundary Uniform Manifold Approximation and Projection (UMAP) grouped into nine clusters based on factor occupancy. Example profiles for clusters 5–7 show distinct enrichments for CTCF, CP190, pioneer factors and H3K4me3 (shown tracks[9,11,22,42]). Information on all datasets used is provided in Supplementary Data 3. **b**, NC14 Pico-C contact map with boundary cluster annotations. **c**, NC14 loop–anchor UMAP grouped into seven clusters based on factor occupancy. Representative profiles for clusters 1, 2 and 4 show similar distinct enrichment for factors such as CTCF, CP190, GAF, Zelda and H3K4me3[9,11,22,42]. Information on all datasets used is provided in Supplementary Data 3. **d**, NC14 Pico-C contact map with loop–anchor cluster annotations. **e**, Twenty-two genomic signals were compiled across four developmental stages and Zelda knockdown (Zelda⁻) background to define 20 ChromHMM states. Each row corresponds to a state and its assigned annotation. Annotation colors denote major categories: light cyan (quiescent), blue (ATAC associated), purple

(pioneer factor enriched), orange (genic), yellow (transcribing near TTS), green (transcribing near TSS), red (promoter like) and gray (HP1 associated). The left heatmap shows emission probabilities for each signal and the right one shows normalized state enrichment across stages and genetic backgrounds. **f**, Example of highly enriched de novo motifs (HOMER) from selected states. **g**, Aggregate analysis for A.1 and B.2 subcompartments at stages NC12 and NC14. A.1 regions display interactions as early as NC12. Aggregates were generated with one compartment-length flanks (1x comp.) and visualized as observed/expected (Obs/exp). **h**, ChromHMM state enrichment profiles across compartments at three developmental stages: pre-majZGA(a) (left), pre-majZGA(b) (middle) and ZGA (right). A.1 compartments are enriched for genic and active chromatin states, whereas B.2 compartments show general enrichment for quiescent states. **i**, NC14 Pico-C contact map at the *eve* (*even-skipped*) locus with annotated boundary clusters, ChromHMM chromatin states and compartment calls.

It is interesting that, on abrogation of transcriptional elongation, we found that a subset of early chromatin loops persisted and closer inspection revealed that the anchors of these retained loops overlapped with early activated genes, previously characterized by low levels of paused Pol II[42] (Fig. 3g and Extended Data Fig. 7d).

Consistently, Pol II ChIP from our previous work[18] on α-amanitin-treated embryos showed that these regions lose Pol II at the gene body but retain Pol II near the promoter (Extended Data Fig. 7d). Moreover, these changes are in agreement with those observed previously by Hi-C[18] in both α-amanitin-injected and triptolide-injected embryos

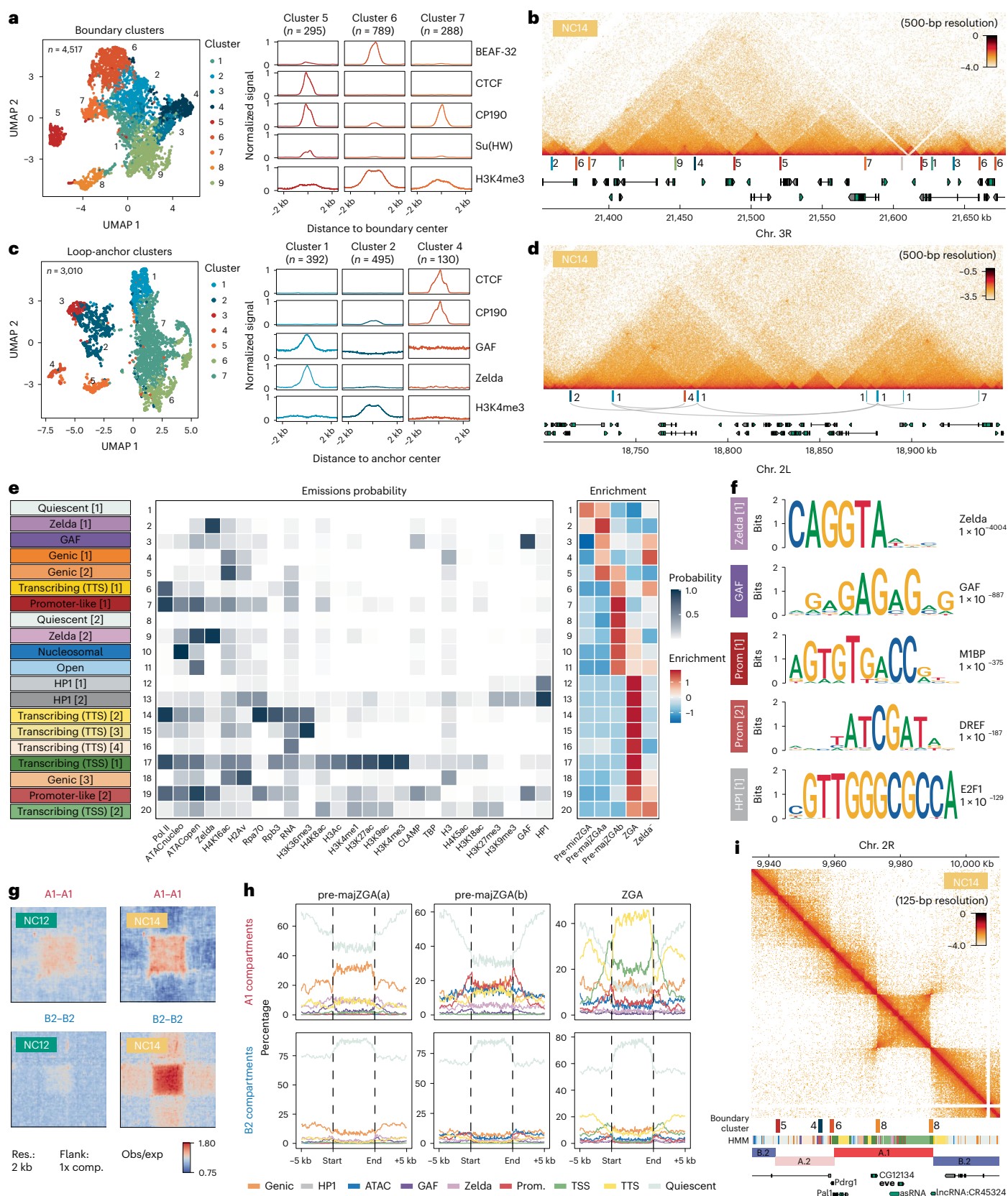

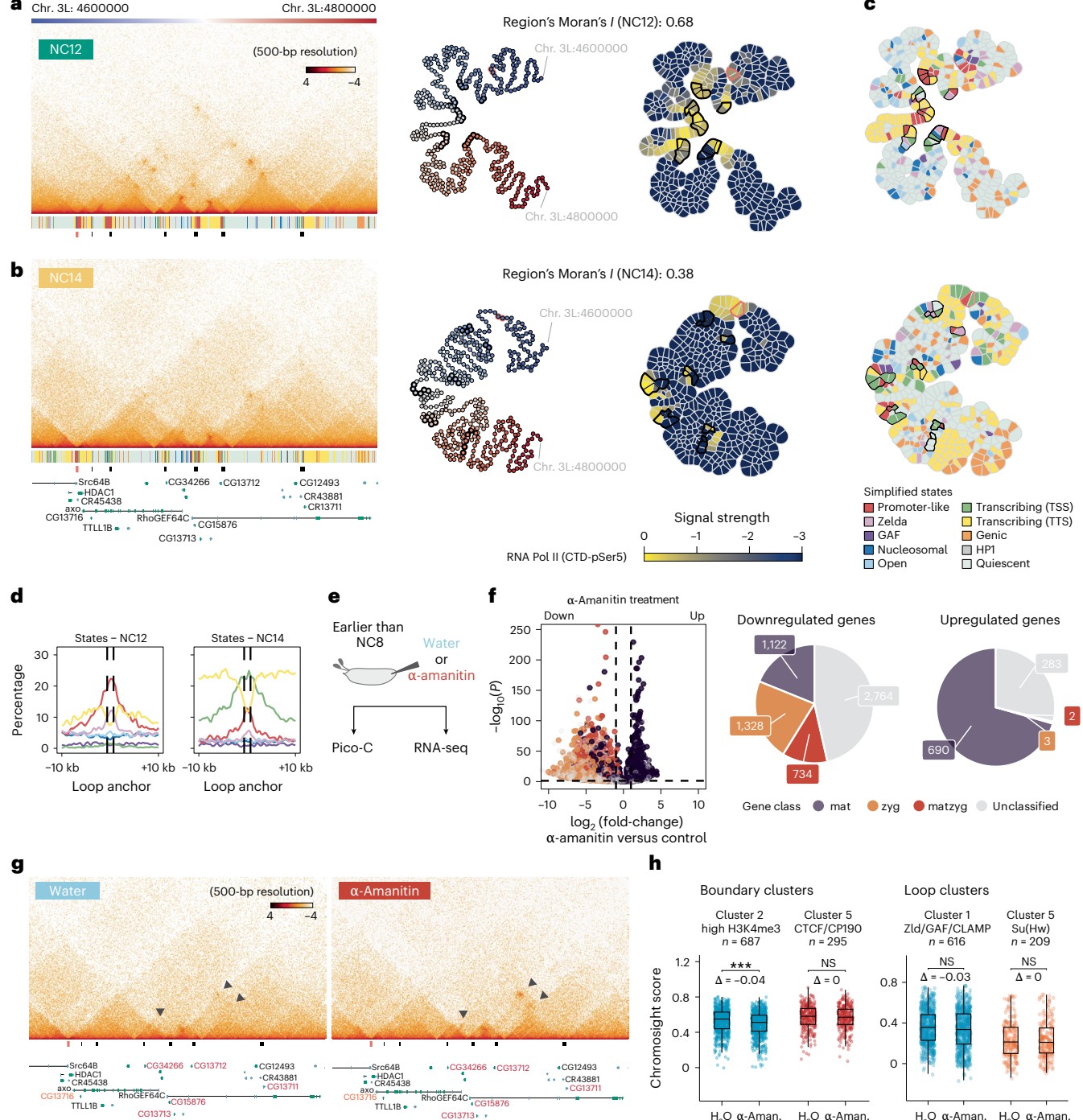

**Fig. 3 | Fine-tuned Pol II regulation alters chromatin architecture in a cluster-specific manner. a,b**, Pico-C maps of the *HDAC1* locus at NC12 (**a**) and NC14 (500-bp resolution) (**b**), with corresponding 2D bead-on-a-string models (the start is demarcated in blue and the end in red) and Gaudi plots overlaid with stage-matched Pol II ChIP signal[59]. Loop anchors (black) and the *HDAC1* promoter boundary (pink) are highlighted. Global Moran's *I* is displayed for the region, indicating spatial autocorrelation across the locus. A higher score means higher clustering of Pol II signal relative to chromatin structure. **c**, Gaudi plots from **a** and **b** overlaid with simplified ChromHMM states. **d**, Genome-wide simplified ChromHMM profiles of loop anchors present at NC12 but lost by NC14. Quiescent, genic and HP1 states (which showed no marked changes) were removed from the plot for clarity. **e**, Schematic of transcriptional inhibition by α-amanitin. Embryos at or older than NC8 were discarded immediately after injection. The remaining embryos were allowed to develop to the NC14 stage, then either fixed for Pico-C or collected for RNA extraction. **f**, RNA-seq showing strong depletion of zygotic transcripts and modest retention or upregulation

of maternal RNAs—likely due to impaired zygotic-driven clearance of maternal RNAs. **g**, Pico-C maps of water-injected (controls) or α-amanitin-injected embryos showing, on transcriptional inhibition, retention of some early loops. The genes marked in red are characterized as early expressed genes[42] that anchor retained loops, whereas the gene highlighted in orange (*CG13716*) represents an early gene where the associated loop is lost despite α-amanitin treatment. **h**, Chromosight analysis showing cluster-specific decreased strength of architectural features at boundaries (left) and loops (right). The scores are grouped by cluster identity. The annotated values show the mean score difference between α-amanitin-injected and water-injected embryos. Statistical comparisons performed using unpaired Wilcoxon's rank-sum tests were performed per cluster and *P* values were adjusted using Benjamini–Hochberg false discovery rate methods. The significance is indicated as follows for adjusted *P* values: $^{***}P_{adj} < 0.001$, $^{**}P_{adj} < 0.01$ and $^{*}P_{adj} < 0.05$. α-Aman., α-Amanitin; NS, not significant.

(Extended Data Fig. 7e). In addition, even within individual loci, not all early loops were retained after transcriptional abrogation (Fig. 3g), making it difficult to pinpoint whether loop persistence reflects Pol II pausing[43], delayed eviction or a downstream effect. The heterogeneous response suggests that early loop stability is similarly governed by multiple, distinct regulatory mechanisms.

Nevertheless, these maps and their corresponding controls enabled us to evaluate the architectural consequences of abrogated transcription elongation inhibition. For this, we employed Chromosight, a computer vision-based tool that quantitatively scores loop and boundary strength[44]. This analysis revealed a modest, but often cluster-specific, reduction in both loop and boundary signals after transcriptional inhibition (Extended Data Fig. 7f–i). For boundaries, the most prominent reductions were observed in clusters enriched for the active promoter mark H3K4me3—specifically, cluster 2 and the BEAF-32-associated cluster 6, whereas cluster 5 enriched for insulators CTCF and CP190 showed no decrease in insulation (Fig. 3h and Extended Data Fig. 7f). Moreover, these changes appeared largely locus specific rather than representing a uniform trend across regions within a given cluster (Extended Data Fig. 7g). For loops, no loop–anchor cluster passed the significance threshold ($P > 0.05$, unpaired Wilcoxon's rank-sum test, Benjamini–Hochberg adjusted; Extended Data Fig. 7h). Like boundaries, there was no clear trend across loop–anchor regions. However, cluster 1—enriched for Zld, GAF and CLAMP and exhibiting the highest average RNA output—showed the greatest, albeit still weak, decrease in loop strength, whereas loop–anchor cluster 5, associated with Su(Hw), remained unaffected (Fig. 3h and Extended Data Fig. 7h,i). It is interesting that CBP depletion, which similarly disrupts transcriptional elongation, also leads to modest chromatin changes, supporting that much of the chromatin landscape in *Drosophila* is shaped by additional regulatory mechanisms[45].

Together, these results suggest that early genome organization is coordinated by distinct regulatory inputs.

## Sequence-based models predict motif diversity in 3D genome

To systematically identify the underlying sequence features associated with the dynamic changes and their potential regulators in the early embryo, we employed machine learning to perform in silico perturbations of the system. Using the Orca algorithm[46], which integrates sequence information to predict chromatin interactions, we trained two models for NC12 and NC14 at a 1-kb resolution. The NC14 model had a high correlation with experimental data (Pearson's correlation, $r = 0.68$) compared to the NC12 model ($r = 0.42$), likely reflecting the more defined chromatin structure present at NC14 (Supplementary Table 2). Highlighting the specificity of the approach, both models predict architecture accurately (Fig. 4a and Extended Data Fig. 8a). Moreover, when we mutated Zld motifs in silico at the *bitesize (btsz)* locus, a site known to be affected by Zld depletion[18], the models predicted a local weakening of boundary insulation at the correct position in both NC12 and NC14 (Extended Data Fig. 8a,b). Although the effect is smaller than in the experimental data, it is notable that the model captures the expected direction and location of the change based solely on DNA sequence.

We then utilized a multiplexed in silico mutagenesis approach[46] to screen for sequence disruptions that could induce local structural remodeling within a 0.25-Mb distance in both models. We calculated a structural impact score for each region of the genome by introducing three random mutations every 10 bp across the genome. This score indicates the predicted impact of a sequence mutation on local architecture, with higher values reflecting greater structural disruption. Regions with high structural impact scores were typically associated with transcriptionally active, promoter-like states (Fig. 4a). To systematically examine the association of these scores with the chromatin landscape, we plotted the structural impact score distribution across all 20 states (Fig. 4b). We observed that regions with promoter-like states (states 7 and 19) had significantly higher impact scores (Fig. 4b

and Extended Data Fig. 8c; $P < 2 \times 10^{-16}$, analysis of variance (ANOVA)). To test whether promoter-like regions outside TSSs also carry high structural impact, we split states 7 and 19 into TSS-overlapping and TSS-nonoverlapping subsets, finding that, although non-TSS regions had lower scores, in state 19 they still ranked significantly higher than other states (Extended Data Fig. 8d). This suggests that promoter-like chromatin states, particularly state 19, may have the capacity at the sequence level to act similarly to promoters, even in regions distant from annotated TSSs, potentially marking putative enhancers.

Motif enrichment analysis at regions with high structural impact scores revealed several known factors, including M1BP, Dref, BEAF-32, Clamp, GAF and Zld, as well as factors such as Spps, sqz, Hr3, L(3)neo38, Lmd and Pnr, which were previously uncharacterized as chromatin structural regulators (Fig. 4c and Extended Data Fig. 8e,f). De novo motif analysis revealed similar enriched motifs as before, along with additional elements such as C-repeats and canonical promoter sequences (Fig. 4d). To determine whether these factors are present at this developmental stage, we used RNA-seq data for NC7–NC14 that utilized 5-ethynyl uridine to distinguish between maternal and zygotic transcripts[7]. This showed that a number of these candidate factors are zygotically present (Extended Data Fig. 8e). Moreover, these factors shared similar motifs, including ATCGA-like motifs (BEAF-32, Dref and Pnr), GGTCAC-like motifs (M1BP and Hr3) and C-rich motifs (Spps, L(3)neo38, Lmd and Sug) (Extended Data Fig. 8e). In addition, Sqz, which has an A-rich motif and is predicted to have RNA Pol II-specific activity (FlyBase ID FBgn0010768), was present both maternally and zygotically (Extended Data Fig. 8e).

To test the contribution that each factor has to genome architecture, we performed in silico-directed mutagenesis for each factor's motifs. M1BP showed the strongest individual effect, whereas Sqz, a putative regulator, had a greater impact than GAF or Zld (Extended Data Fig. 8g). Although single-factor depletion did not disrupt chromatin architecture substantially, consistent with previous findings for specific factors[18–20], simultaneous in silico mutagenesis of multiple factor motifs led to a pronounced reduction in insulation (Fig. 4e). To assess whether these effects were built on each other, we compared insulation changes from combined versus individual mutations and found a strong correlation (Fig. 4f). In addition, pairwise comparisons of insulation changes from single-factor mutagenesis showed largely uncorrelated effects (Extended Data Fig. 8h). These results suggest that genome architecture in *Drosophila* is additively encoded by multiple factors, with each factor contributing uniquely to chromatin architecture.

## Genome establishment is coordinated by distinct orthogonal inputs

To investigate how the chromatin state relates to architectural features during genome establishment, we first examined the distribution of chromatin states at loop anchors and boundaries across developmental time and in Zld-depleted embryos. This analysis revealed that promoter-like states remain enriched at loop anchors and boundaries relative to their local genomic context, even in Zld knockdown (Zelda−) ChromHMM states (Extended Data Fig. 9a). To explore this further, we plotted chromatin state dynamics across regulatory clusters and found that many clusters—despite differing in their factor composition—converge on a similar active chromatin landscape (Extended Data Fig. 9b,c). This suggests that distinct regulatory configurations can produce similar chromatin outputs, potentially explaining the limited impact on promoter-like state enrichment despite perturbations. Such resilience may reflect orthogonal pathways contributing to genome architecture.

To directly test whether genome establishment involves orthogonal inputs, we used the Jabba-Trap system to simultaneously deplete Zld and GAF and performed both ATAC−seq and Pico-C on embryos at NC14 (Fig. 5a and Extended Data Fig. 10a). Jabba-Trap inactivates GFP-tagged proteins by sequestering them to lipid droplets[47], effectively removing the tagged proteins from their nuclear targets (Fig. 5a

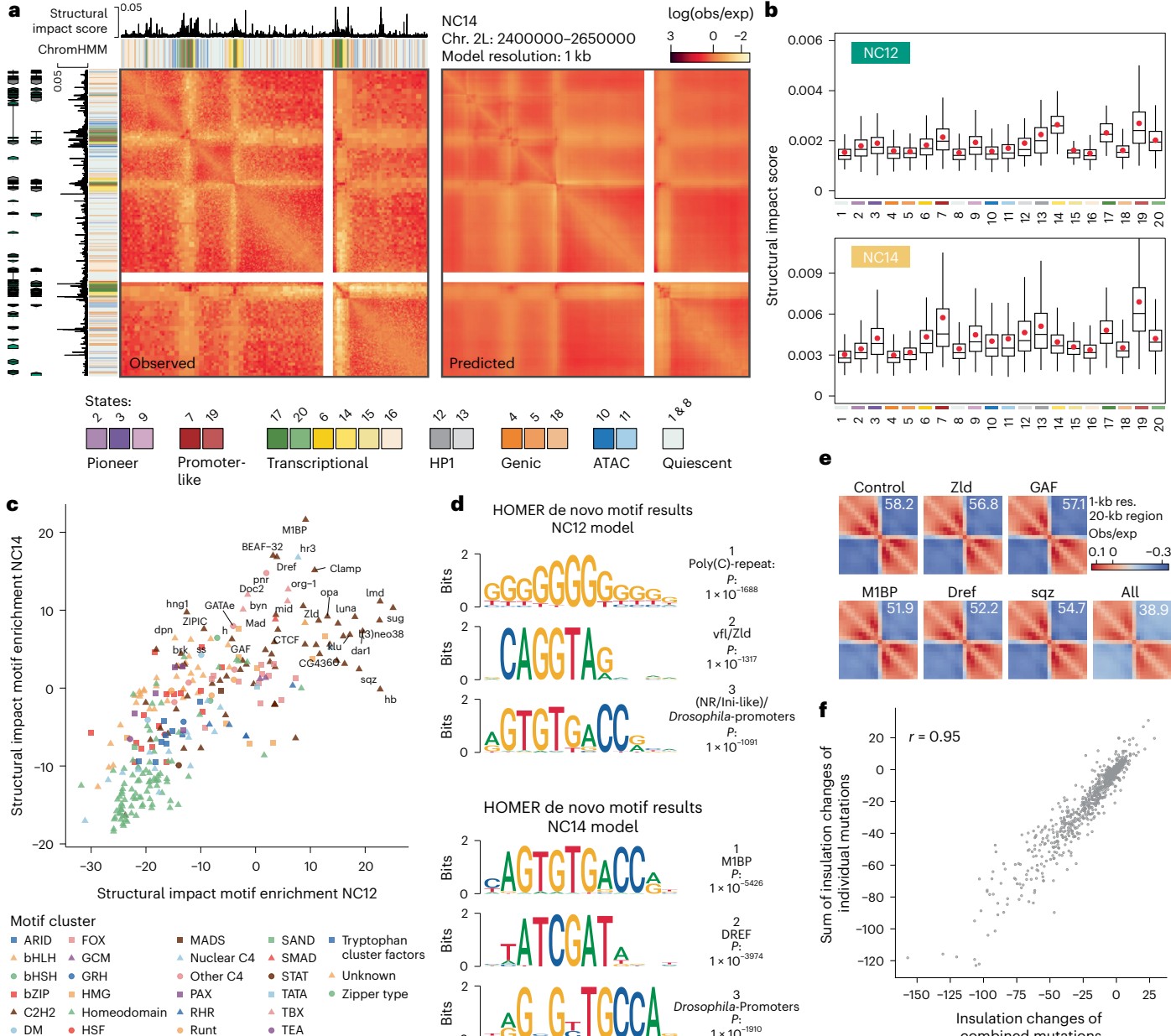

**Fig. 4 | Modeling of chromatin architecture using Orca on Pico-C maps highlights promoter-linked state regions and reveals a diverse set of sequences associated with predicted architecture. a**, Example region from the validation set (not used in training) showing high concordance between the model's prediction and observed data. Left: ChromHMM states and the structural impact score, calculated by mutating random sequences genome wide. Higher structural impact scores indicate regions where mutations are predicted to affect chromatin structure more strongly. **b**, Comparison of structural impact score distributions across matched ChromHMM state regions between NC12 and NC14 models. For each chromatin state, the median is shown as the central line within the box, whereas the red points indicate the mean. **c**, Motif enrichment analysis of regions with high structural impact scores (≥0.02 for NC14, ≥0.01 for NC12),

identifying motifs associated with sequences predicted to be highly sensitive to in silico mutagenesis. **d**, HOMER de novo motif enrichment in regions with high structural impact scores (>0.01 for NC12, >0.02 for NC14). NC12-enriched motifs include poly(C) repeats, Zld and promoter-linked sequences, whereas NC14 is dominated by M1BP, DREF and other promoter-associated elements. **e**, Aggregate boundary analysis on the validation chromosome comparing the control model with in silico-directed mutagenesis of the factors identified in **b**. The 'all' condition represents simultaneous mutation of all listed factors. Boundary strength is quantified using the insulation score (top right of each aggregate), calculated from the log(observed/expected) signal within a 20-kb window. **f**, Comparison of insulation changes from combined mutations versus the sum of individual mutations showing a strong correlation ($r = 0.95$).

and Extended Data Fig. 10a). To assess the regulatory consequences of combined depletion, we performed de novo motif enrichment at regions with reduced ATAC–seq accessibility in double-knockdown (DKD) embryos. This analysis revealed Zld and GAF motifs as the most significantly enriched (Fig. 5b), supporting depletion of both factors. Notably, highly resolved Pico-C maps on these mutants revealed that architectural disruptions at both Zld-associated[18,48] and GAF-associated features[20]

co-occur in the DKD (Fig. 5c and Extended Data Fig. 10b). To further query whether these disruptions are visible in a cluster-specific manner, we conducted Chromosight analysis comparing these matrices with their controls (Fig. 5d and Extended Data Fig. 10c–f). We found that boundary clusters 2 (Zld-low), 3 (Zld-medium) and 4 (Zld-high and GAF-high) exhibited the most pronounced weakening on removal of both factors (Fig. 5d and Extended Data Fig. 10c,d). A similar pattern was observed for

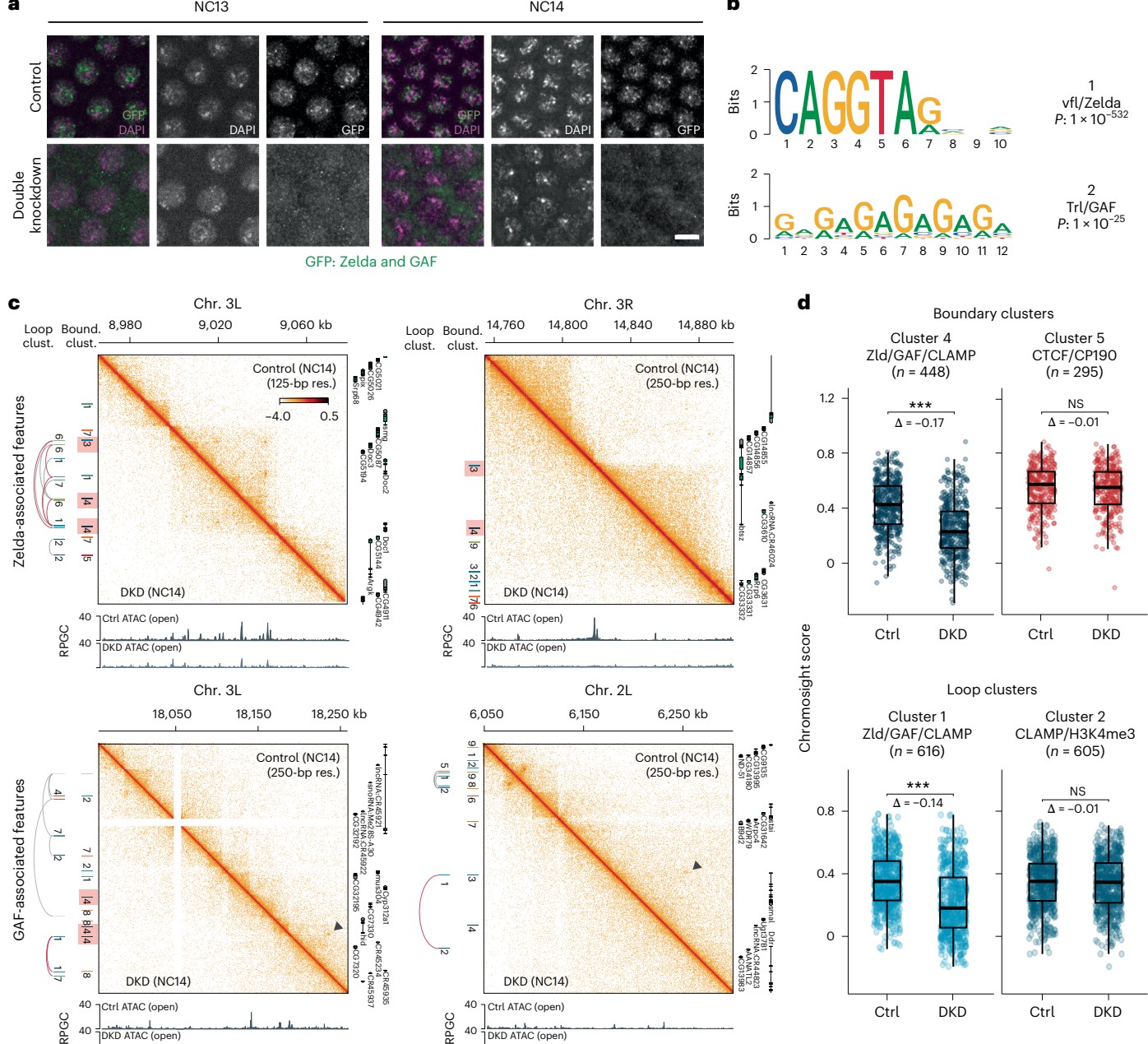

**Fig. 5 | Zelda and GAF regulate genome organization through orthogonal mechanisms. a**, Jabba-Trap system simultaneously depleting Zelda and GAF in embryos. GFP, which marks both GAF and Zelda, is excluded from the nuclei. Scale bar: 5 μm. **b**, HOMER de novo motif enrichment analysis at lost open chromatin regions in DKDs showing enrichment of the predicted Zld motif and GAF motif. **c**, Examples of NC14 loci with altered chromatin architecture in DKD embryos. The loops are shown as arcs, with anchors colored and numbered by loop–cluster identity (Loop clust.). The red arcs denote loops that are disrupted. The boundaries altered in DKD embryos are marked by red boxes next to the heatmaps and are similarly labeled by boundary cluster identity (Bound. clust.). Most disrupted features correspond to Zld-enriched or GAF-enriched clusters. Matched ATAC−seq tracks show locus-specific loss of open chromatin at these sites. Previously defined GAF-dependent loops[20] are indicated with black arrowheads. **d**, Chromosight analysis showing cluster-specific reductions in architectural strength at boundaries and loops after Zld or GAF depletion. Boundary cluster 4 (Zld or GAF enriched) displays a strong loss of insulation, whereas cluster 5 (CTCF or CP190 enriched) is largely unaffected. A similar pattern is observed at loop anchors: cluster 1 (Zld or GAF associated) shows pronounced weakening, whereas cluster 2 (CLAMP and H3K4me3 enriched) is only modestly impacted. Statistical comparisons were performed using an unpaired Wilcoxon's rank-sum test with Benjamini−Hochberg correction. The $P_{adj}$ thresholds are: ***$P_{adj} < 0.001$, **$P_{adj} < 0.01$, *$P_{adj} < 0.05$. Ctrl, control; DKD, double knockdown.

loops, where the greatest effect was seen in the Zld−GAF cluster 1 (Fig. 5d and Extended Data Fig. 10e,f). In contrast, other clusters associated with different factors showed weak to no reductions, many of which were not statistically significant ($P > 0.05$, unpaired Wilcoxon's rank-sum test, Benjamini−Hochberg adjusted; Fig. 5d and Extended Data Fig. 10e,f). These results support a model in which early genome architecture is shaped by distinct regulatory pathways. Importantly, this does not exclude the presence of regions that are co-regulated by both factors. In our boundary clustering analysis, for example, clusters 2 and 3 were predominantly enriched for Zld, whereas cluster 4−marked by both Zld and GAF−showed the most pronounced loss of insulation in Chromosight analysis (Extended Data Figs. 3b and 10e,f). Similarly, subclustering of loop anchors identified a subgroup (subcluster 1.3) with high occupancy of Zld, GAF and Clamp (Extended Data Fig. 3f).

Together, these findings suggest that, although genome architecture is shaped by orthogonal inputs, these pathways are not mutually exclusive and may act cooperatively at specific loci.

## Discussion

This work introduces a versatile framework for exploring the regulatory logic of genome architecture during early embryogenesis. First, we generated a high-resolution dataset that captured the dynamic establishment of 3D chromatin structure in *Drosophila* embryos. This resource offered a sharp view of genome folding as it emerges, enabling precise identification of architectural features across developmental time, extending recent Micro-C observations that such structures can arise before genome activation[49]. Building on this, we integrated publicly available datasets to characterize distinct inputs underpinning conformation, revealing that these can be stratified by differential enrichment of transcription factors and insulator proteins. Furthermore, by modeling chromatin architecture in relation to sequence, we identified a diverse set of motifs associated with structural elements. Notably, many of these motifs corresponded to transcription factors with little to no prior characterization in the context of early genome activation, providing a rich set of candidates for future functional interrogation. Although these results offer a layered view of genome organization, this work will continue to benefit from new datasets. For example, we still observe regions that lack clear factor enrichment, suggesting that additional regulators or regulatory modes remain to be discovered.

Overall, our results showed that *Drosophila* genome establishment relies on multiple distinct regulatory pathways. This differs from vertebrate systems, where pioneer factors often function redundantly. In *Drosophila*, both Zld and GAF are essential for embryonic viability. By comparison, in zebrafish, Nanog, Pou5f3 and Sox19b (NPS) act cooperatively to activate the zygotic genome[50,51]. In the mouse, Dux contributes to ZGA but is dispensable for development[52,53]. The recent identification of Obox4 as a co-regulator of Dux[54] supports the idea that pioneer factor activity in vertebrates is often shared among partially redundant proteins. In contrast, our work suggests that genome regulators in *Drosophila* function additively.

These findings raise other broader questions about how early chromatin architecture is programmed. Our results suggest that proper genome establishment involves both factor-specific mechanisms and broader context-dependent constraints. For example, loop anchors in our analysis can belong to multiple regulatory clusters despite looping together, implying that this is not governed solely by the presence of specific factors, but also potentially by the regulatory history and neighborhood context of a locus. In this light, it is notable that depletion of Zld and GAF produces stronger architectural effects than blocking transcription with α-amanitin. One possible explanation is that pioneer factors act upstream in establishing chromatin conformation, whereas transcription represents a more downstream process[55]. It will be interesting in the future to target the pre-initiation complex with more refined genetic tools to continue examining the interplay of chromatin conformation and landscape[18,43,56,57]. These genetic perturbations combined with temporal data will be particularly powerful in dissecting the role of hysteresis in genome architecture, further elucidating the interplay of factor specificity, order and timing of regulatory events and conformation. Together, our work lays the foundation for dissecting how diverse regulatory inputs, both orthogonal and cooperative, drive robust chromatin architecture in early development.

## Online content

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

## Methods

### *Drosophila* stock maintenance

PCNA::EGFP flies used for interphase-staged Pico-C were generously provided by S. A. Blythe and E. Wieschaus (Princeton University). The *sfGFP-Zelda* and *GAF-sfGFP* flies were generated as described previously[11,60]. Jabba-Trap was expressed from the maternal α-tubulin 67 promoter, kindly provided by S. A. Blythe[61]. Flies were maintained on standard cornmeal-agar food and raised at 25 °C.

### Embryo fixation and collection

A full step-by-step protocol covering embryo fixation can be accessed at protocols.io (https://doi.org/10.17504/protocols.io.rm7vz9ro5gx1/v1). Briefly, PCNA::EGFP flies were allowed to lay eggs on 0.4% acetic acid agar plates at 25 °C for 1 h pre-collection (NC12–NC14) or 2 h pre-collection (NC9–NC11). After egg laying, plates were incubated for 30 min (NC9–NC11) or 1–1.5 h (NC12–NC14) before fixation. Embryo fixation was performed following the published Micro-C protocol[57], with the exception that EGS was used as the sole secondary crosslinker. Embryos were first crosslinked in 1% formaldehyde, quenched with Tris-HCl (final 0.75 M), washed in phosphate-buffered saline with Triton X-100 (0.5% final concentration) and then crosslinked in 3 mM EGS for 45 min at room temperature. After a second quench, embryos were washed, staged and sorted. Interphase embryos showed clear nuclear PCNA signals and NC was assigned based on nuclear density. Sorted embryos were snap-frozen in liquid nitrogen and stored at −80 °C.

### Pico-C

The complete step-by-step protocol, including the micrococcal nuclease digestion test, is available at protocols.io (https://doi.org/10.17504/protocols.io.kqdg31nm1l25/v1). Pico-C libraries were generated following the published Micro-C protocol[57,62] with several key modifications. Nuclei were immobilized on concanavalin A-coated magnetic beads to increase DNA recovery and eliminate centrifugation steps before DNA extraction. Approximately 10 embryos (~60,000 nuclei) were used for the NC14 libraries, with input doubled at each earlier stage (up to ~300 embryos for NC9). After nuclear extraction, samples were resuspended in 900 µl of MB1 and 100 µl of 10× binding buffer (200 mM Hepes-KOH, pH 8, 100 mM KCl, 10 mM CaCl₂, 10 mM MnCl₂ and 5 mM spermidine) along with 10 µl of pre-washed concanavalin A-coated beads. The mixture was rotated for 10 min at room temperature to allow nuclei to bind the beads.

Bound nuclei were digested with micrococcal nuclease at 37 °C for 10 min with shaking (950 rpm). The reaction was stopped by adding EGTA ((ethylenebis(oxonitrilo))tetra-acetate, final 20 mM concentration) and samples were washed 3× with cold MB2 buffer (50 mM NaCl, 10 mM Tris-HCl, pH 7.5, and 10 mM MgCl₂) while kept on the magnet. End-repair and phosphorylation were performed as previously described[62] using T4 PNK and Klenow fragment.

Biotinylation was extended to 4 h to improve ligation efficiency and reduce downstream PCR requirements. At the 2-h mark, 1.5 µl of 100 mM ATP and 50 U of Klenow fragment were added and incubation continued for the remaining 2 h at 25 °C. The reaction was stopped with 0.5 M EDTA and samples were washed twice with 500 µl of cold MB3 buffer (50 mM Tris-HCl, pH 7.5, and 10 mM MgCl₂).

Proximity ligation was carried out using T4 DNA ligase (10,000 U total, added in two steps) for 5 h at room temperature. Unligated ends were treated with exonuclease III to remove free biotin-dNTPs. Crosslinks were reversed by overnight incubation at 65 °C in the presence of proteinase K and sodium dodecylsulfate (SDS). DNA was purified by extraction with phenol and chloroform. To maximize yield from low-input samples, total DNA was carried forward into the final library construction phase and size selected for dinucleosomal fragments (350–500 bp) after PCR amplification using Ampure XP beads. Final libraries were eluted in 10 mM Tris-HCl, pH 8, and quantified by Qubit before sequencing.

### Embryo injections

Injections were performed as previously described[18]. Briefly, α-amanitin (0.5 mg ml⁻¹ in water) or water alone was injected evenly into a PCNA::EGFP embryo. Embryos older than NC8 after injection were discarded. The remaining embryos were monitored until NC14, then either fixed and staged for Pico-C or pooled (10 embryos per batch) in 30 µl of TRIzol and flash-frozen for RNA extraction. All samples were hand sorted to ensure correct staging. A subset of embryos was allowed to develop, revealing consistent arrest in amanitin-injected embryos, whereas controls developed normally as shown before[18].

### RNA-seq in embryos

For each replicate, 40 embryos were pooled and homogenized with a metal pestle. RNA was extracted by chloroform phase separation and purified from the aqueous layer using the NEB Monarch RNA Cleanup Kit (NEB, cat. no. T2030). All samples showed RNA integrity no. >7 on a Bioanalyzer. Libraries were prepared with the Watchmaker mRNA Enrichment and Library Prep Kits according to the manufacturer's protocol, dual indexed, pooled and sequenced on an Illumina NextSeq 2000 to a depth of ≥80 million paired-end, 60-bp reads per sample.

### Single-embryo ATAC–seq on embryos depleted for Zelda and GAF

Single stage 5 (NC14) embryos of the genotype *sfGFP-Zld;Jabba-Trap/CyO;GAF-sfGFP* and *sfGFP-Zld; Sp/CyO;GAF-sfGFP* controls, identified based on morphology, were harvested and processed for ATAC–seq as described previously[11]. Libraries were prepared from four replicates of both experimental and control embryos and paired-end sequencing was performed on a NovaSeq X Plus.

### Immunohistochemistry of embryos depleted for Zelda and GAF

The *sfGFP-Zelda;Jabba-Trap/CyO;sfGFP-GAF* and *sfGFP-Zelda;Sp/CyO;sfGFP-GAF* control flies were put in cages, allowed to lay for 1.5 h and the embryos were aged for an additional hour at 25 °C. *Drosophila* embryo collection, fixation and staining were performed as previously described[11]. The following antibodies were used: rabbit anti-GFP (Abcam, cat. no. ab290, 1:500); DyLight 488-conjugated goat anti-rabbit (Thermo Fisher Scientific, cat. no. 35552, 1:1,000); and DAPI (Thermo Fisher Scientific, cat. no. D1306, 1:1,000). Images were taken using a Nikon Ti-2e Epifluorescent microscope or Nikon A1R+ confocal and processed by Fiji or ImageJ[63] and Adobe Photoshop software.

### Pico-C data processing

Pico-C reads were mapped to the dm6 genome using BWA-MEM v0.7.17 (ref. 64). Aligned reads were then processed with Pairtools v1.0.3 (ref. 65) using the parse2 function to rescue complex walks and any reads below a mapping quality of 3 were removed. The resulting pairs files were sorted and converted to FAN-C v0.9.18 (ref. 66) pairs format. Duplicates were removed and quality control metrics were extracted.

We then generated FAN-C .hic files. Principal component analysis (PCA) was performed on these .hic files using FAN-C, with a sample size of 100,000 interactions, focusing on distances between 100 kb and 1 Mb to assess variation and clustering across the samples. As biological replicates clustered well across all stages (Extended Data Fig. 1a), the .hic files of replicates were merged. The merged files were binned at different resolutions and normalized using iterative correction. We then used a customized Python script to determine the bin size, where 80% of bins had >1,000 reads, as done previously[67], to estimate the optimal resolution of each stage. Finally, the merged FAN-C .hic files were converted to balanced Cooler .mcool files using the `cooler zoomify` utility from Cooler v0.9.3 (ref. 68), with the options --balance --balance-args '--mad-max 0 --max-iters 1000' to perform iterative matrix balancing.

## Pico-C, Micro-C and Hi-C coverage tracks

Genome-wide coverage tracks from Pico-C data were generated using cooltools v0.5.3 (ref. [69]). Full or downsampled matrices (`--count 100000000` using random sample) were used to compute coverage at multiple resolutions with cooltools coverage, producing raw TSV and BigWig outputs. Raw BigWigs were normalized to counts per million (CPM) using wiggletools and converted to the BigWig format via bedGraphToBigWig.

For quantifying signal across chromatin states, per-bin read counts were extracted from 1-kb cooler matrices, intersected with state annotations and scaled by overlap size to obtain total counts per state. Counts were then normalized by region length and total mapped reads to derive bins per million coverage, analogous to transcript per million normalization in RNA-seq.

## Loop and boundary identification

Boundaries were detected from equally downsampled .hic files using FAN-C (500-bp bins, 16-kb window). Boundaries with insulation >0.45 and outside centromeric or low-mappability regions were retained. Stage-specific comparisons were performed in R using GenomicRanges v1.56 (ref. [70]) and ComplexUpset v1.3.3 (ref. [71]) with a 5-kb tolerance. All boundary calls are provided in Supplementary Data 1.

Loops were identified using Mustache v1.3.2 (ref. [27]) at 250-bp, 1-kb, 2-kb, 4-kb and 16-kb resolutions to capture looping dynamics across distances. Loops with $P > 0.01$ or in low-mappability or low-centromeric regions were removed. Resolution-specific calls were merged iteratively with Bedtools v2.29.2 (ref. [72]) (`pairToPair -slop 1000`) to create a unified loop set. Stage-specific loop overlap (NC9–NC14) was assessed with hicVennDiagram v1.2 (ref. [73]) applying the same overlap thresholds used for boundaries. All loop calls are provided in Supplementary Data 2.

## Compartment analysis

To identify chromatin compartments from Pico-C data, we used CALDER2 v0.7 (ref. [35]).

NC14 intrachromosomal contact matrices were generated with fanc dump at multiple resolutions (1–50 kb) and converted to tab-delimited format for CALDER2. A $\log_2$(input-normalized H3K36me3)[42,74] BigWig track was used to define the correct A or B orientation. A 3-kb resolution was chosen because it yielded the strongest correlation (≥0.4) with H3K36me3. Subcompartments were simplified to A.1, A.2, B.1 and B.2 for downstream analyses, excluding previously characterized centromeric regions[16]. Compartment calls are provided in Supplementary Data 6.

## Aggregate analysis

Aggregate analysis for loops calls, boundaries and compartments was done by first calculating the expected interaction frequencies using our Pico-C matrices with cooltools `expected-cis`. Pileup analysis was then carried out using coolpup.py v1.1.0 (ref. [75]).

For boundary local pileups, the score in the right corner reflects the average insulation—calculated by dividing the signals in opposing corners of the aggregate as described in the documentation. Compartment pileups were rescaled using the parameters `--rescale --rescale_flank 1 --rescale_size 99` to normalize feature size and flanking regions.

## Clustering of boundaries and loop anchors

The signal of various epigenetic marks and transcription was extracted for a 5-kb window around the center of individual boundaries and loop anchors (anchors within 1 kb of each other were merged and treated as a single anchor). An enrichment score for each region was calculated by dividing the average signal of the center (−500 bp to +500 bp) by the average signal from the flanking region (±1 kb to ±2.25 kb). Hierarchical clustering of boundaries and loop anchors was performed on the enrichment scores using Euclidean distance and Ward's method. The number of clusters was selected empirically as the largest number that produced epigenetically distinct groups. For visualization purposes, UMAP dimensionality reduction was applied to the region by enrichment score matrix. Loops >500 kb were excluded from this analysis, because they tend to show higher noise and weaker signal due to distance-dependent decay (16.1% of total loops). Cluster results are provided in Supplementary Data 3.

## ChromHMM input data processing

Public datasets were processed using standardized pipelines specific to each data type.

For ChIP–seq data, adapter trimming and deduplication were performed with fastp v0.23.1, followed by alignment to dm6 using Bowtie2 v2.5.4 (ref. [76]) (`--no-unal`, `--no-mixed` and `--no-discordant`). SAM files were converted and sorted with Sambamba v1.0.1 (ref. [77]) and reads with mapping quality <20 or overlapping black-listed regions were removed. ATAC–seq data were processed identically, with additional use of deepTools v3.5.6[78] `alignmentSieve(--ATACshift)`. Open chromatin reads (<100 bp) and nucleosomal fragments (180–250 bp) were separated. RNA-seq data were aligned using STAR v2.7.10a[79]. The resulting BAM files were indexed using Sambamba.

Quality control checks for all datasets were performed using Samtools v1.18 (ref. [80]) and FastQC v0.12.1 (ref. [81]).

## ChromHMM

ChromHMM v1.24 (ref. [29]) was used for chromatin state analysis. Datasets were grouped into five categories: pre-minZGA (NC1–NC8), pre-majZGA(a) (NC9–NC11), pre-majZGA(b) (NC12–NC13), ZGA (NC14) and Zelda⁻, which included Zld knockdown data from NC12–NC14. For pre-majZGA stages, where data were available across both NC9–NC11 and NC12–NC13, shared datasets were used; otherwise, stage-specific data were included where available. Input controls were specified in the cell-marks file and, for RNA signal, flowthrough maternal RNA-seq data[7] were used to minimize maternal transcript influence.

Deduplicated BAM files were used for binarization and model training (with LearnModel options -b 100 -r 400). Models with varying state numbers were tested and a 20-state model was selected based on log(likelihood). States were annotated with ChIPseeker v1.40 (ref. [82]) using enrichment across gene bodies, chromosomal distribution and ChromHMM emission probabilities, and ordered by developmental enrichment. Gene ontology enrichment was performed with clusterProfiler v4.12 (ref. [83]) using enrichGO (BP ontology, $P_{adj} < 0.05$), with terms simplified and visualized as dot plots highlighting key biological processes. De novo motif enrichment was assessed using HOMER v4.11 (ref. [30]) (findMotifsGenome.pl) on regions showing peak enrichment for each state. ChromHMM tracks are provided in Supplementary Data 5.

## Metaloci analysis

ChIP–seq and RNA-seq data were processed per replicate using deep-Tools bamCompare to generate $\log_2$(ratio BigWigs) when inputs are available. For RNA-seq, flowthrough was used as input to control for maternal effects. Replicates were merged using bigwigAverage. The dm6 genome was divided into 200-kb windows (100-kb step), excluding bins with >5% unmappability and/or centromeric overlap.

Pico-C and genomic signals were integrated using an adapted METAloci framework[40]. Spatial autocorrelation was applied to assess correlations between chromatin interactions and genomic features. Stage comparisons (NC12 versus NC14) used two-sided Wilcoxon's tests ($^*P < 0.05$, $^{**}P < 0.01$, $^{***}P < 0.001$) and violin plots show global Moran's distributions with mean values highlighted.

## RNA-seq analysis (α-amanitin-treated versus water-treated embryos)

Adapters and low-quality bases were trimmed using fastp v0.24.0 (ref. [84]) with `--detect_adapter_for_pe` and reads were aligned to the dm6 genome using STAR v2.7.11b (`--outFilterMismatchNoverLmax 0.04`, `--outFilterMultimapNmax 1` and `--outSAMtype BAM SortedByCoordinate`). Only uniquely mapping reads were retained. Gene-level quantification was performed with featureCounts v2.0.8 (ref. [85]) in paired-end mode (-p).

Differential expression analysis between α-amanitin-treated and water-treated embryos was performed using DESeq2 v1.44.0 (ref. [86]). Genes with ≥10 counts in at least 2 samples were retained.

A set of stable maternal transcripts based on prior annotations[8] was used for normalization (≥1,000 counts in ≥2 samples, $|\log_2(\text{fold-change})| < 1$, log(counts per min) > 1). Variance-stabilized counts were used for visualization and significance was defined as $P_{adj} < 0.05$.

Scale-factor-normalized BigWig tracks were generated with deepTools (bamCoverage, bin size = 1) and averaged across replicates using bigwigAverage.

## Chromosight analysis

Chromosight v1.6.3 (ref. [44]) was used to quantify boundary and loop strengths using `--subsample` to have equally sampled contact maps from α-amanitin-injected, water-injected and Zelda or GAF DKD embryos and matched controls. For boundary analysis, scores were computed at 500-bp resolution using `--pattern borders`. Loop strengths were quantified using NC14 loop BEDPE coordinates across multiple resolutions (1–8 kb) to capture interactions at varying genomic distances. The highest-resolution score was retained per loop. Loop scores were analyzed at the anchor level by assigning each loop anchor independently to predefined merged loop-anchor cluster regions.

Scores were compared between samples and controls in R using unpaired Wilcoxon's rank-sum tests (***$P < 0.001$, **$P < 0.01$, *$P < 0.05$). Scores were grouped by boundary or loop–anchor cluster, with anchors participating in multiple loops linked to more than one score. To track feature score trends, normalized score differences (−1 to 1) were calculated per locus-positive values indicating stronger signals in controls and negative in experimental samples. Outliers were removed using the interquartile range method and mean and median differences were summarized per cluster.

## Model training and evaluation

We adapted the second-stage Orca model[46], composed of a hierarchical sequence encoder and a multilevel cascading decoder, to predict the chromatin structures at 125-bp, 250-bp, 500-bp and 1-kb levels from 250-kb DNA sequences. NC12 and NC14 Pico-C data were used and genomic sequences were retrieved from the dm6 reference genome. All chromosomes except chr2L:0–19 Mb were used for training, with this region reserved for validation.

We adopted the same training strategy as Orca[46]. The training process with stochastic gradient descent took about 1,500,000 steps (250-kb sequence with batch size 8 and learning rate 0.01 with momentum 0.98) on a server with 4 NVIDIA Tesla v100 (32 GB) graphics processing units. Model performance on the holdout region was assessed by concatenating and flattening 1-kb prediction matrices and calculating Pearson's correlation between predicted and observed Pico-C contact maps.

## Multiplexed in silico mutagenesis

We employed the same in silico mutagenesis screening approach utilized in Orca[46] at 1-kb resolution to identify individual motifs critical for genome interaction. The disruption impact on local genome interactions is measured by 250-kb structural impact score, which is the average absolute log(fold-change) of interactions between the disruption position and all other positions in the 250-kb window.

For motif enrichment analysis, *Drosophila* nonredundant motifs were downloaded from the JASPAR 2024 database[87]. Motif matches for each 10-bp site were scanned for after extending by 10-bp flanking sequence on each side, and a maximum log(odds score) over the 30-bp window for each motif was calculated. To quantify the enrichment of motifs, two-sided Student's *t*-test (without assuming equal variance) was performed to compare the motif log(odds scores) of the sites with impact score >0.02 for NC14 and >0.01 for NC12 against the background of 100,000 sites randomly drawn among all 10-bp sites screened. Fold enrichment was also computed on the same sites with a motif log(odds threshold) of 10.

## Motif-specific mutagenesis

To assess how specific motifs influence global genome architecture, we analyzed boundary regions following genome-wide mutagenesis. The validation region was tiled into 250-kb windows (125-kb step) and accessible motif instances were defined by intersecting ATAC–seq peaks with motif matches. Each site was extended to 14 bp and randomly mutated to ensure consistent sequence length.

Boundaries identified in the original Pico-C data were used for pileup analysis (log(fold-change) over background within 20 kb). Boundary strength was quantified by the insulation score, defined as the difference between the sums of the diagonal versus off-diagonal quadrants of the interaction matrix centered on each boundary. The insulation change was computed as the difference in the score before and after mutation.

Five motifs—Zelda, GAF, M1BP, Dref and sqz—were tested individually and in combination. Pairwise effects were assessed by comparing insulation changes at shared boundaries and combined effects were evaluated by mutating all accessible instances of the five motifs within each window and comparing total insulation changes to the sum of individual perturbations.

## Structural impact score distribution

Mean structural impact scores were calculated across ChromHMM state regions by averaging scores within annotated regions. Promoter-like states (7 and 19) were further subdivided based on overlap with TSSs. Differences across states were tested using ANOVA ($P < 2 \times 10^{-16}$) and confirmed with a nonparametric Kruskal–Wallis test ($P < 2.2 \times 10^{-16}$). Pairwise comparisons were performed using Tukey's honest significant difference (HSD) test and the results were visualized with mean scores ± 95% confidence intervals and a compact letter display indicating statistically distinct groups.

## Effect size distribution

To quantify the effect size of factors identified as significant in our machine learning analysis, we first gathered the predicted effect data from in silico mutagenesis for each factor and a corresponding set of random regions as controls. The effect size for each factor was determined by calculating the mean difference between the factor-specific predictions and the control values.

## State local enrichment analysis at loop anchors and boundaries

Chromatin state enrichment within loop anchors and boundaries was assessed across developmental stages. Loop and boundary coordinates were intersected with stage-specific ChromHMM annotations and local background regions were defined by extending each feature ±150 kb. Enrichment was calculated as the observed/expected ratio of each state within loops or boundaries relative to their local background, $\log_2$(transformed).

## ATAC–seq analysis

Paired-end ATAC–seq reads were processed as above. BAM files were filtered (MAPQ ≥ 10), deduplicated and merged with sambamba.

Tn5 insertion bias was corrected using deepTools alignmentSieve --ATACshift. Coverage tracks were generated with bamCoverage (RPGC normalization, 5-bp bins, blacklist excluded).

Accessible regions were identified with HMMRATAC v1.2.10 (ref. 88). We note that, in DKD embryos, HMMRATAC detected an increased number of peaks, predominantly corresponding to accessible regions in controls. HOMER de novo motif enrichment was performed on regions that lost accessibility in DKD embryos. Called peaks are provided in Supplementary Data 7.

## Data visualization

Chromatin contact data were visualized using HiGlass v0.4.7 (ref. 89). pyGenomeTracks v3.8 (ref. 90) was used to display chromatin contact data in Figs. 1 and 3 and Extended Data Figs. 1 and 6, using a natural log with scale factor set to 100. All other panels displaying chromatin contact data were plotted with CoolBox v0.3.8 (ref. 91) using a $\log_{10}$ scale.

## Statistics and reproducibility

Statistical analyses were performed in R (v4.4.0) and data visualization was carried out using the ggplot2 package (v3.5.1). Box plots were defined with boxes spanning from the first quartile (25th percentile) to the third quartile (75th percentile), with the median represented by a horizontal line within the box. Whiskers extend to the most extreme data points within 1.5× the interquartile range from the box. No collected data were excluded and data collection and analysis blinding were not applicable to this study. All statistical tests were two sided.

## Reporting summary

Further information on research design is available in the Nature Portfolio Reporting Summary linked to this article.

## Data availability

Pico-C, ATAC–seq and RNA-seq data generated in this study have been deposited in ArrayExpress under accession no. E-MTAB-14477. In addition, we analyzed data from the following publicly available datasets, including for our cluster analysis (listed in Supplementary Data 3) and ChromHMM analysis (cell-mark file information available in Supplementary Data 4): Gene Expression Omnibus accession nos. GSE83851, GSE152771, GSE86966, GSE152770, GSE161588, GSE30757, GSE125575, GSE41700, GSE130334, GSE218019, GSE58935, GSE140539, GSE86966, GSE62925, GSE141538, GSE65837 and GSE152598 and ArrayExpress accession no. E-MTAB-9156. Source data are provided with this paper.

## Code availability

Code and scripts used in this study are available via GitHub at https://github.com/vaquerizaslab/Maziak_et_al_Drosophila_Pico-C and via Zenodo at https://doi.org/10.5281/zenodo.17584456 (ref. 92). In addition, Orca models trained on our data are available through Orca-Fly: https://github.com/jzhoulab/OrcaFly.

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

## Acknowledgements

We thank M. Merkenschlager, L. Miao and E. Ing-Simmons for their critical reading of the manuscript and members of J.M.V.'s laboratory for their helpful input. We also thank the LMS Genomics Facility for assistance with RNA-seq library construction and sequencing, S. Collier at the University of Cambridge Fly Facility for support with fly injections and I. Soluri and S. Blythe for kindly providing the mat-alpha-Jabba-Trap line. This work is supported by the Medical Research Council, UK (award no. MC_UP_1605/10 to J.M.V.), the Academy of Medical Sciences and the Department of Business, Energy and Industrial Strategy (award no. APR3\1017 to J.M.V.) and the National Institutes of Health (grant nos. R35GM136298 to M.M.H. and 5T32HG002760 to H.E.B.).

## Author contributions

Conceptualization: N.M. and J.M.V. Formal analysis: N.M., Y.Z. and F.G. Funding acquisition: J.M.V., M.M.H. and H.E.B. Investigation: N.M. Methodology: N.M., M.M.H., H.E.B., A.M. and Y.K. Resources: F.G., Y.Z. and M.M.H. Supervision: J.M.V., J.Z. and M.M.H. Original draft preparation: N.M. and J.M.V. Review and editing: F.G., Y.Z., J.Z., M.M.H., H.E.B. and Y.K.

## Competing interests

The authors declare no competing interests.

## Additional information

**Extended data** is available for this paper at https://doi.org/10.1038/s41588-026-02503-3.

**Correspondence and requests for materials** should be addressed to Juan M. Vaquerizas.

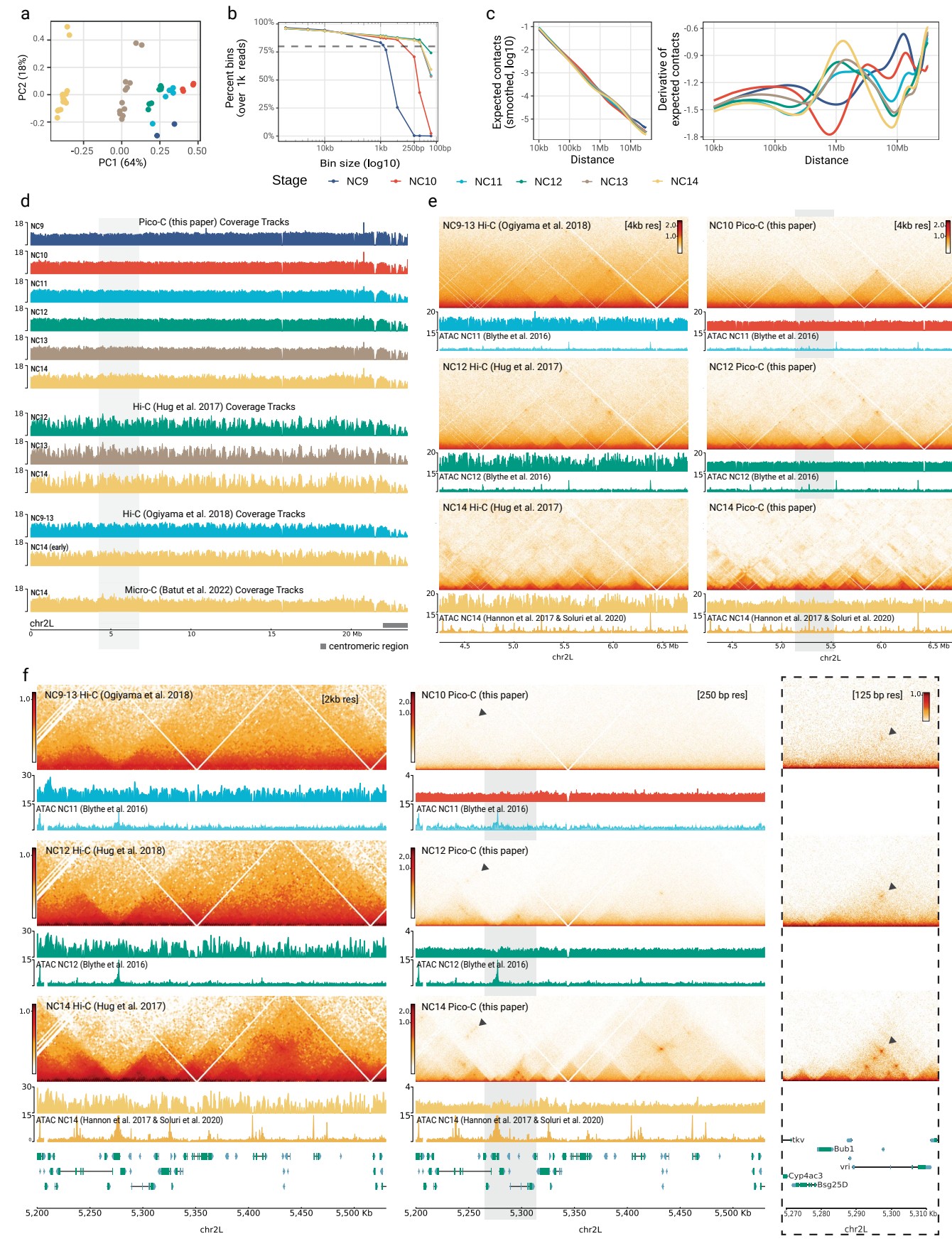

**Extended Data Fig. 1 | See next page for caption.**

**Extended Data Fig. 1 | Pico-C enables high-resolution, stage-specific profiling of early chromatin architecture. a**, Principal component analysis (PCA) conducted on 45 Pico-C libraries from stages NC9-14. **b**, Percent bins with at least 1k reads across different bin sizes. 80% cutoff to determine resolution marked in grey dotted line. **c**, Left: Average contact probability by distance for stages NC9-14. Right: the derivative of contact probability by distance, highlighting a transition in short- and long-range interactions between early (NC9-10) and later stages (NC11-14). Grey shaded area highlights the region in panel **e**. **d**, CPM-normalized coverage tracks from Pico-C (this study), Hi-C[18,25], and Micro-C data[24] down-sampled to equalize read depth. **e**, Comparison of Hi-C and Pico-C matrices with CPM-normalized coverage across a representative genomic region. While Pico-C captures similar chromatin loops as Hi-C, its higher granularity reveals their stepwise establishment more clearly. RPGC-normalized ATAC track reprocessed from public datasets[23,93,94] show no bias toward open chromatin in Pico-C versus Hi-C. **f**, A 300 kb close-up of the region highlighted in panel **e**. The uniform coverage achieved with Pico-C allows higher-resolution visualization of chromatin architecture. This reveals the stepwise formation of boundary-spanning loops (center Pico-C panel, arrowhead) as well as short-range loops at the *uri* gene (right-most panel, showing the shaded region from the middle panel; arrowhead).

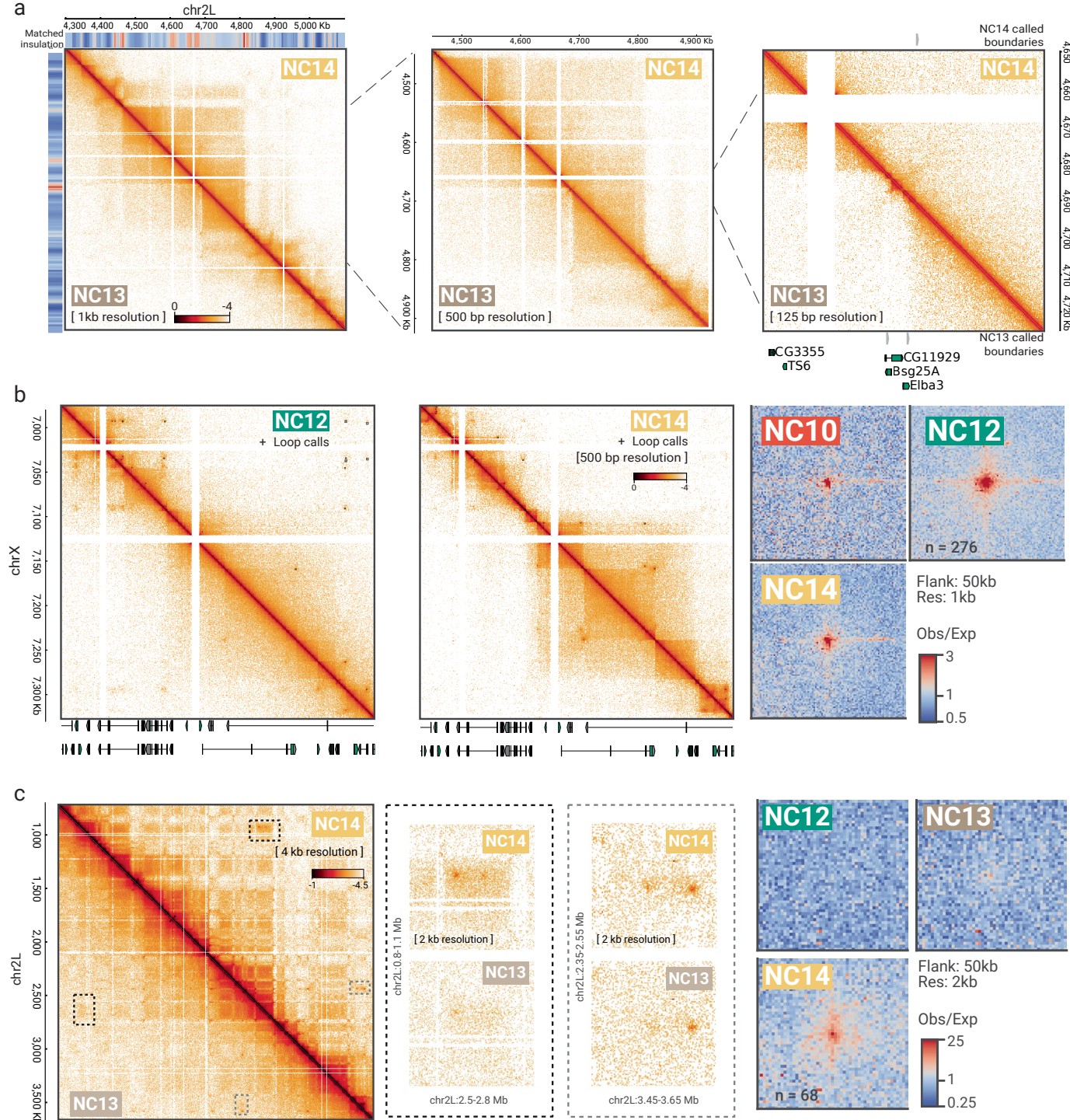

**Extended Data Fig. 2 | Pico-C resolved fine-scale chromatin dynamics in early development. a**, Example of a developmentally dynamic boundary at the *Elba3* locus between NC13 and NC14. At 1 kb resolution, the boundary present in NC13 is lost by NC14. With the high granularity of Pico-C, close-up views of the same location resolve promoter-proximal insulation at *Elba3* and *Bsg25A*, which are located ~3 kb apart. **b**, Example of dynamic looping events between NC12 (left) and NC14 (right). Top panels show normalized Pico-C contact maps with annotated loops calls; bottom panels display the same maps without annotation overlays. Right: Aggregate analysis shows NC12 specific loops have stronger focal dot at NC12 than NC10 and NC14. Aggregates plotted at 1 kb resolution with 50 kb flanks. **c**, Previously described meta-loops[28] are present at NC14 and some detectable as early as NC13. Top: normalized Pico-C contact map for NC14; bottom: the same region at NC13, both shown at 4 kb resolution. Centre: close-up view of meta-loop and meta-loop like loci at both stages, displayed at 2 kb resolution. Right: aggregates of meta-loops[28] show progressive gain first visible at NC13.

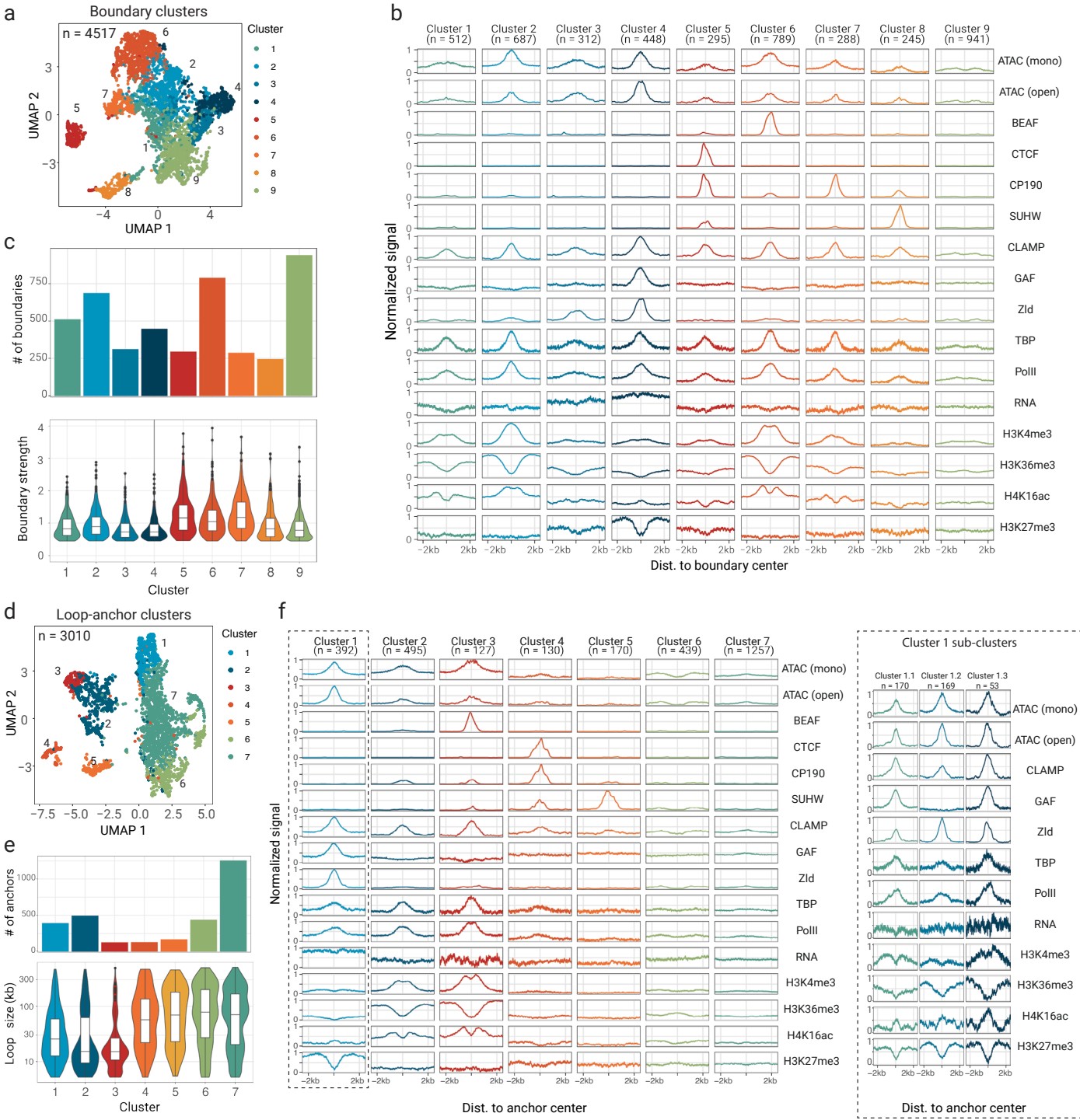

**Extended Data Fig. 3 | Distinct chromatin features define boundary and loop-anchor clusters. a-c, Boundary cluster analysis. a,** UMAP embedding of boundary regions based on chromatin features, colored by cluster assignment (n = 9 clusters). **b,** Normalized signal profiles for transcription factors, histone modifications, and chromatin accessibility (ATAC open <100 bp fragments; ATAC mono 180-250 bp fragments) centered at boundaries across each cluster. All signals are derived from publicly available datasets (a list of datasets used can be found in Supplementary Data 3). **c,** (Top) Bar plot showing the number of boundaries per cluster. (Bottom) Violin plots showing boundary strength (quantified by insulation score) per cluster. **d-f, Loop-anchor cluster analysis.**

**d,** UMAP embedding of loop anchors, colored by loop-anchor cluster (n = 7 clusters). **e,** (Top) Number of anchors per cluster. (Bottom) Violin plots showing loop size distribution per cluster. **f,** Chromatin and transcription factor signal centered at loop anchors by cluster. Cluster 1 shows strong enrichment for pioneer factors (GAF, Zelda, CLAMP) and active chromatin features. This cluster can be further subdivided into three subclusters based on differential pioneer factor occupancy: GAF-high (Cluster 1.1), Zelda-high (Cluster 1.2), and anchors co-bound by both (Cluster 1.3). All signals are derived from publicly available datasets (a list of datasets used can be found in Supplementary Data 3).

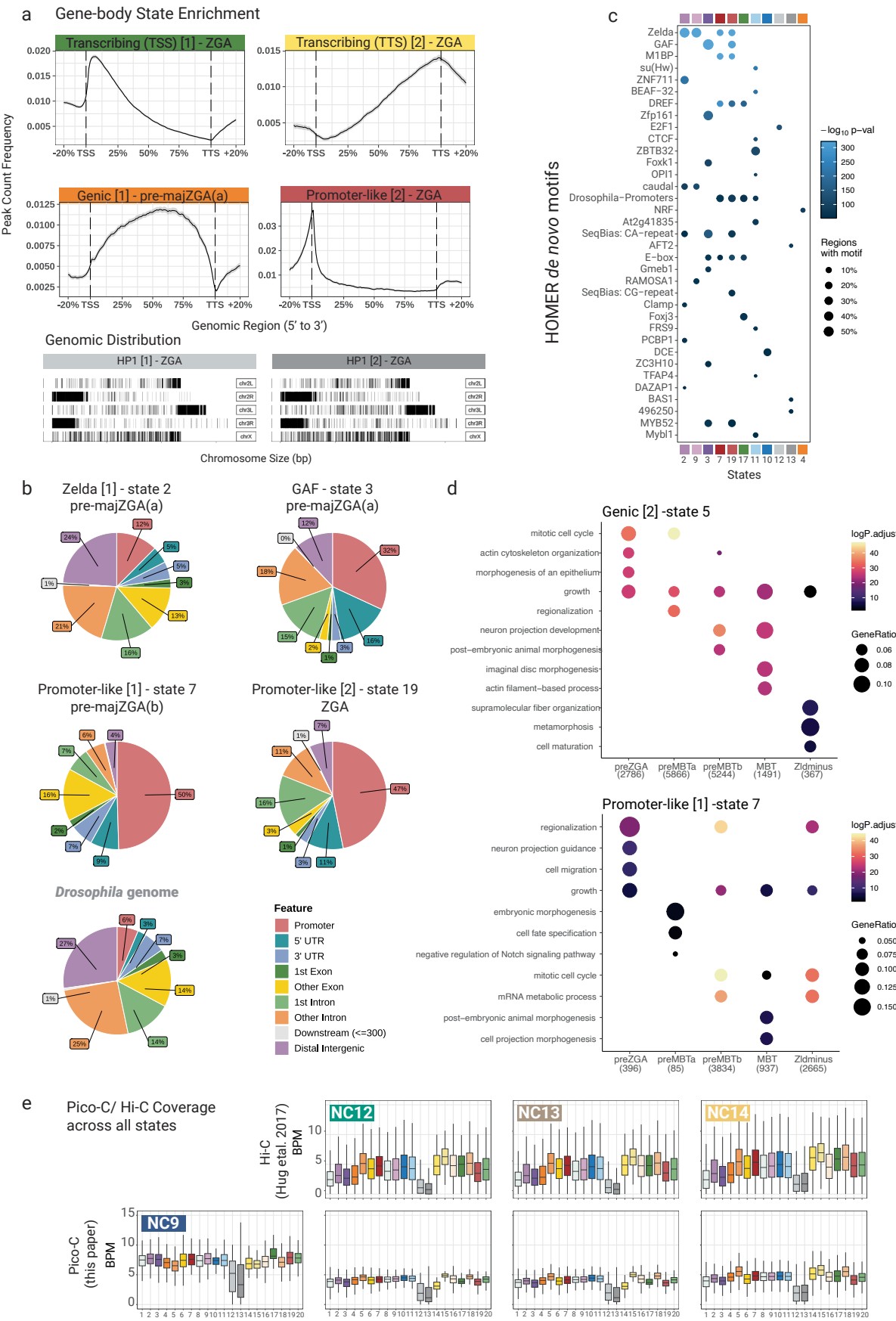

**Extended Data Fig. 4 | See next page for caption.**

**Extended Data Fig. 4 | General enrichment and functional analysis of chromatin states reveal diverse state trends. a**, Top: example gene body enrichment of transcribing, genic, and promoter-like states. Bottom: genome-wide distribution of HP1 state regions shows centromeric and telomeric enrichment. **b**, Pie-graphs of genome wide distribution for pioneer states 2 and 3, and promoter-like states 7 and 19. Bottom shows the proportion across the *Drosophila* genome for comparison. **c**, HOMER de-novo motif enrichment results passing cutoff p-value 1e-50. State colors displayed top and bottom, and state numbers on the bottom. Left to right, purples are pioneer factor states Zelda [1], Zelda [2], and GAF; reds are Promoter-like states [1] and [2]; green is Transcribing (TSS) [1]; blues are open and nucleosomal states, greys are HP1 states [1] and [2]; orange is Genic state [1]. **d**, Gene ontology (GO) results of genes with state 5 (top; Genic [2]) or state 7 (bottom, Promoter-like [1]). **e**, Comparison of average coverage across chromatin states in Hi-C[18] and Pico-C data. The primary states showing differential coverage are the HP1-associated states, likely reflecting reduced mappability at centromeric regions.

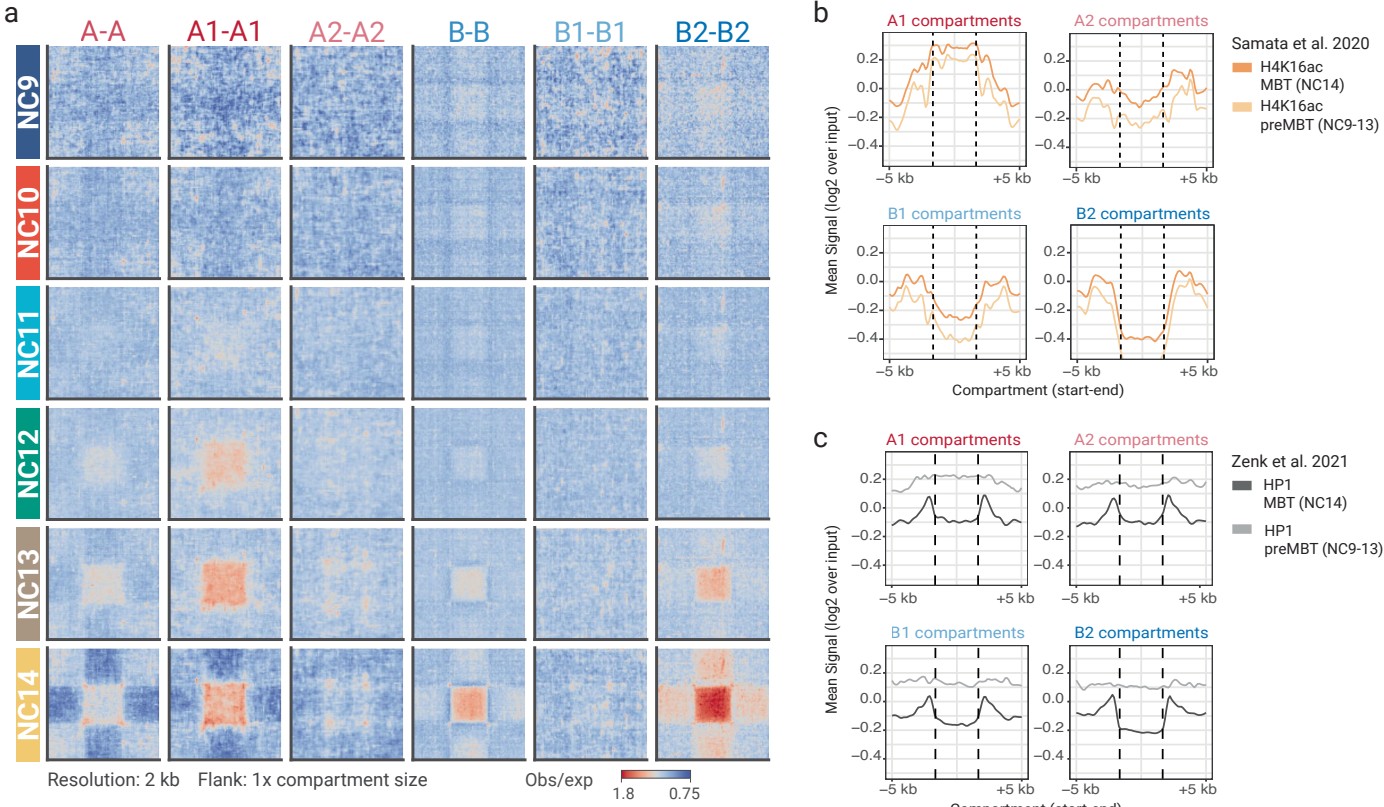

Resolution: 2 kb    Flank: 1x compartment size    Obs/exp 1.8  0.75

**Extended Data Fig. 5 | Pico-C captures temporal dynamics of compartmentalization preceding ZGA. a**, Aggregate contact maps (observed/ expected) centered on compartment and sub-compartment pairs across developmental stages NC9–NC14, plotted at 2 kb resolution. A.1–A.1 and B.2–B.2 interactions emerge as the strongest and most structured contacts by NC12– NC14, while A.2 and B.1 compartments show weaker interactions overall.

**b**, H4K16ac ChIP-seq enrichment (log$_2$ signal over input) reprocessed from publicly available data[34], centered on each sub-compartment type. A.1 compartments are enriched for H4K16ac prior to ZGA (NC9–13), suggesting an inherited epigenetic signature. **c**, HP1 ChIP-seq enrichment (log$_2$ signal over input) reprocessed from publicly available data[16], centered on each sub-compartment type.

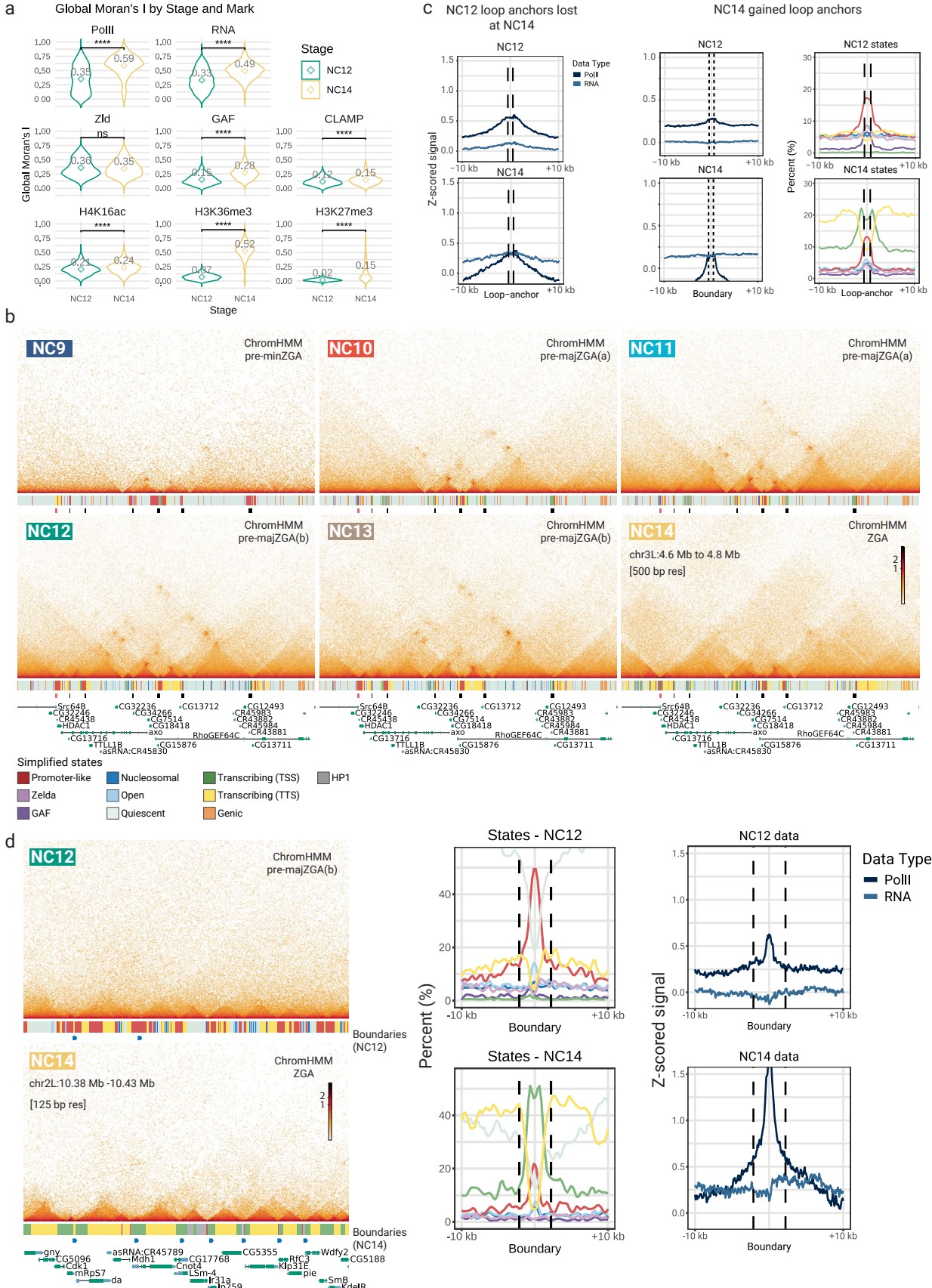

**Extended Data Fig. 6 | See next page for caption.**

**Extended Data Fig. 6 | Genome-wide correlation trends reveal distinct patterns between NC12 and NC14, accompanied by chromatin state changes.**
**a**, Comparison of Global Moran's I at 200 kb sliding windows of the genome at NC12 and NC14 for different mark inputs. The plots represent the distribution and density of Global Moran's I scores across bins (n = 666), The means are highlighted by the diamond shape and their values are written in grey. Statistical comparisons between stages were made using the unpaired Wilcoxon rank-sum test, and significant differences (p-values) are annotated on the plots with the following significance levels: * for p < 0.05, ** for p < 0.01, *** for p < 0.001, and **** for p < 0.0001. **b**, Pico-C contact maps at the *HDAC1* locus from NC9-14 shows dynamic looping events, some visible as early as NC9. Simplified ChromHMM

states below show that loop anchors are marked with promoter-like (red) and pioneer (purple) states even in pre-minZGA ChromHMM. **c**, Scaled Pol II and RNA-seq signals at loop anchors. Left: Average log$_2$ signal over input at NC12 loop anchors, scaled across NC12 and NC14 jointly to allow direct comparison. Center: Scaled signal at NC14-gained loop anchors, also scaled together between stages. Right: ChromHMM states at NC14-gained loop anchors. **d**, Left: Pico-C contact maps (125 bp resolution) of a gene-dense region at NC12 and NC14, with simplified ChromHMM states and boundary calls. Center: ChromHMM states at the same regions. Right: Scaled Pol II and RNA-seq signal at boundaries with ≥3-fold increased insulation at NC14, scaled jointly across stages.

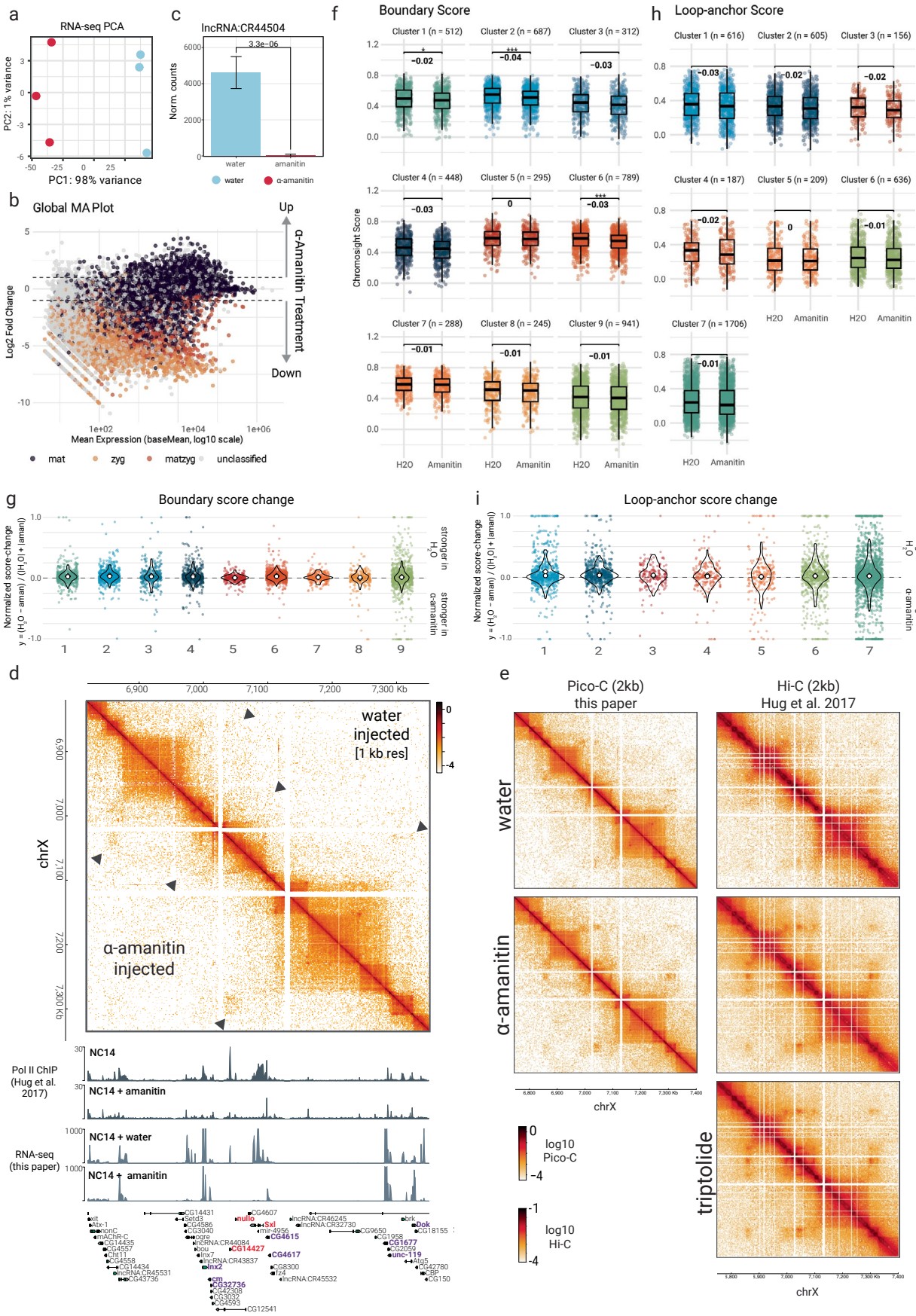

**Extended Data Fig. 7 | See next page for caption.**

**Extended Data Fig. 7 | Transcriptional repression and chromatin architecture changes in α-amanitin-treated embryos. a**, Principal component analysis (PCA) of RNA-seq replicates from water-injected and α-amanitin-injected embryos at NC14. **b**, Global MA plot showing a widespread reduction in expression of zygotically transcribed genes (as defined previously[8]) following α-amanitin treatment. **c**, Expression levels of zygotic long non-coding RNA CR44504, which is predicted to be involved in miRNA-mediated post-transcriptional gene silencing, and mir-309, a zygotic miRNA involved in maternal RNA clearance, is significantly reduced in α-amanitin-treated embryos (p = 3.3e−06). **d**, Example genomic locus showing retention of loops in α-amanitin-treated embryos. Retained loops overlap pre-MBT genes (highlighted in red) and are supported by matched Pol II ChIP-seq[18] and RNA-seq data (this paper). Retained transcripts often correspond to maternally deposited genes and are shown in purple. **e**, Comparison of contact maps at the same locus using Pico-C (this study) and Hi-C[18]. Contact patterns in α-amanitin-treated embryos closely resemble both α-amanitin and triptolide Hi-C maps. **f**, Boxplots of Chromosight scores for boundary clusters comparing water-injected and α-amanitin-injected embryos. Scores are grouped by cluster identity. Statistical comparisons were performed using an unpaired Wilcoxon rank-sum test with Benjamini–Hochberg correction for multiple testing. Annotated values reflect the mean difference in scores between α-amanitin and water-injected embryos. Statistical significance is indicated as follows: *** for adj. p < 0.001, ** for adj. p < 0.01, and * for adj. p < 0.05. **g**, Normalized changes in Chromosight boundary scores at individual loci between water-injected and α-amanitin-injected embryos, grouped by cluster identity. The normalized change was calculated as ($H_2O$−amanitin)/ (|$H_2O$|+|amanitin|) with positive values indicating stronger scores in water- and negative values indicating stronger scores in α-amanitin-injected embryos. Violin plots show the distribution of values per cluster; diamonds denote cluster means and circles denote cluster medians. **h**, Boxplots of Chromosight loop anchor cluster scores comparing water- and α-amanitin-injected embryos. Statistics and annotations as in **f**. **i**, Normalized changes in Chromosight loop-anchor scores between water-injected and α-amanitin-injected embryos, grouped by cluster identity. Methods and annotations as in **g**.

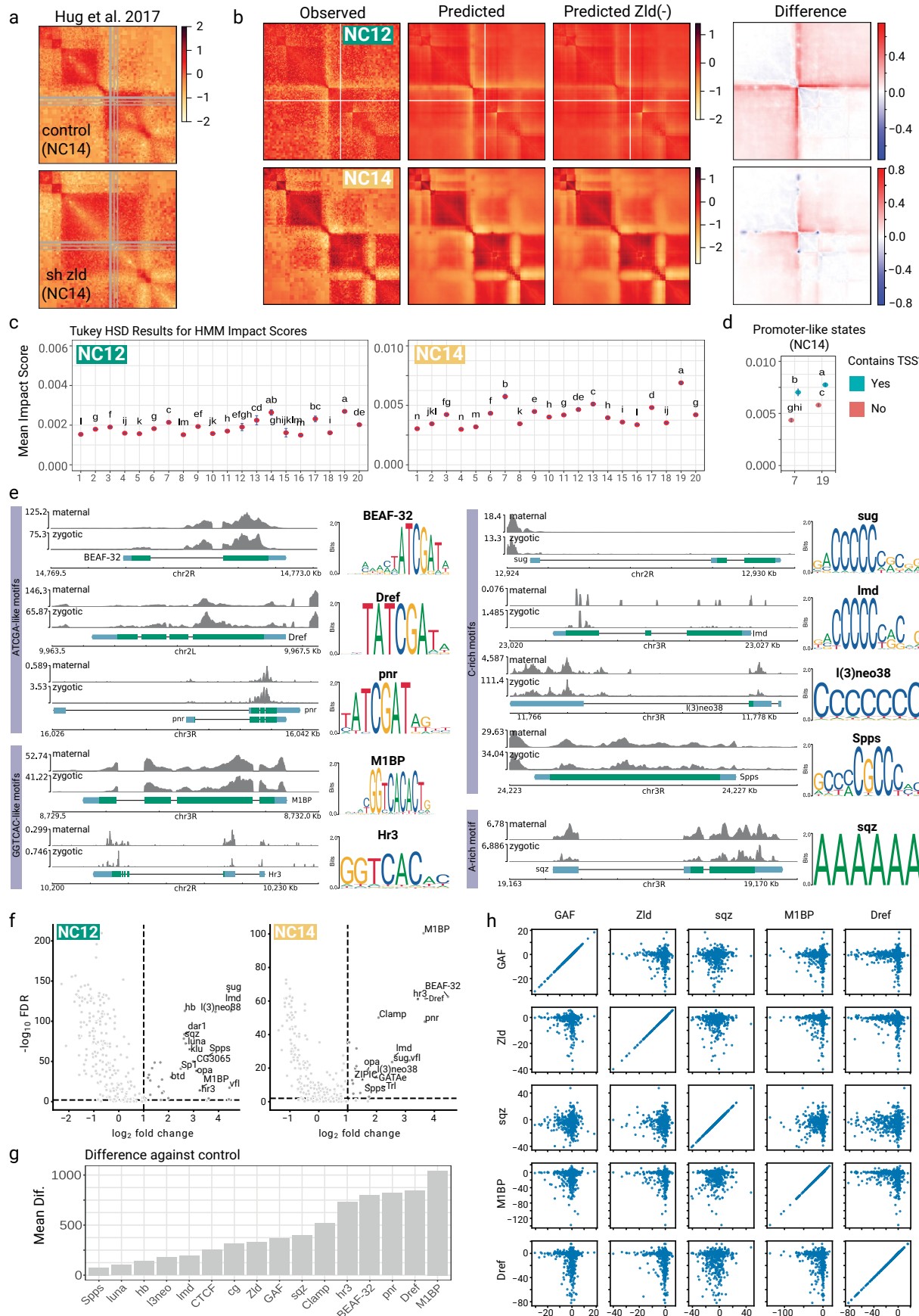

**Extended Data Fig. 8 | See next page for caption.**

**Extended Data Fig. 8 | Orca-based predictions highlight the structural consequences of Zelda mutation and uncover key motifs affecting genome architecture. a**, Experimentally observed loss of insulation at the *bitesize* (*btsz*) locus following Zelda depletion (sh zld)[18]. Normalized contact maps (log$_2$ observed/expected) at NC14 reveal loss of boundary in sh zld compared to control embryos. **b**, Observed data, model predictions, and predictions after mutating Zld motifs at the *btsz* region. The models predict a disruption at the boundary when Zld motifs are mutated in silico, highlighted with brackets, with a pronounced effect observed at NC12. **c**, Statistical analysis used to compare Structural Impact Scores across different ChromHMM state regions. An ANOVA was first conducted to assess overall differences among the states, and the subsequent Tukey HSD test was used for pairwise comparisons. Red points represent the mean values, and blue bars indicate the 95% confidence intervals for each mean. The letters in the compact letter display denote grouping based on the Tukey's test, where states sharing the same letter are not significantly different. **d**, Same analysis carried out in previous panel with promoter-like states 7 and 19 further divided based on whether the regions coincide with promoters in NC14 model predictions. State 19, the predominant promoter state at NC14,

shows that regions not found at promoters still rank higher than other state regions and are assigned a unique letter. **e**, Expression of motif screen hits, grouped by motif similarity, using RNA-seq data spanning NC7-NC14[7], split into maternal and zygotic transcripts. The plot shows RPGC-normalized reads. **f**, The log2 fold change in the frequency of motifs in regions with an impact score > 0.01 for NC12 and > 0.02 for NC14, compared to their frequency in randomly chosen background regions. The motifs listed all exhibit a high fold change and a significant p-value. **g**, The effect size between various factors and control regions tested with NC14 model, with factors ordered by the magnitude of their mean difference. Each factor's effect was calculated by subtracting the mean value of its control group from the mean value of the factor group. The results highlight the relative impact of each factor, with positive values indicating a greater effect than the control. **h**, Pairwise comparison of insulation changes from individual motif mutagenesis across five transcription factors. Each scatterplot compares the predicted insulation change at boundary regions for two different motif perturbations. Strong diagonal trends are absent, indicating largely distinct contributions to insulation from each factor.

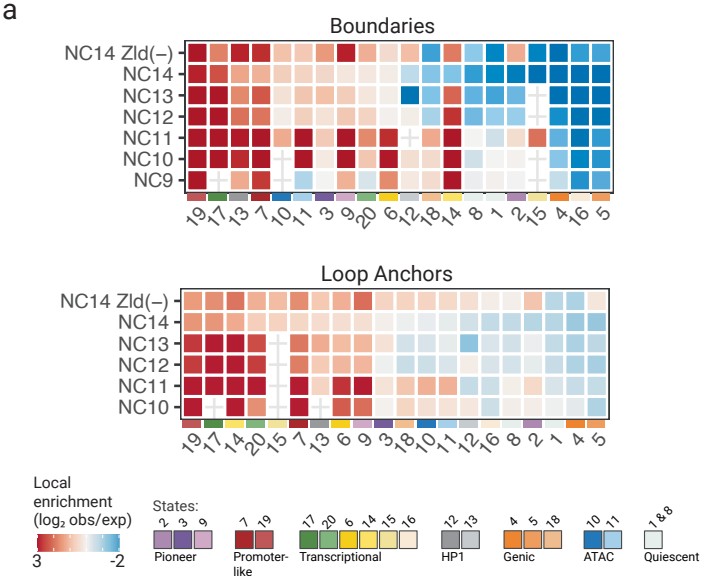

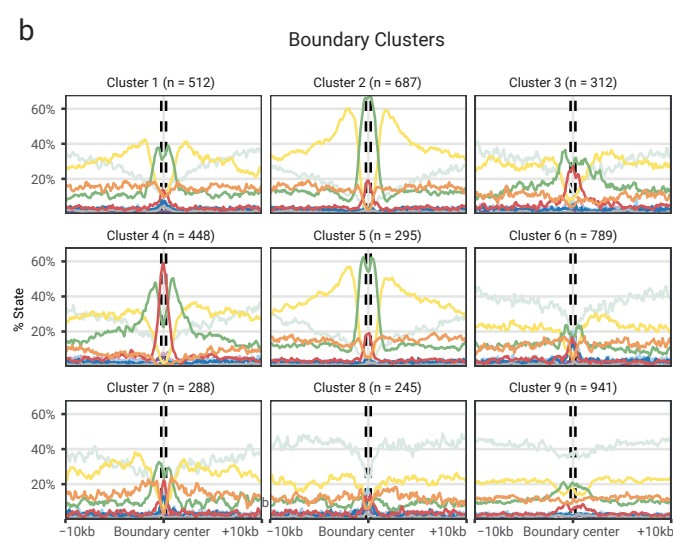

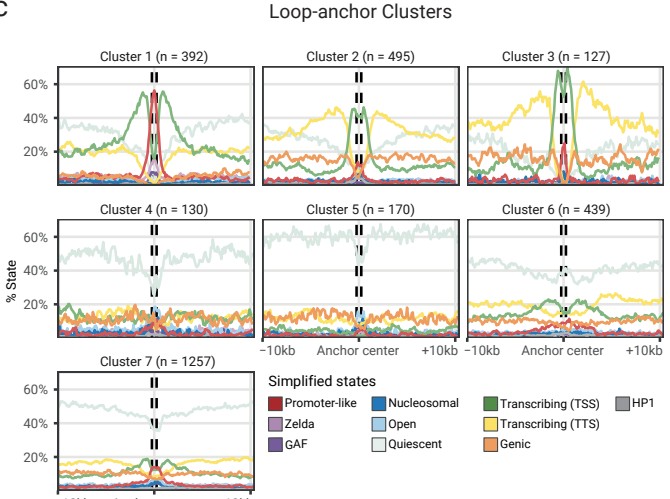

**Extended Data Fig. 9 | Chromatin state enrichment across boundary and loop-anchor clusters. a**, Local enrichment analysis of states at loop anchors and boundaries across developmental stages and in Zelda-depleted embryos (Zld(−)). Local enrichment was calculated using regions 150 kb upstream and downstream of boundaries and loop anchors. Promoter-like states remain enriched at loop anchors even in the absence of Zelda. **b**, Chromatin state composition surrounding boundary centers (±10 kb) across nine boundary clusters. **c**, Chromatin state composition surrounding loop-anchor centers (±10 kb) across seven loop-anchor clusters.

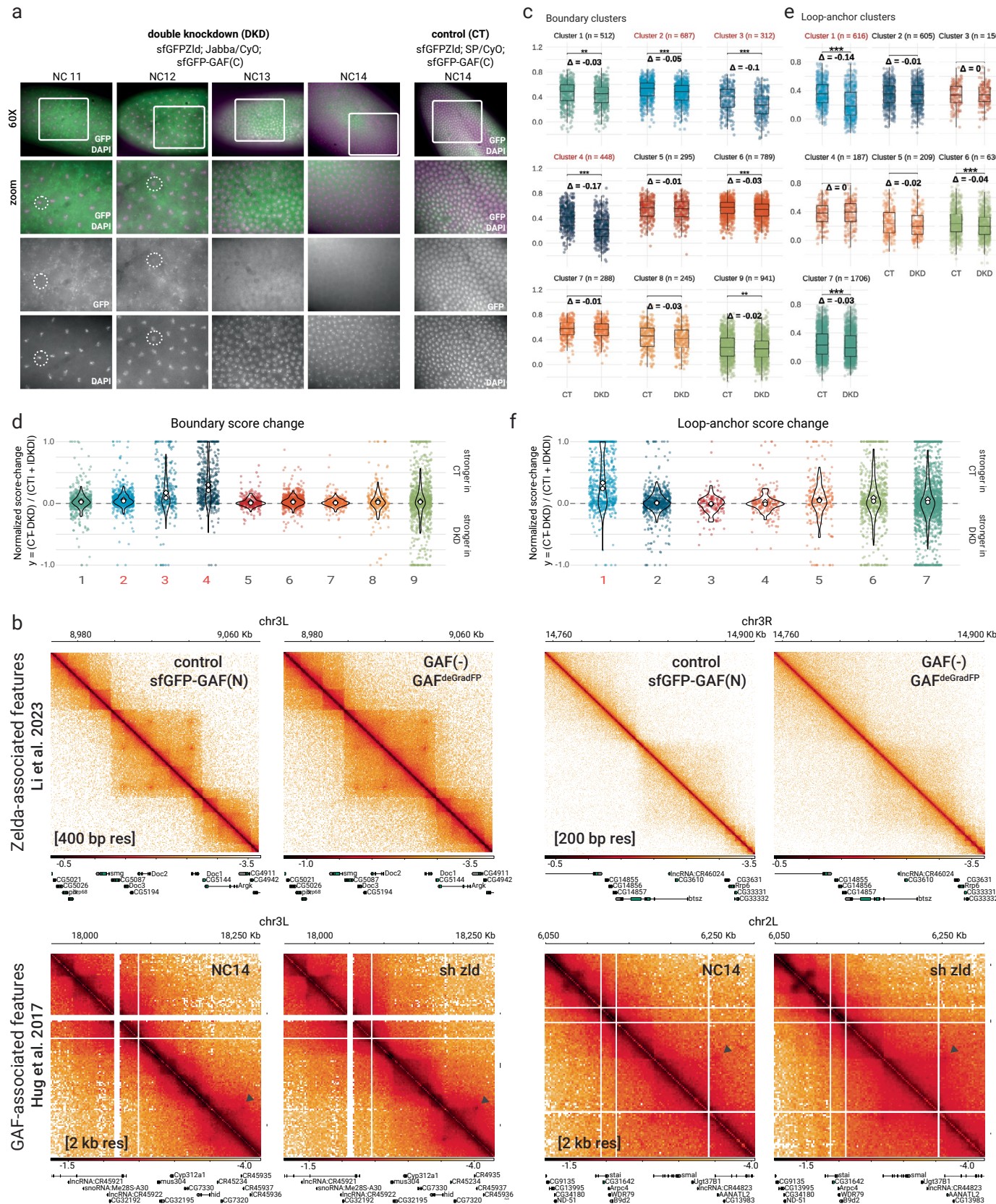

**Extended Data Fig. 10 | See next page for caption.**

**Extended Data Fig. 10 | Simultaneous depletion of Zelda and GAF leads to an additive loss of chromatin architecture. a**, Imaging of embryos at different nuclear cycles (NC11–14) shows exclusion of GFP-tagged Zelda and GAF from nuclei in double knockdown (DKD) embryos (sfGFPZld;Jabba/CyO; sfGFPGAF/C). In contrast, control (CT) embryos (sfGFPZld;Sp/CyO; sfGFPGAF/+) display strong nuclear GFP signal at NC14. Insets show zoomed-in views highlighting the absence (DKD) or presence (CT) of GFP signal in nuclei (dashed circles), with DAPI used as a nuclear stain. **b**, Micro-C and Hi-C maps demonstrate factor-specific chromatin organization. Top: Zelda-associated boundaries and loops[18,48] remain largely intact in GAF-depleted data (GAF[deGradFP])[20], although a mild reduction is observed. Bottom: GAF-dependent loops[20], marked with arrowheads, persist in Zelda-depleted embryos (sh zld)[18]. Resolution for each panel is indicated. **c**, Boxplots of Chromosight scores for boundary clusters comparing controls (CT) with double-knockdown (DKD). Scores are grouped by cluster identity. Clusters associated with GAF and Zelda are labeled in red. Statistical comparisons were performed using an unpaired Wilcoxon rank-sum test with Benjamini–Hochberg correction for multiple testing. Annotated Δ values reflect the mean

difference in scores between DKD and CT embryos. Statistical significance is indicated as follows: *** for adj. p < 0.001, ** for adj. p < 0.01, and * for adj. p < 0.05. **d**, Normalized changes in Chromosight boundary scores at individual loci between CT and DKD embryos, grouped by cluster identity. Normalized changes with positive values indicate stronger scores in CT and negative values indicate stronger scores in DKD. Clusters associated with GAF and Zelda are labeled in red. Violin plots show the distribution of values per cluster; diamonds denote cluster means and circles denote cluster medians. **e**, Boxplots of Chromosight scores for loop-anchor clusters (right) comparing controls with double-knockdown (DKD). Scores are grouped by cluster identity. Clusters associated with GAF and Zelda are labeled in red. Stats and annotations as in c (BH-corrected Wilcoxon; *, **, *** for adj. p < 0.05, 0.01, 0.001). **f**, Normalized changes in Chromosight loop-anchor scores at individual loci between CT and DKD embryos, grouped by cluster identity. Normalized changes with positive values indicate stronger scores in CT and negative values indicate stronger scores in DKD. Methods and annotations as in d.

# Reporting Summary

## Statistics

For all statistical analyses, confirm that the following items are present in the figure legend, table legend, main text, or Methods section.

| n/a | Confirmed | |
|-----|-----------|---|
| ☐ | ☒ | The exact sample size (*n*) for each experimental group/condition, given as a discrete number and unit of measurement |
| ☒ | ☐ | A statement on whether measurements were taken from distinct samples or whether the same sample was measured repeatedly |
| ☐ | ☒ | The statistical test(s) used AND whether they are one- or two-sided<br>*Only common tests should be described solely by name; describe more complex techniques in the Methods section.* |
| ☒ | ☐ | A description of all covariates tested |
| ☒ | ☐ | A description of any assumptions or corrections, such as tests of normality and adjustment for multiple comparisons |
| ☐ | ☒ | A full description of the statistical parameters including central tendency (e.g. means) or other basic estimates (e.g. regression coefficient) AND variation (e.g. standard deviation) or associated estimates of uncertainty (e.g. confidence intervals) |
| ☐ | ☒ | For null hypothesis testing, the test statistic (e.g. *F*, *t*, *r*) with confidence intervals, effect sizes, degrees of freedom and *P* value noted<br>*Give P values as exact values whenever suitable.* |
| ☒ | ☐ | For Bayesian analysis, information on the choice of priors and Markov chain Monte Carlo settings |
| ☒ | ☐ | For hierarchical and complex designs, identification of the appropriate level for tests and full reporting of outcomes |
| ☒ | ☐ | Estimates of effect sizes (e.g. Cohen's *d*, Pearson's *r*), indicating how they were calculated |

*Our web collection on statistics for biologists contains articles on many of the points above.*

## Software and code

Policy information about availability of computer code

| | |
|---|---|
| Data collection | No software was used for data collection. |
| Data analysis | BWA-MEM version 0.7.17, Li and Durbin, 2009.<br>Bowtie2 version 2.5.4, Langmead and Salzberg, 2012.<br>BEDTools version 2.29.2, Quinlan and Hall, 2010.<br>Pairtools version 1.0.3, Abdennur et al., 2023.<br>FAN-C version 0.9.18, Kruse et al., 2020.<br>Cooler version 0.8.6, Abdennur and Mirny, 2019.<br>cooltools version 0.5.3, Abdennur et al., 2022.<br>coolpup.py version 1.1.0, Flyamer et al., 2020.<br>Mustache version 1.3.2, Roayaei Ardakany et al., 2020.<br>Sambamba version 1.0.1, Tarasov et al., 2015.<br>SAMtools version 1.18, Danecek et al., 2021.<br>STAR version 2.7.10a, Dobin et al., 2012.<br>HOMER version 4.11, Heinz et al., 2010.<br>HMMRATAC version 1.2.10, Tarbell and Liu, 2019.<br>deepTools version 3.5.5, Ramírez et al., 2014.<br>pyGenomeTracks version 3.8, Ramírez et al., 2018.<br>CoolBox version 0.3.8, Xu et al., 2021.<br>FastQC version 0.12.1<br>Calder2 version 0.7, Liu et al. 2021 |

```
Chromosight version 1.6.3, Matthey-Doret et al. 2020
DESep2 version 1.44.0,  Love ,Huber, and Anders 2014

R version 4.4.0
ggplot2 version 3.5.1, Wickham 2016.
ChIPseeker version 1.40.0, Yu et al., 2015.
GenomicRanges version 1.56, Lawrence et al., 2013.
hicVennDiagram version 1.2, Ou, 2024.
ComplexUpset version 1.3.3
```

For manuscripts utilizing custom algorithms or software that are central to the research but not yet described in published literature, software must be made available to editors and reviewers. We strongly encourage code deposition in a community repository (e.g. GitHub). See the Nature Portfolio guidelines for submitting code & software for further information.

## Data

Policy information about availability of data

All manuscripts must include a data availability statement. This statement should provide the following information, where applicable:
- Accession codes, unique identifiers, or web links for publicly available datasets
- A description of any restrictions on data availability
- For clinical datasets or third party data, please ensure that the statement adheres to our policy

The Pico-C data produced in this study will be available from ArrayExpress with the following accession number:
Accession: E-MTAB-14477

We analyzed data from the following publicly available datasets: BioProject accessions PRJNA327205, PRJNA640380, PRJNA343106, PRJNA640381, PRJNA678806, PRJNA144105, PRJNA516863, PRJNA769043, PRJNA177994, PRJNA535514, PRJNA901932, PRJNA254188, PRJNA590156, PRJNA905853, PRJNA266120, PRJNA593838, PRJNA275186, PRJNA639811.

## Research involving human participants, their data, or biological material

Policy information about studies with human participants or human data. See also policy information about sex, gender (identity/presentation), and sexual orientation and race, ethnicity and racism.

| Reporting on sex and gender | *Use the terms sex (biological attribute) and gender (shaped by social and cultural circumstances) carefully in order to avoid confusing both terms. Indicate if findings apply to only one sex or gender; describe whether sex and gender were considered in study design; whether sex and/or gender was determined based on self-reporting or assigned and methods used. Provide in the source data disaggregated sex and gender data, where this information has been collected, and if consent has been obtained for sharing of individual-level data; provide overall numbers in this Reporting Summary. Please state if this information has not been collected. Report sex- and gender-based analyses where performed, justify reasons for lack of sex- and gender-based analysis.* |
|---|---|
| Reporting on race, ethnicity, or other socially relevant groupings | *Please specify the socially constructed or socially relevant categorization variable(s) used in your manuscript and explain why they were used. Please note that such variables should not be used as proxies for other socially constructed/relevant variables (for example, race or ethnicity should not be used as a proxy for socioeconomic status). Provide clear definitions of the relevant terms used, how they were provided (by the participants/respondents, the researchers, or third parties), and the method(s) used to classify people into the different categories (e.g. self-report, census or administrative data, social media data, etc.) Please provide details about how you controlled for confounding variables in your analyses.* |
| Population characteristics | *Describe the covariate-relevant population characteristics of the human research participants (e.g. age, genotypic information, past and current diagnosis and treatment categories). If you filled out the behavioural & social sciences study design questions and have nothing to add here, write "See above."* |
| Recruitment | *Describe how participants were recruited. Outline any potential self-selection bias or other biases that may be present and how these are likely to impact results.* |
| Ethics oversight | *Identify the organization(s) that approved the study protocol.* |

Note that full information on the approval of the study protocol must also be provided in the manuscript.

## Field-specific reporting

Please select the one below that is the best fit for your research. If you are not sure, read the appropriate sections before making your selection.

☒ Life sciences          ☐ Behavioural & social sciences          ☐ Ecological, evolutionary & environmental sciences

For a reference copy of the document with all sections, see nature.com/documents/nr-reporting-summary-flat.pdf

# Life sciences study design

All studies must disclose on these points even when the disclosure is negative.

| | |
|---|---|
| Sample size | No statistical methods were used to predetermine sample size. We followed standards from the literature when determining sample size. |
| Data exclusions | No data were excluded from the analyses. |
| Replication | Principal component analysis and correlations were used to assess similarity of biological replicates. |
| Randomization | Due to the nature of the experiments, there was no need of randomisation. |
| Blinding | Due to the nature of the experiments, the authors were not blinded to allocation during experiments and analysis. |

# Reporting for specific materials, systems and methods

We require information from authors about some types of materials, experimental systems and methods used in many studies. Here, indicate whether each material, system or method listed is relevant to your study. If you are not sure if a list item applies to your research, read the appropriate section before selecting a response.

### Materials & experimental systems

| n/a | Involved in the study |
|---|---|
| ☐ | ☒ Antibodies |
| ☒ | ☐ Eukaryotic cell lines |
| ☒ | ☐ Palaeontology and archaeology |
| ☐ | ☒ Animals and other organisms |
| ☒ | ☐ Clinical data |
| ☒ | ☐ Dual use research of concern |
| ☒ | ☐ Plants |

### Methods

| n/a | Involved in the study |
|---|---|
| ☒ | ☐ ChIP-seq |
| ☒ | ☐ Flow cytometry |
| ☒ | ☐ MRI-based neuroimaging |

## Antibodies

| | |
|---|---|
| Antibodies used | rabbit anti-GFP (Abcam #ab290), DyLight 488-conjugated goat anti-rabbit (ThermoFisher #35552) |
| Validation | From provider: Anti-GFP antibody (ab290) is a rabbit polyclonal antibody and is validated for use in ELISA, EM, ICC/IF, IHC-FoFr, IHC-Fr, IHC-FrFl, IHC-P, IP and WB in human samples.<br>From provider: Product # 35552 has been successfully used in Western blot, IF, ICC, IHC, IP and FACS applications. |

## Animals and other research organisms

Policy information about studies involving animals; ARRIVE guidelines recommended for reporting animal research, and Sex and Gender in Research

| | |
|---|---|
| Laboratory animals | The following Drosophila melanogaster strain was used:<br>yw; eGFP-PCNA<br>sfGFP-Zelda; Jabba-Trap/CyO; sfGFP-GAF<br>and sfGFP-Zelda; Sp/CyO; sfGFP-GAF |
| Wild animals | No wild animals were used in this study. |
| Reporting on sex | The samples were not specifically selected based on sex and represent a mixed population. |
| Field-collected samples | No wild animals were used in this study. |
| Ethics oversight | Drosophila melanogaster culture was performed in accordance with local and national requirements. No specific ethical approval is required for experiments with Drosophila embryos. |

Note that full information on the approval of the study protocol must also be provided in the manuscript.

## Plants

| | |
|---|---|
| Seed stocks | *Report on the source of all seed stocks or other plant material used. If applicable, state the seed stock centre and catalogue number. If plant specimens were collected from the field, describe the collection location, date and sampling procedures.* |
| Novel plant genotypes | *Describe the methods by which all novel plant genotypes were produced. This includes those generated by transgenic approaches, gene editing, chemical/radiation-based mutagenesis and hybridization. For transgenic lines, describe the transformation method, the number of independent lines analyzed and the generation upon which experiments were performed. For gene-edited lines, describe the editor used, the endogenous sequence targeted for editing, the targeting guide RNA sequence (if applicable) and how the editor was applied.* |
| Authentication | *Describe any authentication procedures for each seed stock used or novel genotype generated. Describe any experiments used to assess the effect of a mutation and, where applicable, how potential secondary effects (e.g. second site T-DNA insertions, mosiacism, off-target gene editing) were examined.* |

