## [Peer Review File · Nature Genetics]

3D Genome Reorganization Foreshadows Zygotic Genome Activation in *Drosophila*

Corresponding Author: Professor Juan Vaquerizas

Version 0:

Decision Letter:

4th Nov 2024

Dear Professor Vaquerizas,

Your Article, "Extensive 3D Genome Reorganization Foreshadows Zygotic Genome Activation in *Drosophila*" has now been seen by 3 referees. You will see from their comments copied below that while they find your work of considerable potential interest, they have raised quite substantial concerns that must be addressed. In light of these comments, we cannot accept the manuscript for publication, but would be very interested in considering a revised version that addresses these serious concerns.

Reviewer #1 thinks your study is of incremental advance and lacks validation. They suggest developing the paper further by adding mechanistic insights.

Reviewer #2 appreciates the value of the pico-C datasets and computational analyses, but thinks some points should be clarified and asks for functional work.

Reviewer #3 likes the paper, but thinks the majority of the results are correlative and asks for more mechanistic insights that can help discriminate where 3D genome organization is instructive or a consequence of transcriptional activity in *Drosophila*.

We hope you will find the referees' comments useful as you decide how to proceed. If you wish to submit a substantially revised manuscript, please bear in mind that we will be reluctant to approach the referees again in the absence of major revisions.

To guide the scope of the revisions, the editors discuss the referee reports in detail within the team, including with the chief editor, with a view to identifying key priorities that should be addressed in revision and sometimes overruling referee requests that are deemed beyond the scope of the current study. In this case, we ask you to address reviewers' comments in full. We hope that you will find the prioritised set of referee points to be useful when revising your study. Please do not hesitate to get in touch if you would like to discuss these issues further.

If you choose to revise your manuscript taking into account all reviewer and editor comments, please highlight all changes in the manuscript text file. At this stage we will need you to upload a copy of the manuscript in MS Word .docx or similar editable format.

*2) If you have not done so already please begin to revise your manuscript so that it conforms to our Article format instructions, available [here](http://www.nature.com/ng/authors/article_types/index.html). Refer also to any guidelines provided in this letter.

Please be aware of our [guidelines](https://www.nature.com/nature-research/editorial-policies/image-integrity) on digital image standards.

Link Redacted

If you wish to submit a suitably revised manuscript we would hope to receive it within 6 months. If you cannot send it within this time, please let us know. We will be happy to consider your revision so long as nothing similar has been accepted for publication at Nature Genetics or published elsewhere. Should your manuscript be substantially delayed without notifying us in advance and your article is eventually published, the received date would be that of the revised, not the original, version.

Thank you for the opportunity to review your work.

Sincerely,
Chiara

Chiara Anania, PhD
Associate Editor
Nature Genetics
<https://orcid.org/0000-0003-1549-4157>

Referee expertise:

Referee #1: gene regulation, drosophila

Referee #2: 3D genome

Referee #3: 3D genome, methods

Reviewers' Comments:

Reviewer #1 (Remarks to the Author):

This study develops Pico-C, an improved low-input Micro-C method, to generate very high-resolution chromatin interaction maps of early fly embryos during successive interphases of nuclear cleavage cycles (NC9 to NC14), covering the minor and major waves of zygotic genome activation (ZGA). The authors identified increasing chromatin loop sizes as development progresses. They re-analyzed hundreds of published chromatin and RNA-profiling datasets using ChromHMM to categorize loci into 20 chromatin states across various stages (pre-minZGA, early pre-majZGA, late pre-majZGA, and majZGA). Chromatin state transitions were correlated with the emergence of 3D genome structures. RNAPII clustering peaked at NC12, while H3K36me3 clustering peaked at NC14, indicating dynamic structural changes over short developmental times. Loops at NC12 which disappeared by NC14 transitioned from "promoter-like" and "pioneer factor" states at NC12 to "transcriptional" states by NC14. Strengthened TAD boundaries at NC14 gained transcriptional states nearby. Machine learning models were then trained to identify motifs at structural features like loop anchors and boundaries. In silico mutagenesis revealed sequence disruptions predicted to induce local structural remodeling, including at motifs bound by transcription factors (TFs), some of which do not have a characterized role as chromatin structural regulators. Regions with high structural impact scores were associated with transcriptionally active, promoter-like states. In silico mutagenesis of

individual TF motifs was not predicted to disrupt chromatin structure, whereas combined in silico mutagenesis of several TF motifs was predicted to reduce insulation. Predicted redundancy in TFs shaping chromatin folding was suggested to arise from compensatory binding of TFs in the absence of pioneer factors like Zld and GAF, as shown by ATAC-seq footprinting in Zld- or GAF-depleted embryos.

The study's strengths include the very high-quality Micro-C maps generated on precisely staged embryos during ZGA and the integration of these maps with hundreds of published chromatin profiling data, revealing correlations between 3D genome folding and chromatin state. However, despite the detailed insights into chromatin organization during ZGA, only an incremental conceptual advance was arguably achieved. Predicted novel genome folding factors were not experimentally validated, likely due to their predicted redundant functions and minimal individual contributions to genome folding, as also suggested by previous studies showing limited effects of knocking down pioneer factors like Zld or GAF.

Below I list some remaining questions that could potentially be further explored, together with minor comments.

Major:

1. Boundaries: Are TAD boundaries observed in the Pico-C datasets only explained by transcriptional activity, or is there evidence for a contribution to insulation by architectural proteins such as insulator proteins?
2. Loops: Can the authors speculate further on the molecular mechanisms underlying chromosomal loop formation, and how diverse these mechanisms are for potentially different classes of observed loops? Are the dynamics of loop appearance/disappearance over development explained by similar or distinct mechanisms?
3. Compartments: How does heterochromatin maturation impact genome folding?

Minor:

4. I did not find the description of the ChromHMM results on page 6 well integrated with the rest of the results section. The results in this section seemed to be mostly described as validating published data.
5. Fig. 5B: It is unclear which peaks underlie the loop anchors observed in Fig. 5A.
6. How were Zld- or GAF-depleted embryos generated? It was not clear whether all of the data shown in these mutants was published or generated for this study.
7. The scale bars showing negative values on the right are a little confusing (decreasing values from left to right).
8. I did not find that the Discussion was clearly structured or written, see examples below:
 - a. Lines 382-389: It may be helpful to describe the observation that Zld depletion does not affect TAD boundary insulation in the Results section, because it seemed to only be mentioned for the first time in the Discussion (or I may have missed this).
 - b. Lines 396-408: I was not able to follow the logical argumentation in this entire paragraph.

Reviewer #2 (Remarks to the Author):

Review of NG-A66634.

In this manuscript, Maziak and coworkers study the 3D chromatin reorganization process during the early stages of *Drosophila* embryo development, in the window spanning from the minor to the major zygotic genome activation (ZGA). They first set up a method which they dub Pico-C, which is a low cell micro-C. Applying this method to nuclear cycles 9, 10, 11, 12, 13 and NC14, they detect an extensive set of changes at all levels of chromatin organization, including changes in chromatin looping that are detected with high resolution. They then perform extensive meta-analysis of omics data collected on wt embryos but also in embryos depleted of pioneer TFs ZeldA (Zld) and GAF. They classify the genomic chromatin in 17 states that can be grouped into 7 categories and find that chromatin states gradually evolve and complexify between the minor and major ZGAs. They find that active promoter states associate with chromatin boundary formation and they identify a relatively large group of TFs that seem to cooperate with Zld and GAF to set up chromatin architectural landmarks, which they suggest to explain why little chromatin accessibility and architectural changes are detected in Zld or GAF-depleted embryos. They suggest that this might be part of an evolutionary conserved mechanism supporting genome organization resilience.

The work presents a valuable 3D genome folding dataset that is complemented by a state-of-the-art computational analysis. I have several points that should be considered:

- Line 69: the authors cite Cavalheiro et al to discuss redundancy of boundary proteins in flies, but they should add a citation to Kaushal et al., PMID: 35559678, which corroborates the conclusion and was published before the Cavalheiro paper.

- Concerning the Pico-C, one well-known bias in the published micro-C data is that the genomic sequencing coverage is not homogeneous and that regions corresponding to different chromatin states correspond to different sequencing depth. This bias probably depend on technical details such as degree of MNase digestion and method of library preparation, which might or might not use DNA fragment sizing and how tight is sizing. If coverage has a broad range, this can potentially skew the data, even after applying the normalization tools used by the authors. To check for this, the authors could just show a genome coverage track above chromosome regions of different sizes (whole chromosome arm, 5 Mb, 1 Mb). Furthermore, they could use ChromHMM to produce violin plots to show the distribution of sequencing depth in genome segment populations included in each of the 17 states.

- Concerning the Pico-C results, see lines 124-130 and Fig 1f. The figure panel shows aggregate loop plots at different distances during development. It is awkward that loops at distances >1Mb, even more at >5Mb, do not show a focal point in the plot center but instead many little points scattered around the center. How many loops were scored in these plots? Please indicate this at the bottom of each aggregate plot. How do the authors explain this?

- as for the detection of pre-major ZGA loops, loops between active chromatin regions were previously identified at NC 9-13 by Ogiyama et al, 2018, this should be cited.

- concerning Figure 2, S2 and Tables S2; S3, the authors segment the genome into 17 states, with several of them being related to each other and being stronger or weaker versions of a main state. They see that there is a significant state dynamics, with a general tendency toward complexification and addition of states related to transcription, which makes sense since 3 of the 4 developmental times were taken prior to the major ZGA. Nevertheless, there are a couple of issues to be discussed. First, several of the measured marks/protein binding profiles are not available at the pre-minor ZGA state. This could skew the HMM, even if the authors performed the segmentation using all data. How can one take this into account in order to make sure that the modeling is not missing early states? Furthermore, did the authors revisit the data quality of the various time points? Very early profiles suffer from the low cell numbers and the possibility of contamination from later embryos, this can also potentially affect the segmentation. Finally, there is K27me3 but no emission as a separate state. Even if there is little K27me3 at very early points, Lovino and coworkers did show that K27me3 coats some of the Polycomb domains, so one emission different from HP1 or transcribing TSS 2 (which are actually a bit surprising and deserve analysis, since K27me3 is not known to associate to any of the two) would be expected.

- Lines 203-204: provide a more detailed definition and explanation of high and low global Moran's I, this is not understandable to the non-specialists.

- One issue, before the deep dive in data analysis and modelling, is to test what the detected transient loops correspond to, when confronted to an orthogonal method of proximity analysis. One possibility would be to perform a DNA FISH experiment with 2 or 3 probes, in order to test for differences in clustering of the signals. Regardless of the result, this experiment would also give an idea of what actual chromatin distances correspond to, when compared to Gaudi plots of the same regions.

- At line 265, the authors suggest that Orca predicts boundary reduction upon Zld site mutation at the Bitesize locus, which would validate model performance since Zld depletion in embryos also reduces the boundary at the locus, as shown by Hug et al in 2017. If they refer to a data shown in Fig 7B of that paper, the boundary reduction is much stronger, two neighbouring TADs merge in the experimental result, whereas in Fig S4 one sees only a marginal reduction of the inter TAD boundary. If it was another data from Hug that one should compare to, please show the experimental map in order to enable readers to compare model results to experimental ones.

- In Figs 4 and S4, the authors train a model to find motifs and proteins that might affect structure maximally at NC14. They find that the M1BP motifs have the strongest effect. Did their model identify sequence motifs that are not targets of known TFs, potentially suggesting the existence of novel structural or regulatory proteins?

- Still concerning this, one experimental suggestion would be to show whether mutation or downregulation of M1BP does decrease boundaries, since the effect seems to be much stronger than the boundary proteins that have been tested so far (4-fold stronger than CTCF for example).

- The authors use TOBIAS to try to identify differences in footprinting from ATAC-seq data in wt versus mutant fly embryos. Their analysis suggests that loss of Zld or of GAF is associated with deeper footprints for other factors. The authors conclude that this compensation in binding by other factors contribute to architectural resilience of chromatin. On the technical side, the increased GAF footprint in Zlf- condition that is shown in Fig 5d does not seem to be very convincing, same for the Clamp footprint in GAF- condition shown in S5a. Some of the other changes also look rather small. What statistical testing was used to detect significance? Please comment.

- Furthermore, the TOBIAS evidence is interesting, but it is important to provide at least one experimental confirmation, depleting Zld and showing that binding of at least one of the up factors corresponds to increased ChIP-seq or CUT&RUN binding at the Zld sites.

Reviewer #3 (Remarks to the Author):

Changes in genome organization coincide with changes in chromatin state and transcription, but despite immense efforts, assigning cause-consequence relationships has remained challenging. Here, Maziak and colleagues refine the Micro-C assay to describe genome reorganization over the course of Zygotic Genome Organization in the fly embryo, followed by machine learning-based segmentation (using a large set of available genomics data on chromatin structure and transcription) and in-silico perturbation models to pinpoint instructive factors and elements.

Overall, this is an interesting study that combines a valuable modification to the Micro-C protocol ("pico-C") to generate 3D maps at impressive resolution with the discovery of important leads about factors that have direct or indirect structuring functions in this process. My main reservation is that the large majority of the results are of a correlative nature, and do not distinguish between chromatin state and downstream changes in transcription (which, I think, should be considered separately). Considering the increasingly established impact of transcription on genome organization, further functional

validation will improve our understanding of underlying mechanisms and may help to discriminate where 3D genome organization is instructive and where it's a consequence of changes in transcriptional activity.

Major comments:

1. Whereas I consider the Pico-C assay an elegant refinement of the Micro-C protocol, it will benefit from additional description and benchmarking (beyond the cis-trans and cis-local values provided in Table S1). Benchmarking: considering that high quality conventional Micro-C data from NC14 is available (Batut et al, 2022), how do their own data compare in resolution, sequencing depth (including useful read pairs), number of required library preparations (from what I gather in Table S1, quite many for most stages?), etc. Regarding detail, from the Material and Methods section, I gather that Pico-C requires around 60,000 nuclei. Providing this number in the main text, rather than the undefined "low-input" description, will be helpful.

2. The core of the manuscript relies on a staged segmentation of the genome, using the large number of datasets that are provided in Table S3. Not all data is available for all timepoints (or for the Zld- condition), which will introduce biases in the emitted ChromHMM states and stage specific transitions. How do the authors think this influences the outcomes? They should discuss the potential impact of this limitation (from line 188?) and consider, if essential, to generate missing data sets.

3. The staged reorganization of the HDAC1 locus, as elegantly analyzed and visualized in Fig. 3, suggests a dynamic and coordinated multi-component process. Yet, the correlative nature of the investigations precludes the identification of causative events (e.g. line 209). The transition of the pre-major ZGA to major ZGA appears associated with a major reorganization of transcription at this locus, which raises the important question if 3D genome reorganization (mediated by chromatin states?) instructs this transcriptional reorganization, or rather if changes in genome organization are a consequence. Combined with the emerging importance of Paused RNA PolII in genome organization (cited Barshad et al, 2023, also Ghavi-Helm et al, 2014), the understanding of mechanisms (this figure, and possibly the remainder of the study) will strongly gain in impact if the authors can add insights on Paused RNA PolII distribution (e.g. Ser5-ChIP-seq) or ongoing transcription (e.g. PRO-seq) and ideally Pico-C data where transcription (or initiation to elongation) is inhibited at certain stages.

In a similar vein, the identification of robustness to pioneer factor loss (Fig. 5) is an interesting result, but can't be uncoupled from a (potential) drastic effect on either global ZGA dynamics or more gene-specific transcription changes. Could the identified compensatory factors modulate transcriptional output in a more subtle manner than picked up by the used segmentation approach? Could the authors investigate this using the available RNA-seq in the Zld- background? Moreover, this result raises the question how Zld and GAF can influence binding of other factors. Do these footprints localize close to Zld and GAF binding sites (if I understand correctly, they don't overlap)? Could a potential distance-dependent effect be extracted from the available data?

Minor remarks:

- The addition of arrowheads in the contact maps in Fig. 1C and 3A,C will help to highlight relevant structures. I'm not familiar with "tethering elements" (Fig. 1C; Antp gene). This will benefit from additional detail in the text (line 112).
- Line 115: it has remained controversial if 3C-based assays can describe chromatin compaction. Better to rephrase.
- Line 122: reduced at NC14, not lost?
- Line 140: why did the authors decide on 20 states? Was this rationalized?
- Line 203: a statistical analysis based on the Global Moran I is not conventionally used. It will benefit from some additional detail in the text. From what I gather, it's not used to quantify the degree of spatial clustering, but it is the measure itself.
- From line 223 and Fig. 3B,D: I'm not sure if I understand correctly: are these the Global Moran I values that were determined genome wide? Is it relevant to interpret these values for differences at the single locus?
- Line 256: chromatin and transcriptional landscape
- Fig. 1D: are these stage-specific boundaries, or the same set of boundaries for all time points? Would it be insightful to add aggregates for lost/reduced boundaries as well?
- Fig. 1F: add information on the number of loops in each distance category, and possibly distribution of sizes within each category?
- Fig. 4D: could the authors add the size of the genomic interval that is visualized (is this 20kb resolution in the figure as well)?
- Fig. 5D and S5: what does "position" mean in these graphs? Are these pile-ups of footprints around all GAF and BEAF-32 peaks, as present in untreated cells? These panels will benefit from additional detail.

Version 1:

Decision Letter:

27th Aug 2025

Dear Professor Vaquerizas,

I apologize for the delay in sending you the decision, which is due to my recent absence from the office. Thank you for your patience during this process.

Your Article, "Extensive 3D Genome Reorganization Foreshadows Zygotic Genome Activation in *Drosophila*" has now been seen by 3 referees. You will see from their comments below that reviewers are still supportive and find your work of interest, however there are a few remaining points that should be addressed. We are interested in the possibility of publishing your study in Nature Genetics, but would like to consider your response to these concerns in the form of a revised manuscript

before we make a final decision on publication.

To guide the scope of the revisions, the editors discuss the referee reports in detail within the team, including with the chief editor, with a view to identifying key priorities that should be addressed in revision and sometimes overruling referee requests that are deemed beyond the scope of the current study. In this case, we please ask you to perform the last experiment required by Reviewer #2 in order to disentangle the effect of transcription from that of TFs and address all the other minor points raised by Reviewer #1 and #3. We hope that you will find the prioritized set of referee points to be useful when revising your study. Please do not hesitate to get in touch if you would like to discuss these issues further.

We therefore invite you to revise your manuscript taking into account all reviewer and editor comments. Please highlight all changes in the manuscript text file. At this stage we will need you to upload a copy of the manuscript in MS Word .docx or similar editable format.

*2) If you have not done so already please begin to revise your manuscript so that it conforms to our Article format instructions, available

[here](http://www.nature.com/ng/authors/article_types/index.html).

*3) Include a revised version of any required Reporting Summary: <https://www.nature.com/documents/nr-reporting-summary.pdf>

Please be aware of our [guidelines](https://www.nature.com/nature-research/editorial-policies/image-integrity) on digital image standards.

EXTENDED DATA FIGURES

Link Redacted

We hope to receive your revised manuscript within four to eight weeks. If you cannot send it within this time, please let us know.

Nature Genetics is committed to improving transparency in authorship. As part of our efforts in this direction, we are now requesting that all authors identified as 'corresponding author' on published papers create and link their Open Researcher and Contributor Identifier (ORCID) with their account on the Manuscript Tracking System (MTS), prior to acceptance. ORCID helps the scientific community achieve unambiguous attribution of all scholarly contributions. You can create and link your ORCID from the home page of the MTS by clicking on 'Modify my Springer Nature account'. For more information please visit please visit www.springernature.com/orcid.

Best,
Chiara

Chiara Anania, PhD
Associate Editor
Nature Genetics
<https://orcid.org/0000-0003-1549-4157>

Reviewers' Comments:

Reviewer #1 (Remarks to the Author):

The authors majorly revised the manuscript by updating or clarifying analyses and incorporating two new major Pico-C (and accompanying RNA-seq or ATAC-seq) datasets to describe effects of transcription elongation inhibition or of double depletion of pioneer factors. The data are of high quality. The authors' conclusions paint a picture of early genome folding being surprisingly dynamic over cleavage cycles, and suggesting – and partially demonstrating through perturbation experiments – that it is shaped by diverse parallel (and in some cases synergistic) mechanisms. The value of the work therefore lies in the clarity of the genome folding maps in many successive interphases of early embryo cleavage cycles which enabled a new appreciation of the dynamics and diversity of loci involved in 3D genome folding – although other mechanisms driving folding remain to be clarified.

Below are remaining minor comments.

1. Line 72: This sentence does not capture the conclusion of the referenced papers accurately. Depletion of Cp190 weakens or abolishes almost one quarter of all TAD boundaries, which seems contradictory with the statement that “removal of insulator proteins shows no genome-wide changes”.
2. Line 140: Clarifying that only a subset (not all) meta-loops are captured in early embryos can avoid confusion with the fact that most meta-loops only form in differentiated neurons.
3. Line 348: The use of “remarkably” was not fully clear. It was arguably expected to observe architectural disruptions at GAF-associated features in the double knock-down as these were previously reported (Li et al. 2023), though Zld-dependent folding defects are novel.
4. Fig. 3A legend: Replace second “left” by e.g. “center”.
5. Fig. 5A: Could the authors comment on the somewhat unexpected fact that pioneer states are more enriched at loop anchors or boundaries after Zld depletion.
6. Fig. 5E: “GAF-associated”

Reviewer #2 (Remarks to the Author):

Maziak et al have extensively revised their manuscript, which is an important contribution that should ultimately be published.

I have one last point, related to comment 3 raised by reviewer #3, concerning experiments required to disentangle the effects of transcription from that of specific TFs. The authors have partly addressed this by performing transcription elongation inhibition by alpha-amanitin and then performing Pico-C. I think this could be complemented by inhibiting initiation using triptolide and providing Ser5-ChIP-seq or PRO-seq and Pico-C with or without inhibition. This would be very interesting and important irrespective of the results that will be obtained.

Reviewer #3 (Remarks to the Author):

In their revised manuscript, Maziak and colleagues have taken considerable efforts to address the comments from the reviewers. I particularly appreciate the results from the Zelda/GAF depletion and a-Amanitin experiments, which provide important insights into cause/consequence of gene regulation/TF binding and changes in transcription. Except for a few minor remarks, I now consider the manuscript a good fit for publication in Nature Genetics.

Minor remarks:

- The Chromosight analysis of boundaries and loops (Fig. 3h, 5f and associated supplementary panels) provides valuable insight into the extent of chromatin reorganization. Considering that the analysis is pairwise, between control and treated conditions, it would be more informative to use a visualization that retains this information (e.g. something like this: <https://stackoverflow.com/questions/49370705/implementing-paired-lines-into-boxplot-ggplot2>)
- As mentioned, the addition of Zelda/GAF depletion and a-Amanitin studies provides important insights into the underlying mechanisms. I'm intrigued to see how much stronger the impact of the Zelda and GAF pioneer factors are, as compared to transcription itself. This difference is briefly mentioned in the abstract and introduction. I think it merits attention, in more detail, in the discussion section as well.
- The references to Supplementary figure 1 are not completely accurate? Line 116: Fig. S1c-d? Line 125: Fig. S1e-f?

Version 2:

Decision Letter:

Our ref: NG-A66634R1

16th Sep 2025

Dear Dr. Vaquerizas,

Thank you for submitting your revised manuscript "Extensive 3D Genome Reorganization Foreshadows Zygotic Genome Activation in *Drosophila*" (NG-A66634R1). We find that the paper has improved in revision, and therefore we'll be happy in principle to publish it in Nature Genetics, pending minor revisions to satisfy the referees' final requests and to comply with our editorial and formatting guidelines.

Congratulations!

Sincerely,
Chiara

Chiara Anania, PhD
Associate Editor
Nature Genetics
<https://orcid.org/0000-0003-1549-4157>

Extensive 3D Genome Reorganization Foreshadows Zygotic Genome Activation in *Drosophila*

Noura Maziak^{1,2}, Yuchen Zhang³, Fabian Groll^{1,2}, Haley E. Brown⁴, Alla Madich⁵, Yadwinder Kaur⁴, Melissa M. Harrison⁴, Jian Zhou³, Juan M. Vaquerizas^{1,2,*}

¹MRC Laboratory of Medical Sciences, London, W12 0HS, UK

²Institute of Clinical Sciences, Faculty of Medicine, Imperial College London, London, W12 0HS, UK

³Lyda Hill Department of Bioinformatics, University of Texas Southwestern, Dallas TX 75390, USA

⁴Department of Biomolecular Chemistry, University of Wisconsin-Madison, Madison, WI 53706, USA

⁵Department of Genetics, University of Cambridge, Downing Site, Cambridge, CB2 3EH, UK

* Correspondence to: J.M.V. (j.vaquerizas@lms.mrc.ac.uk)

Response to Reviewers

We would like to thank our three reviewers for their insightful comments and constructive criticism. Their feedback has been valuable in improving our manuscript. In response, we have revised the text for clarity and incorporated substantial new analyses and datasets. These new data show that genome establishment is governed by a network of intersecting but distinct regulatory inputs.

Broadly, we have added the following new analyses and datasets:

1. A thorough technical analysis of the Pico-C data including coverage tracks at multiple zoom levels and in comparison to publicly available datasets of both Micro-C and Hi-C experiments. This analysis underscored the uniform coverage, and increased granularity of our Pico-C data, and the alignment of our results against existing reference datasets.
2. A systematic cluster analysis of boundaries and loop anchors, revealing a diverse chromatin landscape with distinct enrichments underlying these features. These analyses provided critical insight into the effects of subsequent experimental perturbations applied to our system.
3. A detailed compartment analysis of the chromatin contact data. This identified that sub-compartments A.1 and B.2 form most compartment-compartment interactions. Remarkably, we capture an uncoupled dynamic where A.1-A.1 interactions precede B.2 and show enrichment of H4K16ac preceding ZGA.
4. Pharmacological inhibition of active transcription on embryos. We conducted RNA-seq and Pico-C on embryos injected with α -amanitin. In agreement with our previous findings (Hug et al. 2017), we show that despite global inhibition of transcriptional elongation, overall, there are modest architectural changes. However, the increased granularity of the Pico-C data allowed us to capture loop persistence upon transcriptional inhibition. In addition, by integrating the new domain boundary classification analysis,

we now show that boundary weakening, albeit modest, is significant at sites within an H3K4me3-enriched boundary clusters.

5. To further dissect the contributions of distinct regulators to genome architecture, we performed simultaneous depletion of both Zelda and GAF and conducted Pico-C and ATAC-seq in NC14 embryos. This resulted in a pronounced loss of chromatin architecture at regions regulated by either or both factors, indicating that parallel inputs mediate genome establishment. To the best of our knowledge, this is the first experiment measuring chromatin conformation changes performed with the simultaneous loss of pioneer factors in *Drosophila*.

Below, we provide a point-by-point response to the reviewers' comments. For visual ease, we have marked reviewer comments in blue italic typography while leaving our replies in black.

Reviewer Comments:

Reviewer #1:

(Remarks to the Author):

This study develops Pico-C, an improved low-input Micro-C method, to generate very high-resolution chromatin interaction maps of early fly embryos during successive interphases of nuclear cleavage cycles (NC9 to NC14), covering the minor and major waves of zygotic genome activation (ZGA). The authors identified increasing chromatin loop sizes as development progresses. They re-analyzed hundreds of published chromatin and RNA-profiling datasets using ChromHMM to categorize loci into 20 chromatin states across various stages (pre-minZGA, early pre-majZGA, late pre-majZGA, and majZGA). Chromatin state transitions were correlated with the emergence of 3D genome structures. RNAPII clustering peaked at NC12, while H3K36me3 clustering peaked at NC14, indicating dynamic structural changes over short developmental times. Loops at NC12 which disappeared by NC14 transitioned from “promoter-like” and “pioneer factor” states at NC12 to “transcriptional” states by NC14. Strengthened TAD boundaries at NC14 gained transcriptional states nearby.

Machine learning models were then trained to identify motifs at structural features like loop anchors and boundaries. In silico mutagenesis revealed sequence disruptions predicted to induce local structural remodeling, including at motifs bound by transcription factors (TFs), some of which do not have a characterized role as chromatin structural regulators. Regions with high structural impact scores were associated with transcriptionally active, promoter-like states. In silico mutagenesis of individual TF motifs was not predicted to disrupt chromatin structure, whereas combined in silico mutagenesis of several TF motifs was predicted to reduce insulation. Predicted redundancy in TFs shaping chromatin folding was suggested to arise from compensatory binding of TFs in the absence of pioneer factors like Zld and GAF, as shown by ATAC-seq footprinting in Zld- or GAF-depleted embryos.

The study’s strengths include the very high-quality Micro-C maps generated on precisely staged embryos during ZGA and the integration of these maps with hundreds of published chromatin profiling data, revealing correlations between 3D genome folding and chromatin state. However, despite the detailed insights into chromatin organization during ZGA, only an incremental conceptual advance was arguably achieved. Predicted novel genome folding factors were not experimentally validated, likely due to their predicted redundant functions and minimal individual contributions to genome folding, as also suggested by previous studies showing limited effects of knocking down pioneer factors like Zld or GAF.

We thank the reviewer for their constructive feedback. We appreciate the recognition of the strengths of our study, particularly the high-resolution Pico-C maps and the integrative analysis across chromatin and transcriptional datasets. In response to the concern regarding conceptual novelty and the lack of functional validation for predicted genome folding factors, we have incorporated new functional experiments to directly address these issues.

Specifically, we now include Pico-C data from α -amanitin-injected embryos to assess the impact of transcription inhibition on chromatin organization. Additionally, we performed simultaneous depletion of Zelda (Zld) and GAF—two key pioneer factors—and generated new Pico-C datasets under these conditions. These new data, together with updated analyses, demonstrate that genome architecture in *Drosophila* is not established via simple redundancy, but instead arises from parallel pathways mediated by distinct sets of factors. Notably, the more granular effects

observed in the double-knockdown condition are largely restricted to regions normally influenced by these specific factors, underscoring the modular and localized nature of genome folding regulation.

We also wish to emphasize that our data themselves represent a significant new technical and conceptual advance to the field. To our knowledge, there are no existing chromatin conformation maps with the temporal resolution we present here, capturing the early dynamics of genome organization during the minor wave of zygotic genome activation in any system. These high-resolution, time-resolved maps provide new opportunities to interrogate the developmental emergence of genome architecture, and we believe they will be a valuable resource for future mechanistic and comparative studies.

Major:

1. Boundaries: Are TAD boundaries observed in the Pico-C datasets only explained by transcriptional activity, or is there evidence for a contribution to insulation by architectural proteins such as insulator proteins?

We thank the reviewer for this insightful question. We agree that transcriptional activity alone is unlikely to fully explain the formation of TAD boundaries, even though RNA Polymerase II (Pol II) is frequently enriched at these sites.

To explore this further, we performed unsupervised clustering of boundary calls and loop anchors at NC14, as now shown in Figure 2a–d and Figure S3. This analysis revealed that while Pol II is broadly enriched at boundaries, the chromatin landscapes of these features are heterogeneous, suggesting the involvement of additional regulatory mechanisms.

To functionally dissect the contributions of transcription and architectural proteins, we leveraged two new experimental perturbations. First, we inhibited transcriptional elongation using α -amanitin. Despite a global reduction in transcription (Fig. 3f; Fig. S7b,c), boundary insulation was only modestly affected, and this effect was statistically significant only in clusters enriched for H3K4me3 (Fig. S7e). These findings are consistent with our previous observations (Hug et al. 2017) and suggest that transcriptional elongation is not the primary driver of boundary formation. We also note that a recent preprint (Oak et al. 2025) reported loss of H3K4me3 upon α -amanitin treatment in xenopus embryos, supporting the idea that certain loci are more susceptible to transcriptional perturbation.

Second, we performed co-depletion of the pioneer factors Zld and GAF, which led to more pronounced, region-specific disruptions in boundary insulation—particularly at clusters associated with these factors. These results indicate that architectural proteins, including Zld and GAF, contribute to boundary formation in a modular and context-dependent manner.

Together, these findings support a model in which transcriptional activity is associated with, but not sufficient for, boundary formation. Instead, boundary insulation appears to be shaped by a combination of transcription-independent mechanisms and the action of specific architectural proteins.

2. Loops: Can the authors speculate further on the molecular mechanisms underlying chromosomal loop formation, and how diverse these mechanisms are for potentially different classes of observed loops? Are the dynamics of loop appearance/disappearance over development explained by similar or distinct mechanisms?

The reviewer raises an important point. While our study does not directly dissect the molecular mechanisms of loop formation, our data provide several insights that allow us to speculate on the diversity and dynamics of these mechanisms during early embryogenesis.

First, our clustering of loop anchors at NC14 (Fig. 2c–d; Fig. S3d–f) revealed distinct chromatin signatures associated with different classes of loops. Some loop anchors were enriched for pioneer factors such as Zld, GAF, and CLAMP, and could be further sub-clustered into GAF-high/Zld-low, Zld-specific, and Zld–GAF co-bound groups. These pioneer-associated clusters showed high levels of transcriptional marks and RNA Polymerase II occupancy. We also identified loop anchor clusters enriched for the transcription factor BEAF-32, which similarly exhibited transcriptional activity. In contrast, clusters associated with classical insulator proteins – including CTCF, CP190, and a distinct group marked by high Su(Hw) signal – showed little to no enrichment for transcriptional marks, suggesting a more structural rather than transcription-linked role at these sites. This suggests that loop formation is not governed by a single mechanism, but rather by a combination of transcription-associated and transcription-independent processes.

Second, our perturbation experiments support this mechanistic diversity. Inhibition of transcriptional elongation using α -amanitin had no significant effects on loop formation, and primarily affected loops associated with transcriptional marks (Fig. 3h, Fig. S7e). In contrast, co-depletion of Zld and GAF led to more pronounced and region-specific disruptions in loop architecture, particularly at sites enriched for these factors (Fig. 5f, Fig S10c). These findings suggest that many loops depend on the binding of specific transcription factors rather than transcriptional activity.

Regarding developmental dynamics, we observed that certain loops appear transiently—for example, loops present at NC12 but absent by NC14. These dynamic loops often correspond to regions transitioning from “promoter-like” or “pioneer factor” states to “transcriptional” states (Fig. 3a–d), suggesting that loop dynamics may reflect underlying changes in chromatin state and regulatory factor occupancy. Interestingly, we also find that some of these transient looping events, upon abrogation of transcriptional elongation with α -amanitin, are retained at NC14 (Fig. 3g; Fig. S7d). These findings suggest that early loops are shaped by a variety of regulatory inputs, with RNA Polymerase II dynamics playing a contributory role. The broader regulatory context of the surrounding genomic neighborhood potentially also influences their behavior.

Together, these observations support a model in which chromatin loops arise through multiple, temporally regulated mechanisms involving both transcriptional and architectural inputs. We have added a brief discussion of this point in the revised manuscript (Discussion, p. 10–11).

3. Compartments: How does heterochromatin maturation impact genome folding?

We thank the reviewer for this important question. To address the role of heterochromatin maturation in genome folding, we have now included a compartment analysis, shown in Figure 2g–i and Supplemental Figure 5a–c. This analysis focused on NC14, the stage where compartmentalization is most prominent. To optimize signal detection in the compartment analysis, we tested multiple resolutions ranging from 50 kb to 1 kb. Based on correlation with H3K36me3, we identified 3 kb as the optimal resolution for capturing compartmental features (see Methods for details, p. 17–18). To ensure robust and interpretable results, we excluded centromeric regions due to poor mappability.

This approach allowed us to resolve not only canonical A and B compartments but also finer sub-compartments, including A.1 through B.2. Notably, A.1 regions, associated with highly active chromatin, engage in compartmental interactions earlier than B.2 regions, suggesting a temporal decoupling between A and B compartment dynamics. This is particularly intriguing in light of prior work showing that HP1 depletion has minimal effect on A compartments (Zenk et al. 2021)

Consistent with this, we observed that A.1 regions are enriched for intergenerationally maintained H4K16ac even prior to the major wave of zygotic genome activation. While we did not observe a specific enrichment of HP1 at these regions, this aligns with previous findings that ~46% of HP1 binding sites lie within A compartments and co-localize with active chromatin marks (Zenk et al. 2021). It's also worth noting that HP1 is strongly enriched at centromeric regions – which we excluded due to low mappability and high background signal.

Altogether, these findings suggest that heterochromatin maturation contributes to the emergence of compartmentalization in a temporally and spatially selective manner. Our results highlight a previously underappreciated decoupling between the establishment of active and inactive compartments during early embryogenesis, offering new insight into the layered organization of the genome during zygotic genome activation.

Minor:

4. I did not find the description of the ChromHMM results on page 6 well integrated with the rest of the results section. The results in this section seemed to be mostly described as validating published data.

We appreciate the reviewer's comment regarding the integration of the ChromHMM results. In response, we have significantly revised both the text and Figure 2 to better integrate the ChromHMM analysis with the rest of the dataset.

Specifically, we have reshaped Figure 2 to present the clustering of architectural features, ChromHMM-based chromatin state annotation, and compartment analysis within a unified framework. This integrated view highlights how these orthogonal approaches, from identifying heterogeneous landscape at architectural features, chromatin state segmentation, and 3D compartment structure, converge to reveal coherent and functionally distinct chromatin domains.

We have also simplified and refocused the descriptive aspect of this section to move beyond validation of prior findings. Instead, the revised narrative emphasizes how our ChromHMM analysis provides a foundational view of the chromatin landscape that informs subsequent functional and structural analyses. For example, we now show that A.1 compartment regions are enriched for genic and H4K16ac-rich chromatin states – a pattern we further confirmed by plotting ChIP-seq signal directly (Fig. 2h; Fig S5b). These observations help establish a conceptual link between chromatin state, transcriptional activity, and 3D genome architecture.

We believe these revisions improve the coherence of the Results section and clarify the role of ChromHMM in contextualizing the dynamic genome folding events described in later sections.

5. Fig. 5B: It is unclear which peaks underlie the loop anchors observed in Fig. 5A.

We appreciate the reviewer pointing this out. In the original version of the manuscript, the loop anchors shown in Fig. 5A corresponded almost one-to-one with regions annotated as “red promoter” chromatin states in the ChromHMM model. However, to accommodate new data from

the Zld/GAF double knockdown experiments, we have since revised this figure and removed the original panel in question.

The updated figure now focuses on the functional consequences of perturbing key architectural factors, and we believe it provides a clearer and more informative view of the mechanisms underlying loop formation. We have ensured that the revised figure panels are more directly interpretable and better integrated with the rest of the manuscript

6. How were Zld- or GAF-depleted embryos generated? It was not clear whether all of the data shown in these mutants was published or generated for this study.

We thank the reviewer for pointing this out. The Zld- and GAF-depleted embryo datasets included in the original manuscript were obtained from previously published work conducted by multiple laboratories (Gaskill et al. 2021; Hannon, Blythe, and Wieschaus 2017; Schulz et al. 2015; Li et al. 2014; Blythe and Wieschaus 2015; Duan et al. 2021). These datasets were not newly generated for this study.

Upon revision, we have generated Pico-C datasets for embryos with simultaneous depletion of both Zld and GAF. To our knowledge, this is the first time that a double-depletion of architectural factors is performed in vivo in *Drosophila*.

We acknowledge that the provenance of these data may not have been sufficiently clear in the original manuscript. We have now revised the relevant sections of the Methods, Results and Figures to explicitly state the source of these datasets and to clearly distinguish between previously published and newly generated data throughout the manuscript.

7. The scale bars showing negative values on the right are a little confusing (decreasing values from left to right).

We appreciate the reviewer catching this! We agree that the orientation of the scale bars was potentially confusing. We have corrected the axis direction in the affected figures so that values now increase from left to right, as expected.

*8. I did not find that the Discussion was clearly structured or written, see examples below:
a. Lines 382-389: It may be helpful to describe the observation that Zld depletion does not affect TAD boundary insulation in the Results section, because it seemed to only be mentioned for the first time in the Discussion (or I may have missed this).*

We thank the reviewer for this helpful suggestion. The observation that Zld depletion does not broadly affect TAD boundary insulation was initially mentioned in the Introduction to provide context for our functional experiments. However, we agree that this finding is more appropriately placed and emphasized within the Results section.

In response, we have revised the manuscript to more clearly present this observation in the Results, where it is now directly linked to our new functional data. We have also updated the Discussion to better integrate and contextualize this point in light of the new double-depletion experiments. We hope these changes improve the clarity and logical flow of the manuscript.

b. Lines 396-408: I was not able to follow the logical argumentation in this entire paragraph.

We thank the reviewer for this helpful feedback. We agree that the paragraph in lines 396–408 lacked clarity and did not effectively convey the intended argument. In the revised manuscript,

we have substantially restructured the Discussion section to improve its logical flow and to better integrate the new results, including those from the Zelda/GAF double-depletion experiments.

The revised paragraph now more clearly articulates the relationship between chromatin state transitions, transcriptional activity, and 3D genome architecture, and avoids speculative or ambiguous phrasing. We hope these changes make the discussion more accessible and coherent for the reader.

Reviewer #2:

(Remarks to the Author)

Review of NG-A66634.

In this manuscript, Maziak and coworkers study the 3D chromatin reorganization process during the early stages of Drosophila embryo development, in the window spanning from the minor to the major zygotic genome activation (ZGA). They first set up a method which they dub Pico-C, which is a low cell micro-C. Applying this method to nuclear cycles 9, 10, 11, 12, 13 and NC14, they detect an extensive set of changes at all levels of chromatin organization, including changes in chromatin looping that are detected with high resolution. They then perform extensive meta-analysis of omics data collected on wt embryos but also in embryos depleted of pioneer TFs Zelda (Zld) and GAF. They classify the genomic chromatin in 17 states that can be grouped into 7 categories and find that chromatin states gradually evolve and complexify between the minor and major ZGAs. They find that active promoter states associate with chromatin boundary formation and they identify a relatively large group of TFs that seem to cooperate with Zld and GAF to set up chromatin architectural landmarks, which they suggest to explain why little chromatin accessibility and architectural changes are detected in Zld or GAF-depleted embryos. They suggest that this might be part of an evolutionary conserved mechanism supporting genome organization resilience.

The work presents a valuable 3D genome folding dataset that is complemented by a state-of-the-art computational analysis. I have several points that should be considered:

We thank the reviewer for their positive assessment of our work and for recognizing the value of the 3D genome folding dataset we generated, as well as the computational framework we developed to analyze it. We appreciate the reviewer's thoughtful summary of our study and the constructive feedback that follows. Below, we address each of the reviewer's points in detail.

- Line 69: the authors cite Cavalheiro et al to discuss redundancy of boundary proteins in flies, but they should add a citation to Kaushal et al., PMID: 35559678, which corroborates the conclusion and was published before the Cavalheiro paper.

We thank the reviewer for this helpful suggestion. We agree that the study by Kaushal et al. (PMID: 35559678) provides important supporting evidence for the redundancy of boundary proteins in Drosophila. We have now added this reference alongside Cavalheiro et al. in the revised manuscript.

- Concerning the Pico-C, one well-known bias in the published micro-C data is that the genomic sequencing coverage is not homogeneous and that regions corresponding to different chromatin states correspond to different sequencing depth. This bias probably depend on technical details such as degree of MNase digestion and method of library preparation, which might or might not use DNA fragment sizing and how tight is sizing. If coverage has a broad range, this can potentially skew the data, even after applying the normalization tools used by the authors. To check for this, the authors could just show a genome coverage track above chromosome regions of different sizes (whole chromosome arm, 5 Mb, 1 Mb). Furthermore, they could use ChromHMM to produce violin plots to show the distribution of sequencing depth in genome segment populations included in each of the 17 states.

We thank the reviewer for raising this important point regarding potential biases in sequencing coverage. We have now addressed this directly in the revised manuscript.

As shown in Figure S1d-f, the Pico-C coverage across genomic regions is relatively uniform across developmental stages, even when compared to published Hi-C and Micro-C datasets. To further assess coverage uniformity, we generated genome coverage tracks at multiple scales (whole chromosome arm, ~3 Mb, and ~300 kb windows), using both equal sub-sampled data and full-depth datasets. These analyses revealed no significant differences, ruling out overplotting artifacts.

To evaluate whether chromatin state influences coverage, we compared Pico-C signal to matched or closely matched ATAC-seq profiles and found no obvious correlation between chromatin accessibility and Pico-C coverage (Fig. S1d-f). Additionally, we plotted sequencing depth across the 17 chromatin states for both Hi-C and Micro-C datasets, as suggested by the reviewer (Fig. S4e). These plots show generally uniform coverage across states, with the exception of HP1-associated states, which are enriched at centromeric regions (Fig. S4a). We attribute this deviation to reduced mappability in these repetitive regions.

Finally, we acknowledge that MNase digestion can introduce biases. To mitigate this, we applied a carefully optimized and consistent digestion protocol across all samples, aiming to avoid both over-digestion (which could deplete signal in open chromatin) and under-digestion (which could bias toward accessible regions). This controlled approach minimizes digestion-related artifacts to the best of our ability.

- Concerning the Pico-C results, see lines 124-130 and Fig If. The figure panel shows aggregate loop plots at different distances during development. It is awkward that loops at distances >1Mb, even more at >5Mb, do not show a focal point in the plot center but instead many little points scattered around the center. How many loops were scored in these plots? Please indicate this at the bottom of each aggregate plot. How do the authors explain this?

We thank the reviewer for this observation. In response, we have now added the number of loops (n) scored in each distance bin to the bottom of the corresponding aggregate plots in the revised figure.

Regarding the appearance of the long-range loops (>1 Mb and >5 Mb), we agree that the focal enrichment appears more diffuse or fragmented compared to shorter-range loops. This is likely due to the fixed 1 kb resolution used for all aggregate plots, which we chose to maintain consistency across distance bins. At longer genomic distances, the focal point of looping interactions tends to span broader regions, and the 1 kb resolution may be too fine to capture this signal cohesively – resulting in a more scattered appearance.

We have clarified this point in the revised figure legend to help guide interpretation of these plots.

- as for the detection of pre-major ZGA loops, loops between active chromatin regions were previously identified at NC 9-13 by Ogiyama et al, 2018, this should be cited.

We thank the reviewer for pointing this out. We agree that the study by Ogiyama et al., 2018 is highly relevant and should have been cited. This was indeed an oversight on our part, which we recognized shortly after submission. We have now included the appropriate citation in the revised manuscript.

In addition, we now highlight in Figure S1e that these early loops are clearly captured in our Pico-C data and display dynamic behavior. For instance, at NC10, we observe two prominent long-range loops that form large domain-like structures.

- concerning Figure 2, S2 and Tables S2; S3, the authors segment the genome into 17 states, with several of them being related to each other and being stronger or weaker versions of a main state. They see that there is a significant state dynamics, with a general tendency toward complexification and addition of states related to transcription, which makes sense since 3 of the 4 developmental times were taken prior to the major ZGA. Nevertheless, there are a couple of issues to be discussed.

First, several of the measured marks/protein binding profiles are not available at the pre-minor ZGA state. This could skew the HMM, even if the authors performed the segmentation using all data. How can one take this into account in order to make sure that the modeling is not missing early states?

We thank the reviewer for this insightful comment. We agree that the absence of certain histone marks and transcription factor binding profiles at the earliest developmental stages could, in principle, limit the resolution of chromatin state segmentation at pre-ZGA time points.

Our primary goal with the ChromHMM analysis was to generate an integrated model of chromatin states across early embryogenesis, rather than to define stage-specific segmentations. To assess the potential impact of missing data at early stages, we performed additional segmentations using only the subset of chromatin features available across all time points. These analyses yielded a core set of chromatin states that were highly consistent with those identified in the full model, suggesting that the segmentation is not overly skewed by the missing data at early stages.

Moreover, although our earliest Pico-C time point (NC9) occurs shortly after the onset of the minor ZGA, we find that the pre-minZGA model is still able to identify meaningful regulatory features at this stage. For example, promoter-like and active states are detected at key early loci near the HDAC1 locus (Fig. S6b), supporting the model's ability to capture biologically relevant chromatin features even with limited input data.

Finally, we have consistently validated the segmentation output by cross-referencing with the underlying ChIP-seq signal, which provides additional confidence in the model, particularly at early developmental stages.

Furthermore, did the authors revisit the data quality of the various time points? Very early profiles suffer from the low cell numbers and the possibility of contamination from later embryos, this can also potentially affect the segmentation.

We appreciate the reviewer's concern regarding data quality and potential contamination, particularly at the earliest developmental stages. We have carefully assessed the quality of all ChIP-seq and RNA-seq datasets used in the segmentation. This included manual inspection of coverage profiles, principal component analysis (PCA) to confirm consistency across replicates and time points, and individual comparisons with independent datasets where available. We also performed standard quality control using FastQC and employed matched input controls for ChIP-seq whenever possible.

To minimize the risk of misinterpreting maternally deposited transcripts as early zygotic expression, we used maternal flow-through RNA as input for RNA-seq (see Methods). This approach helps distinguish true zygotic transcriptional activity from maternal contributions.

Regarding staging accuracy and potential contamination, we relied on datasets generated by groups that implemented rigorous embryo collection protocols, including hand-sorting and pre-

collection steps to ensure precise staging. The stage-specific patterns observed in our data further support the temporal resolution of these profiles.

While we acknowledge that very early samples may have lower coverage due to limited cell numbers, we do not believe this introduces systematic bias into the ChromHMM segmentation. The model operates on binarized representations of enrichment over input, which helps buffer against the impact of moderate coverage differences across datasets. Furthermore, we incorporated multiple datasets, including those from independent sources, which helps mitigate batch effects and study-specific biases.

That said, we recognize that ChromHMM segmentations are ultimately models—useful for summarizing chromatin patterns and generating hypotheses, but not definitive representations of chromatin state. We have added a note to this effect in the revised manuscript to clarify the interpretive scope of our analysis.

Finally, there is K27me3 but no emission as a separate state. Even if there is little K27me3 at very early points, Iovino and coworkers did show that K27me3 coats some of the Polycomb domains, so one emission different from HP1 or transcribing TSS 2 (which are actually a bit surprising and deserve analysis, since K27me3 is not known to associate to any of the two) would be expected.

We thank the reviewer for this thoughtful observation. We agree that the presence of H3K27me3 in a transcription-associated state (TSS2) warrants clarification.

While it may seem surprising, similar patterns have been reported previously. For example, one study observed that in early *Drosophila* embryos, both active and repressive chromatin marks—including H3K27me3—can co-occur at highly expressed genes, potentially reflecting a poised or bivalent chromatin state (Munden et al. 2022). In our data, we also observe H3K27me3 signal at transcriptionally active regions, as seen in the ChIP-seq signal from our cluster analysis (Fig. S3b, f; boundary clusters 3–4 and loop cluster 1), and when browsing the genome. Notably, in the data from Iovino et al., H3K27me3 often co-localizes with H3K4me1 (Zenk et al. 2017: see specifically Fig. 4C and Suppl. Fig. 4d–g of that publication), both of which are features of our TSS2 state.

We also note that all datasets used here are from whole-embryo experiments, thus reflecting a composite signal across all cell types at each time point. This means that opposing chromatin states, such as active and repressive marks, may derive from distinct cell populations and not necessarily co-occur in the same cells.

It is also important to note that ChromHMM states are probabilistic. An emission probability of ~0.2 for H3K27me3 in a given state means that only ~20% of the bins in that state show signal for this mark. Thus, the presence of H3K27me3 in TSS2 does not imply uniform co-occurrence but rather reflects a subset of regions where this combination may be biologically meaningful—possibly representing early Polycomb engagement or transitional chromatin states.

Regarding HP1, we appreciate the reviewer’s attention to this point. We were also intrigued by the presence of HP1-associated states near loci such as the *Antp* promoter (see Reviewer Figure 1, panel a below). Indeed, this region shows HP1 enrichment in our data and was also identified as an HP1 peak in Zenk et al., 2021. Interestingly, we also observe H3K27me3 signal at this locus.

This highlights the complexity of chromatin organization at this developmental stage. HP1 is known to be broadly distributed across both A and B compartments in early embryos, with strongest enrichment at centromeric regions. Therefore, overlap between HP1 and other chromatin marks in non-centromeric regions is not unexpected.

As for state 13 (HP1-related), we note that it is relatively rare—covering only ~2% of the genome—and is often found at the boundaries of state 12, which spans ~7%. This suggests that state 13 may represent a transition zone between HP1-enriched domains and adjacent chromatin environments.

Finally, we emphasize that emission probabilities in ChromHMM should not be interpreted as uniform enrichment. For example, although state 13 has an H3K27me3 emission probability of ~0.3, plotting H3K27me3 signal across all state 13 regions reveals overall depletion on average (Reviewer Figure 1, panel b), but clear enrichment at a subset of sites—consistent with the probabilistic nature of the model. We have also repeated this analysis across multiple iterations and consistently observed similar results. Additionally, re-running the model for stages with only datasets available throughout still produced comparable H3K27me3 emission probabilities (Reviewer Figure 1, panel c).

We have now clarified these points in the revised manuscript.

Reviewer Figure 1 | **a** Track view showing ChromHMM states, ChIP-seq data, and RNA-seq near the *Antp* promoter highlights HP1 and H3K27me3 enrichment within an HP1-associated state (marked by yellow lines). **b** Heatmaps of H3K27me3 and HP1 signal at state 13 regions (HP1-associated at ZGA) reveal that while a subset shows H3K27me3 enrichment, the mark is generally depleted across these regions. **c** ChromHMM reanalysis using only available datasets recapitulates similar H3K27me3 emission probabilities.

- Lines 203-204: provide a more detailed definition and explanation of high and low global Moran's *I*, this is not understandable to the non-specialists.

We have now amended this and tried to explain this better in the first parts of the section (p. 7):

“Having created precise Pico-C maps and defined the underlying chromatin states, we wanted to investigate the relationship between them. To do this, we first utilized Metaloci (Kim et al. 2025) which uses spatial autocorrelation (Moran’s I) to examine how strongly a genomic signal correlates with itself across neighboring regions in nuclear space. **Higher Global Moran’s I values indicate that spatially proximal regions exhibit similar signal intensities, while values near zero reflect spatial randomness. Negative values suggest anti-correlation, where nearby regions display opposing signal levels.** This metric enables us to assess how specific chromatin features are spatially organized, and how these relationships evolve across developmental timepoints.”

- One issue, before the deep dive in data analysis and modelling, is to test what the detected transient loops correspond to, when confronted to an orthogonal method of proximity analysis. One possibility would be to perform a DNA FISH experiment with 2 or 3 probes, in order to test for differences in clustering of the signals. Regardless of the result, this experiment would also give an idea of what actual chromatin distances correspond to, when compared to Gaudi plots of the same regions.

We thank the reviewer for this thoughtful suggestion. We agree that orthogonal validation methods such as DNA FISH would provide valuable insight into the physical distances underlying transient loops, particularly in terms of spatial clustering and frequency. However, in the case of loci like HDAC1, the loops we detect occur over relatively short genomic distances, making FISH-based resolution and interpretation particularly challenging in these contexts.

That said, we note that some of the longer-range loops identified in our dataset, especially those highlighted in Supplemental Figure 1, have been independently validated through imaging-based approaches by Ogiyama et al. 2018 and Huang et al. 2021, supporting the robustness of these interactions.

We are also highly interested in understanding the frequency and dynamics of such looping events at single-cell resolution, and we believe this will be an important direction for future studies. We look forward to efforts that build upon our findings by directly comparing population-based and single-molecule or imaging-based views of chromatin architecture.

- At line 265, the authors suggest that Orca predicts boundary reduction upon Zld site mutation at the Bitesize locus, which would validate model performance since Zld depletion in embryos also reduces the boundary at the locus, as shown by Hug et al in 2017. If they refer to a data shown in Fig 7B of that paper, the boundary reduction is much stronger, two neighbouring TADs merge in the experimental result, whereas in Fig S4 one sees only a marginal reduction of the inter TAD boundary. If it was another data from Hug that one should compare to, please show the experimental map in order to enable readers to compare model results to experimental ones.

We thank the reviewer for this important observation. We agree that the boundary reduction observed in Hug et al. 2017 particularly in Fig. 7B of that paper (we include the loci now in Fig. S8a,b of this work as well), is more pronounced than the effect predicted by our model at the Bitesize locus. In our in silico Zld motif mutation, Orca predicts a clear local weakening of insulation at the boundary, but not a full merging of adjacent TADs.

While the magnitude of the predicted effect is smaller, we believe this still represents a meaningful validation of the model. Capturing the correct direction and genomic location of the

boundary change – based solely on DNA sequence and without training on perturbation data – demonstrates the model’s sensitivity to regulatory motif alterations.

To facilitate direct comparison, we have now included the relevant experimental Hi-C map from Hug et al. 2017 alongside our model prediction in the revised Figure S8a. We have also clarified this point in the main text as follows:

“Moreover, when we mutated Zld motifs in silico at the Bitesize locus, a site known to be affected by Zld depletion (Hug et al. 2017), the models predicted a local weakening of boundary insulation at the correct position in both NC12 and NC14 (Fig. S4-b). **While the effect is smaller than in the experimental data, it is notable that the model captures the expected direction and location of the change based solely on DNA sequence.**”

- In Figs 4 and S4, the authors train a model to find motifs and proteins that might affect structure maximally at NC14. They find that the M1BP motifs have the strongest effect. Did their model identify sequence motifs that are not targets of known TFs, potentially suggesting the existence of novel structural or regulatory proteins?

We address this question as part of the next comment.

- Still concerning this, one experimental suggestion would be to show whether mutation or downregulation of M1BP does decrease boundaries, since the effect seems to be much stronger than the boundary proteins that have been tested so far (4-fold stronger than CTCF for example).

We thank the reviewer for these insightful points.

Yes, our model did identify motifs that are not clearly linked to well-characterized transcription factors. One example we found particularly interesting is a motif potentially associated with sqz, a maternally deposited factor that has been described on FlyBase as having Pol II-related activity. We also observed consistent enrichment of C-rich motifs in both our TF motif enrichment analysis (Fig. 4c; Fig. S8e) and in our de novo motif analysis (Fig. 4d), which may point to lesser-studied or novel DNA-binding proteins. These findings open the door to future work exploring uncharacterized factors that may play a role in genome organization.

Regarding M1BP, we decided not to pursue it further experimentally in this study, as its motif is highly prevalent across the genome, which may partly explain its strong predicted effect. Additionally, two previous studies have already investigated M1BP function. One found that M1BP depletion led to loss of TAD structure (Bag et al. 2021), while another reported cell cycle arrest at M phase upon M1BP knockdown (Ramírez et al. 2018), complicating further interpretation.

Given these limitations, we instead focused on the combined depletion of the pioneer factors Zelda and GAF, whose region-specific effects have been well characterized (Hug et al. 2017; X. Li et al. 2023; Espinola et al. 2021). However, their simultaneous removal had not previously been tested in the context of chromatin structure. This approach, together with our clustering analysis, produced clear and interpretable results, showing marked reductions in insulation and loop strength specifically at loci associated with these factors.

- The authors use TOBIAS to try to identify differences in footprinting from ATAC-seq data in wt versus mutant fly embryos. Their analysis suggests that loss of Zld or of GAF is associated with deeper footprints for other factors. The authors conclude that this compensation in binding by other factors contribute to architectural resilience of chromatin. On the technical side, the

increased GAF footprint in Zlf- condition that is shown in Fig 5d does not seem to be very convincing, same for the Clamp footprint in GAF- condition shown in S5a. Some of the other changes also look rather small. What statistical testing was used to detect significance? Please comment.

We address this question as part of the next comment.

- Furthermore, the TOBIAS evidence is interesting, but it is important to provide at least one experimental confirmation, depleting Zld and showing that binding of at least one of the up factors corresponds to increased ChIP-seq or CUT&RUN binding at the Zld sites.

We agree that the TOBIAS footprinting analysis is difficult to interpret with confidence. In our updated analysis, we did not observe a clear compensatory strengthening of architectural features following loss of Zld or GAF (Fig. S10c) across other clusters; if anything, the changes were modest and tended toward slight decreases rather than gains.

We also explored published ChIP data from Zld or GAF-depleted embryos to validate these shifts but found that the results were complicated by confounding factors such as embryo lethality (Gaskill et al. 2021). Given these limitations, we have decided to remove the TOBIAS analysis from the revised manuscript.

Instead, we now focus on the newly generated data from the simultaneous depletion of Zld and GAF (Fig. 5; Fig. S10), which provides clearer and more direct evidence of their role in chromatin architecture. This approach has allowed us to identify more robust effects on boundary insulation and looping, and we believe it strengthens the overall conclusions.

Reviewer #3

(Remarks to the Author)

Changes in genome organization coincide with changes in chromatin state and transcription, but despite immense efforts, assigning cause-consequence relationships has remained challenging. Here, Maziak and colleagues refine the Micro-C assay to describe genome reorganization over the course of Zygotic Genome Organization in the fly embryo, followed by machine learning-based segmentation (using a large set of available genomics data on chromatin structure and transcription) and in-silico perturbation models to pinpoint instructive factors and elements.

Overall, this is an interesting study that combines a valuable modification to the Micro-C protocol (“pico-C”) to generate 3D maps at impressive resolution with the discovery of important leads about factors that have direct or indirect structuring functions in this process. My main reservation is that the large majority of the results are of a correlative nature, and do not distinguish between chromatin state and downstream changes in transcription (which, I think, should be considered separately). Considering the increasingly established impact of transcription on genome organization, further functional validation will improve our understanding of underlying mechanisms and may help to discriminate where 3D genome organization is instructive and where it's a consequence of changes in transcriptional activity.

We thank the reviewer for their thoughtful and constructive comments, and for highlighting both the technical advance of the Pico-C protocol and the biological insights gained into genome reorganization during Zygotic Genome Activation (ZGA).

We fully agree that establishing causality in genome organization remains a major challenge. To address this, we performed functional perturbation experiments in addition to our integrative computational analyses. Specifically, we inhibited transcription pharmacologically using α -amanitin and performed simultaneous maternal depletion of Zld and GAF – two key regulators of early genome activation. These perturbations allowed us to directly test the impact of transcriptional activity and pioneer factor binding on 3D genome architecture.

Our results show that while transcription inhibition leads to widespread disruption of gene expression, the overall 3D genome organization remains largely intact at early stages, suggesting that transcription is not the primary driver of initial genome structuring. In contrast, depletion of Zld and GAF leads to pronounced defects in domain formation and long-range interactions, supporting a more instructive role for these factors in establishing 3D genome architecture.

These findings are now described in detail in the revised Results section and discussed in the context of causality in genome organization (pages 7-9). We believe they provide strong evidence that our approach goes beyond correlation and begins to dissect causal relationships between chromatin state, transcription, and 3D genome structure.

We hope these additions address the reviewer’s concerns and clarify the mechanistic implications of our findings.

Major comments:

1. Whereas I consider the Pico-C assay an elegant refinement of the Micro-C protocol, it will benefit from additional description and benchmarking (beyond the cis-trans and cis-local values provided in Table S1). Benchmarking: considering that high quality conventional Micro-C data from NC14 is available (Batut et al, 2022), how do their own data compare in resolution,

sequencing depth (including useful read pairs), number of required library preparations (from what I gather in Table S1, quite many for most stages?), etc. Regarding detail, from the Material and Methods section, I gather that Pico-C requires around 60,000 nuclei. Providing this number in the main text, rather than the undefined “low-input” description, will be helpful.

We thank the reviewer for their insightful suggestions regarding the benchmarking and description of the Pico-C protocol.

To facilitate a more direct comparison with existing datasets, we have now added a quantification of total valid pairs across our dataset and the published Micro-C data from Batut et al. (2022), now included in the updated Table S1. Notably, while Batut et al. used ~300 embryos per library, our protocol achieves comparable resolution using significantly fewer nuclei per preparation (~60,000 nuclei from 10 embryos at NC14, now explicitly stated in the main text).

As the reviewer correctly notes, our dataset includes a relatively large number of library preparations, particularly at NC13 and NC14. This reflects an extended benchmarking phase aimed at optimizing the biotin fill-in step — a key bottleneck in adapting Micro-C to low-input embryonic material. While these libraries were generated under varying biotinylation conditions (e.g., 3 vs. 4 hours), all were subject to the same quality control criteria, and only those meeting our thresholds were included in downstream analyses (Table S1). Despite these differences, PCA analysis shows clear separation by developmental stage, indicative of the robustness of the data (Fig. S1a).

We have clarified these points in the revised Methods and Results sections and hope this additional detail addresses the reviewer’s request for more comprehensive benchmarking and protocol transparency.

2. The core of the manuscript relies on a staged segmentation of the genome, using the large number of datasets that are provided in Table S3. Not all data is available for all timepoints (or for the Zld- condition), which will introduce biases in the emitted ChromHMM states and stage specific transitions. How do the authors think this influences the outcomes? They should discuss the potential impact of this limitation (from line 188?) and consider, if essential, to generate missing data sets.

We thank the reviewer for raising this important point regarding the completeness of the datasets used for ChromHMM segmentation.

We agree that the absence of certain chromatin marks at specific time points and in the Zeld-depleted condition introduces potential biases in the emission of ChromHMM states and their transitions. To acknowledge this, we have revised the relevant paragraph in the main text (starting at line 166) to begin with:

“To contextualize the broader chromatin state across early development, we applied ChromHMM to integrate diverse epigenomic data into 20 chromatin states, aiming to map broad chromatin features across early development despite some dataset gaps.”

To assess the potential impact of missing data, we performed additional ChromHMM segmentations using only the subset of chromatin marks that are consistently available across all developmental stages. These reduced-input models recovered highly similar core state structures, including promoter-like, transcription-associated, HP1-enriched, and genic states. This suggests that the segmentation is robust to the missing data and that the major chromatin features are reliably captured.

While we acknowledge that a fully complete dataset would be ideal, we do not currently consider it essential to generate additional chromatin profiles for the purposes of this segmentation. Throughout the manuscript, we have been careful not to over-interpret the ChromHMM output, instead using it as a framework to explore broad chromatin patterns and validating key trends directly against the underlying data.

We hope this addresses the reviewer's concern and clarifies the rationale behind our approach.

3. The staged reorganization of the HDAC1 locus, as elegantly analyzed and visualized in Fig. 3, suggests a dynamic and coordinated multi-component process. Yet, the correlative nature of the investigations precludes the identification of causative events (e.g. line 209). The transition of the pre-major ZGA to major ZGA appears associated with a major reorganization of transcription at this locus, which raises the important question if 3D genome reorganization (mediated by chromatin states?) instructs this transcriptional reorganization, or rather if changes in genome organization are a consequence. Combined with the emerging importance of Paused RNA PolII in genome organization (cited Barshad et al, 2023, also Ghavi-Helm et al, 2014), the understanding of mechanisms (this figure, and possibly the remainder of the study) will strongly gain in impact if the authors can add insights on Paused RNA PolII distribution (e.g. Ser5-ChIP-seq) or ongoing transcription (e.g. PRO-seq) and ideally Pico-C data where transcription (or initiation to elongation) is inhibited at certain stages.

We thank the reviewer for their insightful comments regarding the causal relationship between 3D genome reorganization and transcriptional activation.

To begin addressing this, we performed transcriptional inhibition experiments by injecting α -amanitin into early embryos before the minor-wave of ZGA and subsequently conducting pico-C (Fig. 3g, Fig. S7b,d). As expected, this treatment led to a global reduction in zygotic transcription. However, consistent with our previous findings (Hug et al. 2017), the overall 3D genome architecture remained largely intact, suggesting that early genome organization is not solely dependent on transcriptional elongation.

Interestingly, we observed that a subset of early loops persisted even under transcriptional inhibition. Closer inspection revealed that these loops often correspond to pre-MBT genes previously reported to be transcribed early with low pausing indices (Chen et al. 2013). This raises the possibility that loop persistence may not directly reflect paused Pol II activity per se, but could instead be due to stabilization of Pol II at these loci — potentially even in the absence of productive elongation. However, this does not fully explain why some loops are retained while others are not, highlighting the need for future studies to dissect how Pol II occupancy, pausing, and local chromatin context contribute to loop stability.

At the global level, we found that chromatin clusters marked by strong H3K4me3 were the only ones to show modest but consistent architectural reduction upon transcriptional inhibition. This is particularly intriguing in light of a recent preprint suggesting that α -amanitin treatment can reduce H3K4me3 signal at certain loci (Oak et al. 2025), potentially linking transcriptional inhibition to chromatin state changes that secondarily affect 3D structure.

Together, these findings suggest that while active transcriptional elongation is not strictly required to maintain early architectural features, there may be a threshold of Pol II occupancy or chromatin context that supports loop formation or maintenance. This aligns with recent work by Cardamone et al., which demonstrated that even when transcription was globally arrested

through CBP depletion, chromatin accessibility still advanced, likely facilitated by the presence of stalled RNA Polymerase II (Cardamone et al. 2025). We agree that further investigation — including direct profiling of paused Pol II (e.g., Ser5P ChIP-seq) or nascent transcription (e.g., PRO-seq) — will be essential to fully resolve these mechanisms.

In a similar vein, the identification of robustness to pioneer factor loss (Fig. 5) is an interesting result, but can't be uncoupled from a (potential) drastic effect on either global ZGA dynamics or more gene-specific transcription changes. Could the identified compensatory factors modulate transcriptional output in a more subtle manner than picked up by the used segmentation approach? Could the authors investigate this using the available RNA-seq in the Zld-background? Moreover, this result raises the question how Zld and GAF can influence binding of other factors. Do these footprints localize close to Zld and GAF binding sites (if I understand correctly, they don't overlap)? Could a potential distance-dependent effect be extracted from the available data?

We appreciate this thoughtful question. As noted in our response to Reviewer 2, we have removed the footprinting analysis, as we found it difficult to interpret functionally in this context. Instead, we pursued a more direct approach by performing simultaneous depletion of Zld and GAF and generating new Pico-C data in this genetic background. These two factors have well-characterized and largely non-overlapping binding profiles in early embryos, and in the double depletion, we observe additive disruptions in insulation and looping at loci typically associated with either Zld or GAF activity (new Fig. 5).

This pattern is consistent with the idea that Zld and GAF act independently at their respective targets, and that their combined loss leads to parallel architectural defects. While GAF and Zld typically bind independently in early embryos, their interplay cannot be excluded. A recent preprint (Fallacaro et al. 2024) shows that disrupting Zelda's DNA-binding domain led to re-localization of the mutant protein to GAF-binding sites, with increased enrichment at regions containing GAF motifs. Moreover, a subset of these sites showed increased chromatin accessibility following either Zld depletion or DNA-binding disruption, suggesting that transcription factors may redistribute to GAF sites when Zld activity is compromised. This implies that while these factors act independently, their regulatory environments are not entirely separated, and perturbation of one can influence the activity or binding landscape of others. We suspect that aspects of this effect were reflected in the TOBIAS footprinting profiles we initially observed.

Although we think this opens interesting future directions, it is beyond the scope of this study. Notably, even if other factors were compensating at the binding level, we did not observe a global increase in insulation or looping in the double mutant, as shown by our Chromosight analysis, although we agree that compensatory effects at the transcriptional level could be more subtle than captured by our segmentation approach.

Minor remarks:

- The addition of arrowheads in the contact maps in Fig. 1C and 3A,C will help to highlight relevant structures. I'm not familiar with "tethering elements" (Fig. 1C; Antp gene). This will benefit from additional detail in the text (line 112).

We have added numeral and highlighted the underlying genes associated with these structures in red.

- Line 115: it has remained controversial if 3C-based assays can describe chromatin compaction. Better to rephrase.

We have removed this to simply state:

“Additionally, the temporal resolution allows us to detect genome-wide trends in contact probability, revealing shorter-range enrichment at NC9-10 and increased interaction frequencies at longer distances (~1Mb) at stages NC12-14 (Fig. S1c).”

- Line 122: reduced at NC14, not lost?

We have changed this accordingly.

- Line 140: why did the authors decide on 20 states? Was this rationalized?

We have added clarification to the ChromHMM methods section. We tested multiple models with varying numbers of states and selected the 20-state model because it provided the best balance of model fit and interpretability. This model produced clear, biologically meaningful transitions between states.

- Line 203: a statistical analysis based on the Global Moran I is not conventionally used. It will benefit from some additional detail in the text. From what I gather, it's not used to quantify the degree of spatial clustering, but it is the measure itself.

The Global Moran's I does not only reflect the pattern (clustered, random, dispersed), but directly measures the extent of the autocorrelation. In the case of statistically significant results, a higher absolute Moran's I indicates stronger clustering/dispersion of the signal of interest.

- From line 223 and Fig. 3B,D: I'm not sure if I understand correctly: are these the Global Moran I values that were determined genome wide? Is it relevant to interpret these values for differences at the single locus?

These are Global Moran's I values calculated specifically at the locus in question, not genome-wide. We believe it is appropriate to interpret these values at the single-locus level, as the method was developed to capture spatial autocorrelation within individual genomic regions (Kim et al. 2025). In this case, it effectively highlights the dynamic regulation at this locus.

- Line 256: chromatin and transcriptional landscape

We edited this accordingly.

- Fig. 1D: are these stage-specific boundaries, or the same set of boundaries for all time points? Would it be insightful to add aggregates for lost/reduced boundaries as well?

These are aggregates boundary calls at their relevant stage, we have now added the number of boundaries being plotted to clarify this.

In principle, we agree that aggregates of lost or reduced boundaries could provide an additional layer of visualization. However, in practice, this subset represents already a very small fraction of total boundaries. While some cases do show clear differences, others are less specific or are found in regions where nearby boundaries are gained, making the aggregated signal more difficult to interpret reliably. As a result, we chose to focus on the more robust categories where changes are clearer and more consistent.

- Fig. 1F: add information on the number of loops in each distance category, and possibly distribution of sizes within each category?

We have added the number of loops (n values) for each distance category in Fig. 1F, as requested. Additionally, we now include the distribution of loop sizes within each cluster category, as this provides further biological insight into the nature of the loops.

- Fig. 4D: could the authors add the size of the genomic interval that is visualized (is this 20kb resolution in the figure as well)?

We have added the relevant information to the figure; it is 1 kb resolution (that of the models) with 10 kb flank-size.

- Fig. 5D and S5: what does “position” mean in these graphs? Are these pile-ups of footprints around all GAF and BEAF-32 peaks, as present in untreated cells? These panels will benefit from additional detail.

We have removed this panel as mentioned above (see response to Reviewer 2’s last two comments).

- Bag, Indira, Shue Chen, Leah F. Rosin, Yang Chen, Chen-Yu Liu, Guo-Yun Yu, and Elissa P. Lei. 2021. “M1BP Cooperates with CP190 to Activate Transcription at TAD Borders and Promote Chromatin Insulator Activity.” *Nature Communications* 12 (1): 4170. <https://doi.org/10.1038/s41467-021-24407-y>.
- Batut, Philippe J., Xin Yang Bing, Zachary Sisco, João Raimundo, Michal Levo, and Michael S. Levine. 2022. “Genome Organization Controls Transcriptional Dynamics during Development.” *Science* 375 (6580): 566–70. <https://doi.org/10.1126/science.abi7178>.
- Blythe, Shelby A., and Eric F. Wieschaus. 2015. “Zygotic Genome Activation Triggers the DNA Replication Checkpoint at the Midblastula Transition.” *Cell* 160 (6): 1169–81. <https://doi.org/10.1016/j.cell.2015.01.050>.
- Cardamone, Francesco, Annamaria Piva, Eva Löser, Bastian Eichenberger, Mari Carmen Romero-Mulero, Fides Zenk, Emily J. Shields, et al. 2025. “Chromatin Landscape at Cis-Regulatory Elements Orchestrates Cell Fate Decisions in Early Embryogenesis.” *Nature Communications* 16 (1): 3007. <https://doi.org/10.1038/s41467-025-57719-4>.
- Chen, Kai, Jeff Johnston, Wanqing Shao, Samuel Meier, Cynthia Staber, and Julia Zeitlinger. 2013. “A Global Change in RNA Polymerase II Pausing during the *Drosophila* Midblastula Transition.” Edited by Ruth Lehmann. *eLife* 2 (August):e00861. <https://doi.org/10.7554/eLife.00861>.
- Duan, Jingyue, Leila Rieder, Megan M Colonna, Annie Huang, Mary Mckenney, Scott Watters, Girish Deshpande, William Jordan, Nicolas Fawzi, and Erica Larschan. 2021. “CLAMP and Zelda Function Together to Promote *Drosophila* Zygotic Genome Activation.” Edited by Yukiko M Yamashita and Kevin Struhl. *eLife* 10 (August):e69937. <https://doi.org/10.7554/eLife.69937>.
- Espinola, Sergio Martin, Markus Götz, Maelle Bellec, Olivier Messina, Jean-Bernard FICHE, Christophe Houbon, Matthieu Dejean, et al. 2021. “Cis-Regulatory Chromatin Loops Arise before TADs and Gene Activation, and Are Independent of Cell Fate during Early *Drosophila* Development.” *Nature Genetics* 53 (4): 477–86. <https://doi.org/10.1038/s41588-021-00816-z>.
- Fallacaro, Samantha, Apratim Mukherjee, Puttachai Ratchasanmuang, Joseph Zinski, Yara I Haloush, Kareena Shankta, and Mustafa Mir. 2024. “A Fine Kinetic Balance of Interactions Directs Transcription Factor Hubs to Genes.” *bioRxiv*, January, 2024.04.16.589811. <https://doi.org/10.1101/2024.04.16.589811>.
- Gaskill, Marissa M, Tyler J Gibson, Elizabeth D Larson, and Melissa M Harrison. 2021. “GAF Is Essential for Zygotic Genome Activation and Chromatin Accessibility in the Early *Drosophila* Embryo.” Edited by Yukiko M Yamashita and Kevin Struhl. *eLife* 10 (March):e66668. <https://doi.org/10.7554/eLife.66668>.
- Hannon, Colleen E, Shelby A Blythe, and Eric F Wieschaus. 2017. “Concentration Dependent Chromatin States Induced by the Bicoid Morphogen Gradient.” Edited by Joaquín M Espinosa. *eLife* 6 (September):e28275. <https://doi.org/10.7554/eLife.28275>.
- Huang, Shao-Kuei, Peter H. Whitney, Sayantan Dutta, Stanislav Y. Shvartsman, and Christine A. Rushlow. 2021. “Spatial Organization of Transcribing Loci during Early Genome Activation in *Drosophila*.” *Current Biology* 31 (22): 5102-5110.e5. <https://doi.org/10.1016/j.cub.2021.09.027>.
- Hug, Clemens B., Alexis G. Grimaldi, Kai Kruse, and Juan M. Vaquerizas. 2017. “Chromatin Architecture Emerges during Zygotic Genome Activation Independent of Transcription.” *Cell* 169 (2): 216-228.e19. <https://doi.org/10.1016/j.cell.2017.03.024>.

- Kim, Iana V., Cristina Navarrete, Xavier Grau-Bové, Marta Iglesias, Anamaria Elek, Grygoriy Zolotarov, Nikolai S. Bykov, et al. 2025. “Chromatin Loops Are an Ancestral Hallmark of the Animal Regulatory Genome.” *Nature* 642 (8069): 1097–1105. <https://doi.org/10.1038/s41586-025-08960-w>.
- Li, Xiao, Xiaona Tang, Xinyang Bing, Christopher Catalano, Taibo Li, Gabriel Dolsten, Carl Wu, and Michael Levine. 2023. “GAGA-Associated Factor Fosters Loop Formation in the *Drosophila* Genome.” *Molecular Cell* 83 (9): 1519-1526.e4. <https://doi.org/10.1016/j.molcel.2023.03.011>.
- Li, Xiao-Yong, Melissa M Harrison, Jacqueline E Villalta, Tommy Kaplan, and Michael B Eisen. 2014. “Establishment of Regions of Genomic Activity during the *Drosophila* Maternal to Zygotic Transition.” Edited by Robb Krumlauf. *eLife* 3 (October):e03737. <https://doi.org/10.7554/eLife.03737>.
- Munden, Alexander, Mary Lauren Benton, John A. Capra, and Jared T. Nordman. 2022. “R-Loop Mapping and Characterization During *Drosophila* Embryogenesis Reveals Developmental Plasticity in R-Loop Signatures.” *Journal of Molecular Biology* 434 (13): 167645. <https://doi.org/10.1016/j.jmb.2022.167645>.
- Oak, Meghana S., Marco Stock, Matthias Mezes, Tobias Straub, Antony M. Hynes-Allen, Jelle van den Aamele, Ignasi Forne, et al. 2025. “H3K4 Methylation-Promoted Transcriptional Memory Ensures Faithful Zygotic Genome Activation and Embryonic Development.” *bioRxiv*, January, 2025.01.20.633863. <https://doi.org/10.1101/2025.01.20.633863>.
- Ogiyama, Yuki, Bernd Schuettengruber, Giorgio L. Papadopoulos, Jia-Ming Chang, and Giacomo Cavalli, dirs. 2018. *Polycomb-Dependent Chromatin Looping Contributes to Gene Silencing during Drosophila Development*. Vol. 71. United States. <https://doi.org/10.1016/j.molcel.2018.05.032>.
- Ramírez, Fidel, Vivek Bhardwaj, Laura Arrigoni, Kin Chung Lam, Björn A. Grüning, José Villaveces, Bianca Habermann, Asifa Akhtar, and Thomas Manke. 2018. “High-Resolution TADs Reveal DNA Sequences Underlying Genome Organization in Flies.” *Nature Communications* 9 (1): 189. <https://doi.org/10.1038/s41467-017-02525-w>.
- Schulz, Katharine N, Eliana R Bondra, Arbel Moshe, Jacqueline E Villalta, Jason D Lieb, Tommy Kaplan, Daniel J McKay, and Melissa M Harrison. 2015. “Zelda Is Differentially Required for Chromatin Accessibility, Transcription Factor Binding, and Gene Expression in the Early *Drosophila* Embryo.” *Genome Res.* 25 (11): 1715–26. <https://doi.org/10.1101/gr.192682.115>.
- Zenk, Fides, Eva Loeser, Rosaria Schiavo, Fabian Kilpert, Ozren Bogdanović, and Nicola Iovino. 2017. “Germ Line–Inherited H3K27me3 Restricts Enhancer Function during Maternal-to-Zygotic Transition.” *Science* 357 (6347): 212–16. <https://doi.org/10.1126/science.aam5339>.
- Zenk, Fides, Yinxiu Zhan, Pavel Kos, Eva Löser, Nazerke Atinbayeva, Melanie Schächtle, Guido Tiana, Luca Giorgetti, and Nicola Iovino. 2021. “HP1 Drives de Novo 3D Genome Reorganization in Early *Drosophila* Embryos.” *Nature* 593 (7858): 289–93. <https://doi.org/10.1038/s41586-021-03460-z>.

Response to Reviewers

We sincerely thank the reviewers for their insightful comments and constructive suggestions, which have played an important role in improving the clarity and quality of the manuscript.

Below, we provide a point-by-point response to the reviewers' comments. For visual ease, we have marked reviewer comments in blue italic typography while leaving our replies in black.

Reviewer Comments:

Reviewer #1

(Remarks to the Author):

The authors majorly revised the manuscript by updating or clarifying analyses and incorporating two new major Pico-C (and accompanying RNA-seq or ATAC-seq) datasets to describe effects of transcription elongation inhibition or of double depletion of pioneer factors. The data are of high quality. The authors' conclusions paint a picture of early genome folding being surprisingly dynamic over cleavage cycles, and suggesting – and partially demonstrating through perturbation experiments – that it is shaped by diverse parallel (and in some cases synergistic) mechanisms. The value of the work therefore lies in the clarity of the genome folding maps in many successive interphases of early embryo cleavage cycles which enabled a new appreciation of the dynamics and diversity of loci involved in 3D genome folding – although other mechanisms driving folding remain to be clarified.

We thank the reviewer for their thoughtful and positive assessment of our revised manuscript. We agree that additional mechanisms underlying genome folding remain to be clarified in future studies.

Below are remaining minor comments.

1. Line 72: This sentence does not capture the conclusion of the referenced papers accurately. Depletion of Cpl90 weakens or abolishes almost one quarter of all TAD boundaries, which seems contradictory with the statement that “removal of insulator proteins shows no genome-wide changes”.

We appreciate the reviewer's careful reading of this section. We have now revised the sentence (line 72) to:

“Similarly, the removal of insulator proteins shows only local, boundary-specific changes (Kaushal et al. 2022; Cavalheiro et al. 2023).”

Our initial wording was biased on the data from Cavalheiro et al. 2023 (17% boundaries change with only ~8% exhibiting a difference detectable by eye), who analyzed NC14 embryos, which are staged more comparably to our own data whereas Kaushal et al. (2022), used a broader developmental window (2–6h embryos). We believe the revised sentence now appropriately reflects both studies.

2. Line 140: Clarifying that only a subset (not all) meta-loops are captured in early embryos can avoid confusion with the fact that most meta-loops only form in differentiated neurons.

We thank the reviewer for this comment. We have revised the wording in line 140 to clarify that only a subset of meta-loops are captured in early embryos. We also note that some loops may be embryo-specific, which we believe makes these findings particularly exciting. The revised text is below:

“Interestingly, our maps also capture a subset of the previously characterized meta-loops (Mohana et al. 2023) as well as additional meta-loop-like features, with some detectable as early as NC13 (Fig. S2c).”

3. Line 348: The use of “remarkably” was not fully clear. It was arguably expected to observe architectural disruptions at GAF-associated features in the double knock-down as these were previously reported (Li et al. 2023), though Zld-dependent folding defects are novel.

We thank the reviewer for this comment. We have clarified the sentence in line 348. Our intention was not to suggest that disruptions at GAF-associated features were unexpected, as these have indeed been reported (Li et al. 2023), but rather to highlight the co-occurrence of disruptions at both Zld- and GAF-associated features in the same double knockdown embryos. We believe the revised wording makes this point clearer and adds the references to avoid any confusion:

“Notably, highly resolved Pico-C maps on these mutants revealed that architectural disruptions at both Zld- (Hug et al. 2017; Espinola et al. 2021) and GAF-associated features (Li et al. 2023) co-occur in the double knockdown (DKD; Fig. 5e Fig S10b).”

4. Fig. 3A legend: Replace second “left” by e.g. “center”.

We thank the reviewer for catching this. We have corrected the Fig. 3A legend.

5. Fig. 5A: Could the authors comment on the somewhat unexpected fact that pioneer states are more enriched at loop anchors or boundaries after Zld depletion.

We thank the reviewer for raising this point. Figure 5a originally showed local enrichment rather than genome-wide enrichment. This approach was chosen to highlight context-specific patterns around anchors and boundaries while avoiding artificial enrichment of broadly distributed states. The apparent increase of Zelda [2] after depletion should therefore not be interpreted as a biological gain, but as a relative effect of the local enrichment analysis. As shown in Table S4, both Zelda states [1] and [2] are strongly depleted in Zld-depleted ChromHMM, consistent with expectations. Because Zelda states are generally lost after depletion, the surrounding regions often lack Zelda signal, making even small residual segments of Zelda [2] appear locally enriched. In addition, both Zelda states share emission probabilities with ATAC-seq signal, and Zelda [2] also incorporates features such as H2Av, H4K16ac, and RNA-seq. Consequently, a small number of regions may retain or be reassigned to Zelda [2] even in the absence of Zelda binding, due to these additional features.

Given that panels a and b are not central to the main message of the section, and to avoid any potential misinterpretation of the data, we have now moved these to Fig. S9a–c. The text has been clarified as follows:

“This analysis revealed that promoter-like states remain enriched at loop anchors and boundaries relative to their local genomic context even in Zld knockdown ChromHMM (Fig. S9a).”

6. Fig. 5E: “GAF-associated”

We have corrected panel 5E.

Reviewer #2

(Remarks to the Author):

Maziak et al have extensively revised their manuscript, which is an important contribution that should ultimately be published.

I have one last point, related to comment 3 raised by reviewer #3, concerning experiments required to disentangle the effects of transcription from that of specific TFs. The authors have partly addressed this by performing transcription elongation inhibition by alpha-amanitin and then performing Pico-C. I think this could be complemented by inhibiting initiation using triptolide and providing Ser5-ChIP-seq or PRO-seq and Pico-C with or without inhibition. This would be very interesting and important irrespective of the results that will be obtained.

We thank the reviewer for the positive evaluation of our work.

Regarding the proposed experiments, we note that we have already performed similar experiments (using Hi-C rather than Pico-C) in Hug et al., Cell 2017 (PMID: 28388407). In that study, we treated embryos with triptolide and α -amanitin, performed ChIP-seq for pan Pol II and Ser5-Pol II (Fig. S5G in that paper), and conducted Hi-C (Fig. 5 in that paper). The effects observed in α -amanitin-treated embryos using Pico-C in our current manuscript are consistent with those seen using Hi-C. Moreover, the Hi-C data from triptolide-treated embryos closely resembles the α -amanitin-treated ones, as analyzed in detail in our previous publication. We therefore expect the reviewer's proposed experiments to yield results consistent with our existing findings using α -amanitin inhibition, reinforcing rather than extending the conclusions we present here.

Moreover, in the revised manuscript we have done extensive work strengthening our analysis of transcription versus transcription factor regulation by introducing a systematic cluster-based analysis of domain boundaries and loops, together with transcription inhibition and double pioneer factor depletion experiments. These analyses clearly demonstrate differential and specific effects across clusters depending on the perturbation type, thereby disentangling the contributions of transcription and transcription factor regulation, at least for Zelda and GAF.

While we agree that the role of paused RNA Pol II and pre-initiation complex formation is a fascinating topic, as highlighted by the original comment by Reviewer #3 and one we previously discussed in Hug et al., 2017, fully addressing it would require additional genetic or acute depletion approaches beyond pharmacological inhibition, which we consider outside the scope of the present manuscript.

To further address the reviewer's comment, we have included direct comparisons at representative loci showing that the retained looping features captured by Pico-C are also evident in triptolide-treated embryos (Hug et al., 2017) (Figure 1 below for the reviewer, and Supplemental Figure 7e). In addition, we applied Chromosight on NC14 loops and boundaries and our cluster-based analysis to the triptolide Hi-C data and observed similar results (Figure 2 and 3 for the reviewer below). Finally, we have expanded the discussion in the manuscript to further highlight the potential role of the pre-initiation complex in shaping 3D genome organization:

“For example, loop anchors in our analysis can belong to multiple regulatory clusters despite looping together, implying that this is not governed solely by the presence of specific factors, but also potentially by the regulatory history and neighborhood context of a locus. **In this light, it is notable that depletion of Zld and GAF produces stronger architectural effects than blocking transcription with α -amanitin. One possible explanation is that pioneer factors act upstream in establishing chromatin conformation, whereas transcription represents a more downstream process. It will be interesting in the future to target the pre-initiation complex with more refined genetic tools to continue examining the interplay of chromatin conformation and landscape (Ghavi-Helm et al. 2014; Hug et al. 2017; Barshad et al. 2023).** These genetic perturbations combined with temporal data will be particularly powerful in dissecting the role of hysteresis in genome architecture, further elucidating the interplay of factor specificity, order and timing of regulatory events, and conformation.”

Figure 1 | (A,B) Pico-C α -amanitin-treated results recapitulate the effects observed with Hi-C in α -amanitin- and triptolide-treated embryos at a higher resolution.

Figure 2 | ChromSight output on Pico-C and Hi-C data at all scorable NC14 boundaries (A) and loops (B) captured in this study, carried out at 4 kb resolution. The resolution was selected to maximize the capture of most features across the two data types. X- and Y-axes correspond to pixels.

Figure 3 | Pico-C α -amanitin-treated data from this study (A) show similar patterns of Chromosight analysis compared to triptolide-treated embryos (B; Hug et al., 2017), highlighting consistent effects of transcription inhibition across datasets. Similar boundary clusters enriched with H3K4me3 are affected across both treatments.

Reviewer #3

(Remarks to the Author):

In their revised manuscript, Maziak and colleagues have taken considerable efforts to address the comments from the reviewers. I particularly appreciate the results from the Zelda/GAF depletion and α -Amanitin experiments, which provide important insights into cause/consequence of gene regulation/TF binding and changes in transcription. Except for a few minor remarks, I now consider the manuscript a good fit for publication in Nature Genetics.

We thank the reviewer for their positive assessment of our revised manuscript. We are glad that these results provided important mechanistic insights, and we appreciate the reviewer's recognition of our efforts to address the comments.

Minor remarks:

- The Chromosight analysis of boundaries and loops (Fig. 3h, 5f and associated supplementary panels) provides valuable insight into the extent of chromatin reorganization. Considering that the analysis is pairwise, between control and treated conditions, it would be more informative to use a visualization that retains this information (e.g. something like this: <https://stackoverflow.com/questions/49370705/implementing-paired-lines-into-boxplot-ggplot2>)

We thank the reviewer for this helpful suggestion. We originally tried paired-line visualizations but given the large number of loop-anchors and boundaries, the plots quickly became overcrowded, and the overall trend was difficult to interpret.

To solve this, we instead plotted the per-feature normalized Chromosight score change between control and treated/mutant conditions. Specifically, for each loop anchor and each boundary, we computed a normalized change index from the Chromosight scores and visualized the resulting distributions. This per-feature approach allowed us to capture the pairwise information while avoiding the heavy overplotting of paired-line plots, and it provides a clearer view of the overall trend in how individual anchors and boundaries shift between conditions. This is now found in Supplemental Figures 7 (panels g and i) and 10 (d and f).

- As mentioned, the addition of Zelda/GAF depletion and α -Amanitin studies provides important insights into the underlying mechanisms. I'm intrigued to see how much stronger the impact of the Zelda and GAF pioneer factors are, as compared to transcription itself. This difference is briefly mentioned in the abstract and introduction. I think it merits attention, in more detail, in the discussion section as well.

We thank the reviewer for this helpful suggestion. We agree that it is indeed very interesting, especially in the context of genome structure establishment during early embryonic development. We have added this now in the discussion as follows:

“For example, loop anchors in our analysis can belong to multiple regulatory clusters despite looping together, implying that this is not governed solely by the presence of specific factors, but also potentially by the regulatory history and neighborhood context of a locus. **In this light, it is notable that depletion of Zld and GAF produces stronger architectural effects than blocking transcription with α -amanitin. One possible explanation is that pioneer factors act upstream in establishing chromatin conformation, whereas transcription represents a more downstream process. It will be interesting in the future to target the pre-initiation complex**

with more refined genetic tools to continue examining the interplay of chromatin conformation and landscape (Ghavi-Helm et al. 2014; Hug et al. 2017; Barshad et al. 2023). These genetic perturbations combined with temporal data will be particularly powerful in dissecting the role of hysteresis in genome architecture, further elucidating the interplay of factor specificity, order and timing of regulatory events, and conformation.”

- The references to Supplementary figure 1 are not completely accurate? Line 116: Fig. S1c-d? Line 125: Fig. S1e-f?

We thank the reviewer for pointing this out. We carefully checked the references to Supplementary Figure 1 and corrected them as suggested.